# A brain-to-small intestine circuit mediates morphine-induced constipation in male mice

Jun Ma[1,7], Xiaoqi Peng[2,3,7], Mingjun Zhang[4,7], Wei Gao[1], Xinlu Yang[1], Zerui Wang[5], Xiaoqing Chai[1] ✉, Zhi Zhang [1,2,3,6] ✉, Sheng Wang[1] ✉ & Peng Cao [4,6] ✉

Opioid-induced constipation is one of the most common and persistent side effect of opioid analgesics, yet the underlying neural mechanism(s) remain unclear. Here we show morphine-induced constipation is mediated by a neural circuit from glutamatergic neurons in the paraventricular nucleus of hypothalamus (PVN[Glu]) to acetylcholinergic neurons in the dorsal motor nucleus of the vagus (DMV[Ach]), and subsequently to the small intestine in mice. Microendoscopic calcium imaging revealed morphine inhibits the PVN[Glu]→DMV[Ach]→small intestine circuit, and this is accompanied by decreased small intestinal motility. Chemogenetic activation of this circuit, as well as pharmacological inhibition or knockdown of the μ-opioid receptor (MOR) in PVN[Glu] neurons alleviates morphine-induced constipation. Conversely, artificial inhibition of this circuit mimics morphine-induced constipation in naïve mice. Moreover, we show that morphine suppresses tonic NMDA receptor-mediated currents in DMV[Ach] neurons. These findings reveal a brain-gut circuit underlying opioid-induced constipation and suggest potential therapeutic strategies to mitigate this debilitating side effect.

Opioids have been a cornerstone of clinical pain management for several decades. Prolonged use of opioids and opioid-based analgesics, such as morphine and fentanyl, often leads to severe side effects, including gastrointestinal dysfunction[1–3]. In clinical patients with non-cancer pain, the incidence of opioid-induced constipation ranges from 40% to 80%[4,5]. However, the underlying mechanism of opioid-induced constipation is not entirely known.

Current treatments for opioid-induced constipation involve dietary adjustments, exercise, laxatives, and antagonistic medications[6]. Studies have shown that μ-opioid receptor (MOR) antagonists are effective in treating opioid-induced constipation[7]. The finding that activation of MOR in intestinal cells contributed to opioid-induced constipation has motivated the development of peripherally acting MOR antagonists (PAMORAs) as potential interventions. However, clinical data indicate that PAMORAs are only about 50% effective in treating opioid-induced constipation[8–10]. Of note, intracerebroventricular injection of the MOR antagonist naloxone has also been shown to improve opioid-induced constipation[11], suggesting a role for the brain-gut axis in this condition[12,13]. The precise cell type-specific organization and the function(s) of the brain-to-gut circuit mediating opioid-induced constipation remain largely unknown.

In this study, we established a morphine-induced constipation mouse model, characterized by small intestinal dysfunction, including reductions in fecal output, fecal water content, the motility and the transit rate of small intestines. By combining viral tracing, in vivo microendoscopic calcium imaging, and wireless optogenetics, we

[1]Department of Anesthesiology, The First Affiliated Hospital of USTC, Hefei National Laboratory for Physical Sciences at the Microscale, Division of Life Sciences and Medicine, University of Science and Technology of China, Hefei, China. [2]Department of Pain Medicine, The First Affiliated Hospital of Anhui Medical University, Hefei 230022, China. [3]School of Basic Medical Sciences, Anhui Medical University, Hefei, China. [4]Department of Endocrinology and Metabolism, The First Affiliated Hospital of USTC, Division of Life Sciences and Medicine, University of Science and Technology of China, 230026 Hefei, China. [5]Hefei No.1 High School, Heifei, China. [6]Department of Biophysics and Neurobiology, Key Laboratory of Brain Function and Disease of Chinese Academy of Sciences, University of Science and Technology of China, Hefei, China. [7]These authors contributed equally: Jun Ma, Xiaoqi Peng, Mingjun Zhang. ✉e-mail: xiaoqingchai@ustc.edu.cn; zhizhang@ustc.edu.cn; iamsheng2020@ustc.edu.cn; pengcao@ustc.edu.cn

characterized the functional organization of a neural circuit connecting glutamatergic neurons in the paraventricular nucleus of the hypothalamus (PVN[Glu]) to acetylcholinergic neurons in the DMV (DMV[Ach]) to the small intestine. Further, we demonstrated morphine inhibits PVN[Glu] neuronal activity through the MOR, which in turn leads to suppression of its tonic NMDA receptor-mediated currents to DMV[Ach] neurons, and ultimately, the overall hypoactivity of this circuit mediates small intestinal dysfunction. These findings advance our understanding of the brain-gut neural mechanisms underlying opioid-induced constipation.

## Results

### Morphine suppresses small intestinal motility to induce constipation in mice

A mouse model of morphine-induced constipation was established through intraperitoneal injection of morphine in C57 mice[14]. Given reports about potential instability for this model[15], we initially examined the effects of different morphine doses (3 mg/kg, 6 mg/kg, and 10 mg/kg) on constipation (Fig. 1a, b and Supplementary Fig. 1a). Significant dose-dependent reductions in fecal output were observed in the morphine-treated mice compared to saline-treated controls

(Fig. 1c). In addition, the fecal water content was markedly decreased within 6 hours of morphine administration (Fig. 1d). Given that 10 mg/kg morphine produced the most substantial inhibitory effect (manifesting as a 78% reduction in the fecal output), we chose this concentration for all subsequent experiments.

Intestinal motility, defined as the motor patterns of the digestive tract involving coordinated contractions and relaxations of the smooth muscle layers, has been directly linked to constipation[16]. To assess small intestinal transit, we employed an activated carbon gavage test[17], which revealed a significant, dose-dependent reduction in the small intestinal transit rate in morphine-treated mice compared to saline-treated controls (Fig. 1e), indicating that morphine suppressed the small intestinal motility to induce constipation. To examine this inhibitory effect in vivo, we employed real-time patch recordings to monitor small intestinal peristaltic waves by attaching a sensor on the small intestinal wall (Fig. 1f). Morphine administration significantly reduced the frequency but increased the amplitude of small intestinal motility compared to saline-treated controls (Fig. 1g, h).

Given that the colon functions in water and electrolyte absorption to convert waste into solid feces[18], we assessed whether morphine also induces colonic peristalsis. Real-time patch recordings showed that

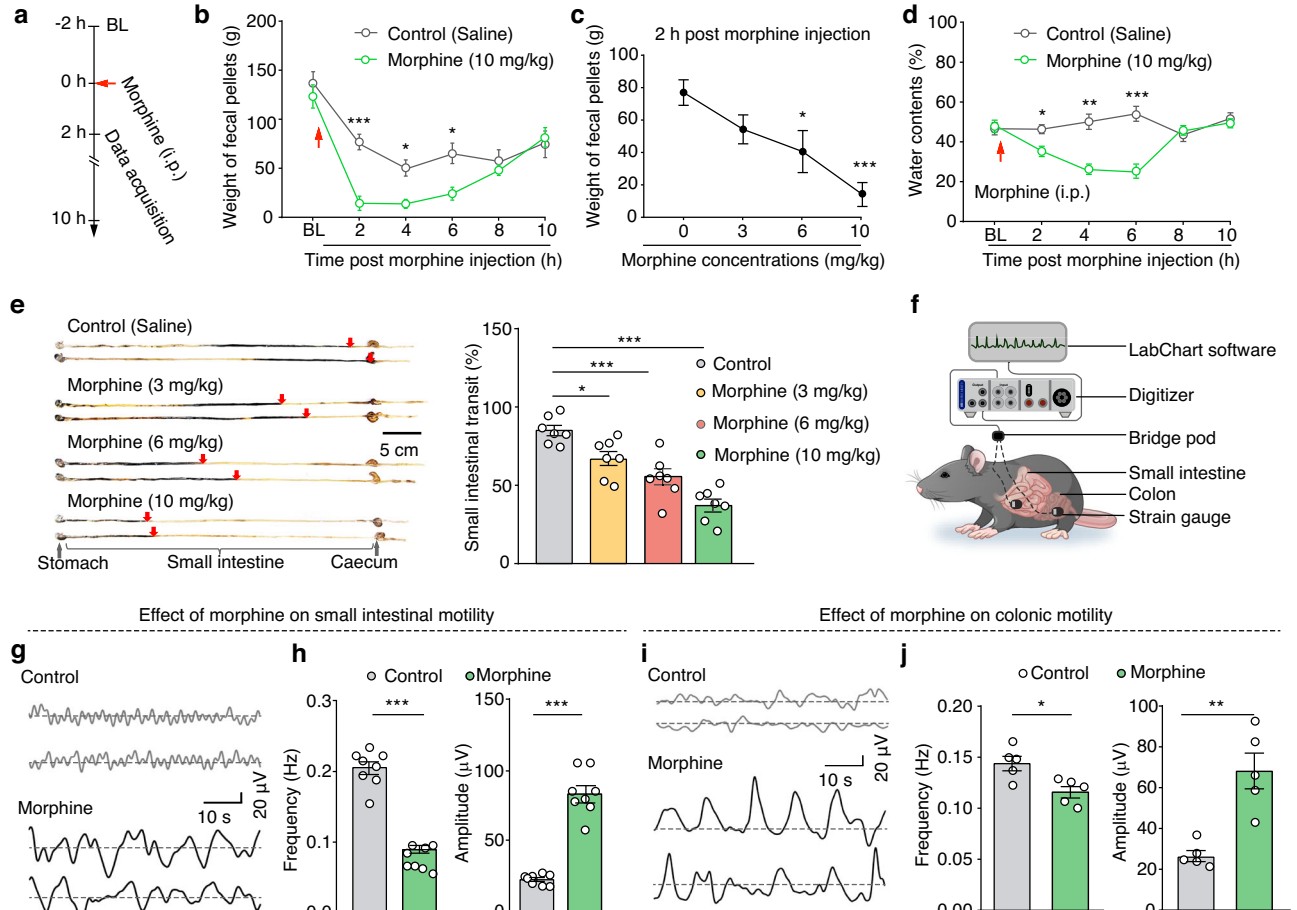

**Fig. 1 | Morphine induces constipation in mice. a** Schematic of the procedure for establishing a morphine-induced constipation mouse model. BL: baseline. **b,c** Summary data for fecal weight (**b**) and a dose-dependent reduction in fecal weight (**c**) in mice following intraperitoneal injection of morphine ($n = 10$, $F_{1,18} = 28.75$, $P < 0.0001$ for 2 h, $P = 0.0392$ for 4 h, $P = 0.0193$ for 6 h, $P = 0.0002$ for 1c). **d** Summary data for the water content of feces following morphine injection ($n = 8$, $P = 0.0143$ for 2 h, $P = 0.0013$ for 4 h, $P = 0.0005$ for 6 h). **e** Representative images (left) and summary data (right) showing changes in the small intestinal transit rate at different morphine concentrations ($n = 7$, $F_{3,24} = 22.61$, 3 mg/kg: $P = 0.0173$, 6 mg/kg: $P = 0.0001$, 10 mg/kg: $P < 0.0001$). **f–h**, Schematic (**f**), sample

traces (**g**), and summary data (**h**) showing the frequency and amplitude values for small intestinal motility ($n = 8$, $P < 0.0001$ for frequency, $P < 0.0001$ for amplitude). **i, j** As indicted in panel (**g** and **h**), but for colonic motility ($n = 5$, $P = 0.0144$ for frequency, $P = 0.0019$ for amplitude). Significance was assessed by two-way repeated-measures ANOVA with post hoc comparison between groups (**b, d**), one-way ANOVA with post hoc Bonferroni's test between groups (**c, e**) and two-tailed unpaired Student's $t$ test (**h, j**). The data are presented as the mean ± SEMs. *$P < 0.05$, **$P < 0.01$, ***$P < 0.001$; n.s., not significant. See also Supplementary Table S1. Source data are provided as a Source Data file.

morphine-treated mice exhibited significantly reduced frequency, but increased amplitude, of colonic motility compared to saline-treated control mice (Fig. 1i, j), indicating that morphine also suppresses colonic motility.

Note that there were no differences in total food intake or water consumption between the morphine and control groups (Supplementary Fig. 1b, c), supporting that the observed morphine-induced constipation in mice does not result from altered food or water consumption. Taken together, these findings demonstrate that the 10 mg/kg morphine model recapitulates the known effects of opioids in humans, inducing constipation through inhibition of both intestinal transit and motility.

## DMV[Ach] neurons directly project to the small intestine and regulate motility

Previous studies have shown that the sympathetic system typically inhibits gastrointestinal motility by suppressing smooth muscle contraction and glandular secretion, while the parasympathetic system exerts excitatory regulatory effects[16]. In particular, the vagus nerve is well known for mediating brain-to-gut communication[13,16,19]. To identify potential connections between the small intestine and the brain, we conducted retrograde trans-monosynaptic tracing by injecting Fluoro-gold (FG, a fluorescent dye) into the small intestinal wall (Supplementary Fig. 2a, b). One week later, FG signals were evident observed on bilateral of the DMV (Supplementary Fig. 2c, d). Subsequent immunofluorescence staining revealed colocalization of FG signals with approximately 95% of DMV cells positive for choline acetyltransferase (ChAT), a marker of cholinergic neurons (Supplementary Fig. 2e, f). These results show a connection proceeding from the DMV[Ach] neurons to the small intestine (DMV[Ach]→small intestine).

We also used a virus-based approach to examine this neural connection between the small intestine and the brain, specifically by infusing a retrograde AAV2/Retro-hSyn-Cre virus into the small intestine and a Cre-dependent AAV-DIO-mCherry virus into the DMV of C57 mice (Fig. 2a). Three weeks after injection, mCherry⁺ neurons were observed in the DMV, which were co-localized specifically with an antibody marking cholinergic neurons (~78%) (Fig. 2b,c), confirming that DMV[Ach] neurons innervate the small intestine. To map the precise distribution of DMV[Ach] neuronal fibers in the small intestine, we bilaterally infused an anterograde AAV-DIO-mCherry-mCherry virus into the DMV of *ChAT-Cre* mice (Fig. 2d). After one month of viral expression, we utilized FDISCO tissue clearing to visualize the small intestine in 3D (Fig. 2e). We observed abundant mCherry⁺ DMV[Ach] neuronal fibers in the small intestinal muscle layer (Fig. 2f, g and Supplementary Movie S1).

Previous studies have demonstrated that the DMV provides motor output to regulate intestinal motility and transit through the efferent vagus nerve[20]. To test whether local optogenetic activation of DMV[Ach] neuronal fibers directly increases the small intestinal motility, we injected an AAV-DIO-ChR2-mCherry virus into the DMV of *ChAT-Cre* mice (Supplementary Fig. 2g). Three weeks later, mCherry⁺ DMV[Ach] neuronal fibers were observed in the small intestinal wall (Supplementary Fig. 2h). Electrophysiological recordings showed that photostimulation of ChR2-containing DMV[Ach] fibers around the intestinal wall reliably induced an increase in the nerve firing rate (Supplementary Fig. 2i, j). In addition, we infused AAV-DIO-ChR2-mCherry virus into the DMV and implanted flexible wireless optoelectronic implants in the small intestine of *ChAT-Cre* mice (Fig. 2h–j). Poststimulation of ChR2-containing DMV[Ach] fibers around the intestinal wall reliably induced a significant higher frequency and lower amplitude of small intestinal motility in ChR2-expressing mice than mCherry-expressing controls under morphine treatment (Fig. 2k, l). These findings suggest that DMV[Ach] neurons directly project to the small intestine and show that optogenetic activation of DMV[Ach] neuronal fibers alleviates morphine-induced suppression of small intestinal motility.

## Morphine inhibits DMV[Ach] neuronal activity to induce constipation

To elucidate whether morphine-induced DMV[Ach] neuronal hypoactivity contributes to constipation, we investigated the changes in neuronal activity within the DMV[Ach]→small intestine circuit by infusing FG into the small intestinal wall of C57 mice (Supplementary Fig. 3a). One week later, whole-cell patch-clamp recordings in brain slices showed that the small intestine-projecting DMV neuronal activity was significantly decreased in morphine-treated mice compared to saline-treated controls (Supplementary Fig. 3b–d), indicating morphine inhibits DMV[Ach] neuronal activity. To dynamically visualize the calcium activity of small intestine-projecting DMV[Ach] neurons in freely moving mice, we injected AAV-DIO-GCaMP6m virus into the DMV and AAV2/Retro-hSyn-Cre virus into the small intestine of C57 mice, accompanied with the mounting of a microendoscopic gradient index (GRIN) lens at the top of the DMV (Fig. 3a). Microendoscopic calcium imaging in vivo showed that the spontaneous calcium transient frequency of DMV[Ach] neurons was significantly decreased after morphine treatment (Fig. 3b–d and Supplementary Movie S2), while no significant changes were observed in the saline-injected vehicle controls (Supplementary Fig. 3e, f). These results reveal that morphine reduces small intestine-projecting DMV[Ach] neuronal activity.

Given the decrease in DMV[Ach] neuronal activity in morphine-treated mice, we employed the chemogenetic designer receptor exclusively activated by designer drugs (DREADDS) technique to selectively activate or inhibit small intestine-projecting DMV[Ach] neurons. We injected AAV2/Retro-hSyn-Cre virus into the small intestinal wall, as well as bilaterally infused AAV-DIO-hM3Dq-mCherry or AAV-DIO-mCherry control virus into the DMV of C57 mice (Fig. 3e and Supplementary Fig. 4a, b). Three weeks later, intraperitoneal injection of clozapine-N-oxide (CNO, a ligand of the chemogenetic designer receptor) at 1 h after morphine administration, which significantly increased both the output and water content of feces in hM3Dq-expressing mice compared to mCherry-expressing control mice (Fig. 3f and Supplementary Fig. 4c). CNO administration also increased the transit rate and motility of the small intestine (Fig. 3g, h and Supplementary Fig. 4d). Conversely, chemogenetic inhibition of these small intestine-projecting DMV[Ach] neurons induced constipation in naïve mice (Fig. 3i–l and Supplementary Fig. 4e–h). These results collectively demonstrate that a reduction in DMV[Ach] neuronal activity is required for the observed morphine-induced constipation.

MORs are well-established as the main regulators of gastrointestinal motility and secretion and have been shown to mediate opioid-induced constipation[10,21,22]. MOR is widely expressed in various brain regions, including the nucleus accumbens, ventral tegmental area, amygdala, cerebral cortex, and hippocampus[23,24]. An ultrastructural study reported the presence of MORs in axons and terminals of GABAergic neurons[25]. Viewed together, these findings suggest that MOR may be expressed in the DMV and may contribute to morphine-induced DMV[Ach] neuronal hypoactivity. Pursuing this, we performed immunofluorescence staining and detected MOR⁺ signals with a nerve fiber-like morphology in the DMV, which were not co-labeled with the neuronal nuclei (NeuN, a marker for neuron), glial fibrillary acidic protein (GFAP, a maker for astrocyte), ionized calcium-binding adapter molecule 1 (Iba1, a marker for microglia), or a ChAT-specific antibody (Supplementary Fig. 5a–d). Notably, these MOR⁺ fibers closely surrounded the DMV[Ach] neuronal somata (Supplementary Fig. 5d), suggesting these MOR⁺ fibers per se innervate DMV[Ach] neurons. Moreover, we found that MOR⁺ signals were co-labeled with VgluT2⁺ neuronal terminals (75.64%) in the DMV of *VgluT2-Ai* transgenic mice (Supplementary Fig. 5e, f). Since VgluT2 is expressed in synaptic vesicles of presynaptic membranes[26], these findings suggest that the MOR⁺ fibers in the DMV are derived from upstream excitatory neuronal projections, likely conveying signals from other brain regions to promote opioid-induced constipation.

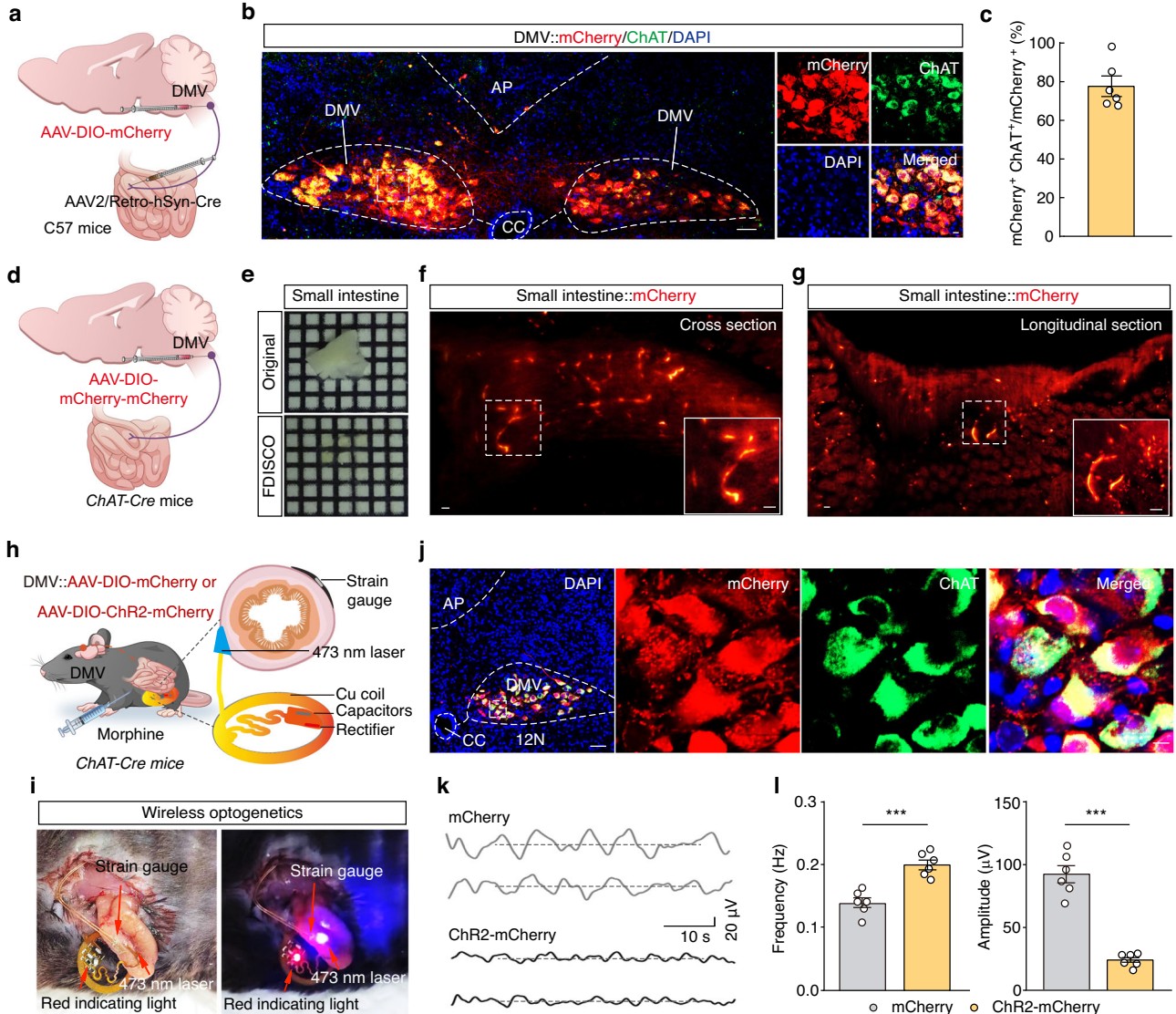

**Fig. 2 | DMV^Ach neurons directly project to the small intestine. a** Schematic diagram for AAV2/Retro-hSyn-Cre injection into the small intestine and AAV-DIO-mCherry injection into the DMV. **b,c** Representative images (**b**) and summary data (**c**) form mCherry⁺ DMV neurons colocalized with a ChAT-specific antibody. Scale bars, 100 μm (overview) and 10 μm (zoom). **d** Schematic diagram for AAV-DIO-mCherry-mCherry virus injection in the DMV of *ChAT-Cre* mice. **e–g** Representative images of small intestine tissue before and after (FDISCO), inducing translucency (**e**). The insets are magnified views from the white-boxed regions in the small intestine (**f,g**). **h** Schematic diagram for wireless optogenetic activation.

**i** Representative photographs showing the strain gauge and wireless optogenetic device under resting conditions (left) and during illumination (right).
**j** Representative image showing viral expression in the DMV. Scale bar, 50 μm and 5 μm. **k,l** Sample traces (**k**) and summary data (**l**) for small intestinal motility in the morphine-treated mice (n = 6, P = 0.0002 for frequency, P < 0.0001 for amplitude). Scale bars, 100 μm. Significance was assessed by a two-tailed unpaired Student's *t* test (**l**). The data are presented as the mean ± SEMs. *** P < 0.001; n.s., not significant. See also Supplementary Table S1. Source data are provided as a Source Data file.

## Defining a PVN^Glu→DMV^Ach→small intestine circuit

To identify the cellular and/or molecular mechanism(s) mediating the observed suppressive effect of morphine on small intestine-projecting DMV^Ach neuronal activity, we conducted retrograde monosynaptic tracing by infusing an AAV2/Retro-hSyn-Cre virus into the small intestine, along with a modified rabies virus (EnvA-pseudotyped RV-ΔG-EGFP) and Cre-dependent helper viruses (AAV-Ef1α-DIO-TVA-mCherry and AAV-Ef1α-DIO-RVG) into the DMV of C57 mice (Fig. 4a and Supplementary Fig. 6a). Three weeks later, we observed that EGFP⁺ signals were distributed in multiple regions, including the PVN and other regions (Fig. 4b and Supplementary Fig. 6b). The PVN had the highest number of EGFP⁺ cells among the examined brain regions (Supplementary Fig. 6c). Considering extensive previous studies reporting that PVN neuronal activity is closely associated with gastrointestinal function[27,28], we performed immunofluorescence staining

to identify the neuronal types in the PVN marked by EGFP: ~ 92% of EGFP⁺ cells were co-stained with a glutamate-specific antibody, while only 12% of EGFP⁺ cells were co-localized with a GABA-specific antibody (Fig. 4c, d). We therefore focused on the PVN^Glu → DMV→small intestine circuit in subsequent experiments.

To further characterize the anatomical connectivity of this circuit, we injected an anterograde AAV2/1-Cre-mCherry virus into the PVN and an AAV-DIO-mCherry virus into the DMV of C57 mice. Two weeks later, FG was injected into the small intestine (Fig. 4e). We subsequently observed FG and mCherry-expressing neurons in the DMV, which were co-localized with a ChAT-specific antibody (Fig. 4f–h), confirming that PVN^Glu neurons directly innervate DMV^Ach neurons that project to the small intestine.

To assess whether small intestine-projecting DMV neurons were functionally innervated by PVN^Glu neurons, we injected the AAV-DIO-

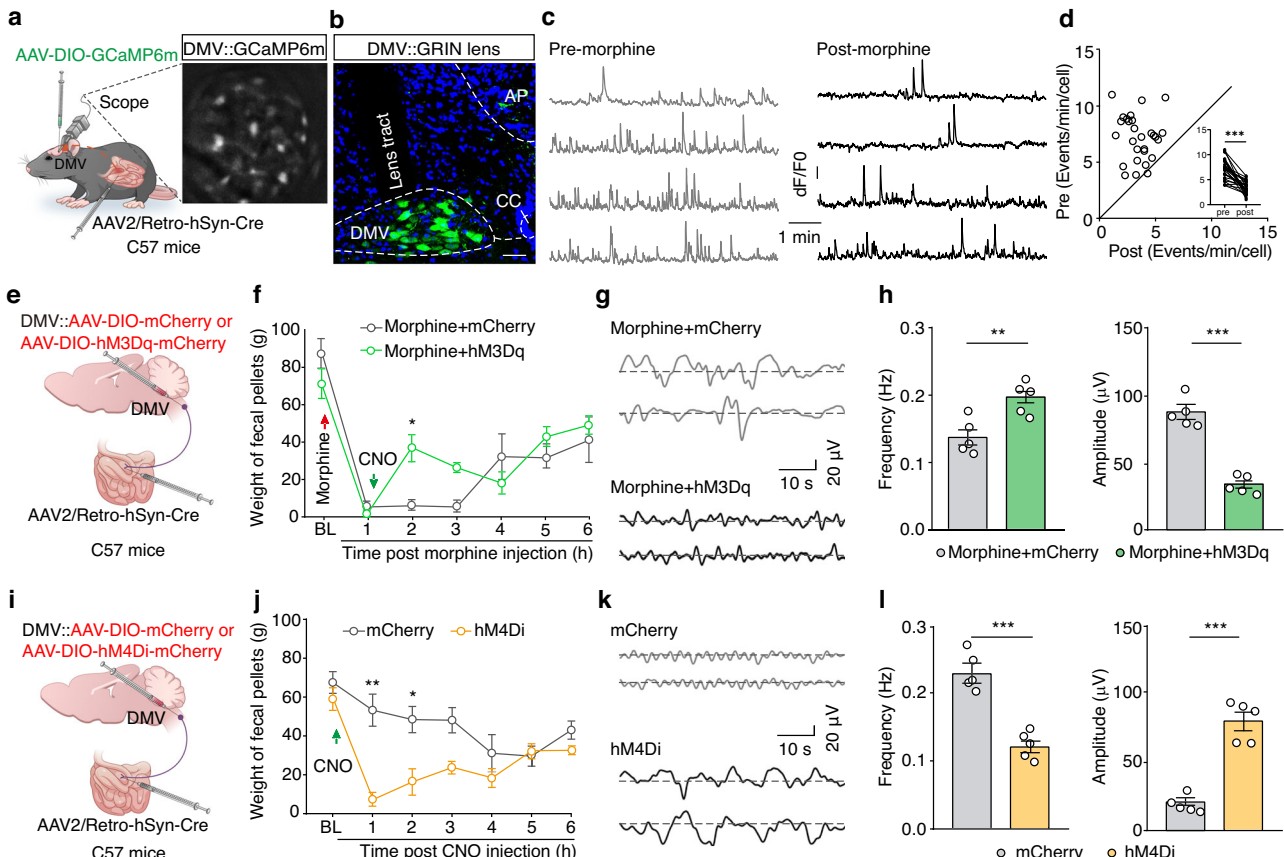

**Fig. 3 | Morphine inhibits DMV$^{Ach}$ neuronal activity to induce constipation.** **a** Schematic of microendoscopic calcium imaging in the DMV of freely moving mice. **b** Representative images of viral expression and the GRIN lens tract in the DMV. Scale bar, 50 μm. **c**, **d** Sample traces (**c**) and summary data (**d**) for calcium transients of GCaMP6m-expressing DMV neurons before and after morphine injection ($n = 28$ cells per group, $P < 0.0001$). **e** Schematic diagram for AAV-DIO-hM3Dq-mCherry virus injection into the DMV and Retro-AAV-hSyn-Cre virus injection into the small intestine of C57 mice. **f** Summary data for weight of feces in the morphine-treated mice from mCherry and hM3Dq groups ($n = 7$, $F_{1,12} = 0.5770$, $P = 0.0341$ for 2 h). **g**, **h** Sample traces (**g**) and summary data (**h**) for small intestinal motility in the morphine-treated mice from mCherry and hM3Dq groups ($n = 5$,

$P = 0.0067$ for frequency, $P = 0.0005$ for amplitude). **i** As indicted in panel (**e**), but for chemogenetic inhibition. **j** Summary data for weight of feces in the mCherry-expressing and hM4Di-expressing mice ($n = 7$, $F_{1,12} = 37.11$, $P = 0.0058$ for 1 h, $P = 0.0427$ for 2 h). **k**, **l** Sample traces (**k**) and summary data (**l**) for small intestinal motility in the mCherry-expressing and hM4Di-expressing mice ($n = 5$, $P = 0.0002$ for frequency, $P < 0.0001$ for amplitude). Significance was assessed by two-tailed paired Student's $t$ test (**d**), two-way repeated-measures ANOVA with post hoc comparison between groups (**f**, **j**), and two-tailed unpaired Student's $t$ test (**h**, **l**). The data are presented as the mean ± SEMs. *$P < 0.05$, **$P < 0.01$, ***$P < 0.001$; n.s., not significant. See also Supplementary Table S1. Source data are provided as a Source Data file.

ChR2-mCherry virus into the PVN and FG into the small intestine of *VgluT2-Cre* mice (Fig. 4i). Under whole-cell voltage clamp at −70 mV, photostimulation of ChR2-containing PVN$^{Glu}$ terminals evoked reliable excitatory postsynaptic currents (EPSCs) from the FG$^+$ DMV neurons (Fig. 4j–l). These EPSCs were blocked by the AMPA receptor antagonist 6,7-dinitroquinoxaline-2,3-dione (DNQX) (Fig. 4l, m). Taken together, these findings establish the presence of a PVN$^{Glu}$→DMV$^{Ach}$→small intestine circuit.

## Morphine reduces PVN$^{Glu}$ neuronal activity via MOR to induce constipation

MOR activation has been shown to modulate neuronal function through well-established G protein signaling mechanisms, such as postsynaptic activation of G protein-coupled inwardly rectifying potassium channels, which causes hyperpolarization and inhibition of neurons[29–31]. To examine whether morphine induces constipation in mice by inhibiting PVN$^{Glu}$ neuronal activity via MOR, we performed immunofluorescence staining in RV-tracing brain slices of normal mice revealed that EGFP$^+$/MOR$^+$ co-labelled cells across several regions, including the PVN, parabrachial pigmented nucleus (PBP), inferior colliculus (IC), primary somatosensory cortex (S1), periaqueductal

gray (PAG), locus coeruleus (LC), mesencephalic reticular formation, raphe magnus nucleus, retrorubral field, parasubthalamic nucleus, and dorsal raphe nucleus (Fig. 5a and Supplementary Fig. 6d). Among these brain regions, the PVN exhibited the highest number of EGFP$^+$/MOR$^+$ cells (Fig. 5b and Supplementary Fig. 6e), supporting MOR expressed on DMV-projecting PVN$^{Glu}$ neurons as primarily responsible for morphine-induced constipation. Subsequently, whole-cell patch-clamp recordings showed a significant decrease in the firing rate of PVN$^{Glu}$ neurons in morphine-treated mice compared to saline-treated controls (Supplementary Fig. 7a–e), confirming that morphine inhibits PVN$^{Glu}$ neuronal activity.

In addition, we conducted microendoscopic calcium imaging to selectively monitor calcium activity in DMV-projecting PVN neurons by infusing AAV2/Retro-hSyn-Cre virus into the DMV and AAV-DIO-GCaMP6m virus into the PVN of C57 mice, accompanied with the mounting of a GRIN lens at the top of the PVN (Fig. 5c, d). We observed that the calcium transient frequency of DMV-projecting PVN neurons was significantly decreased following morphine administration (Fig. 5e, f and Supplementary Movie S3), while no significant changes were detected upon saline injection (Supplementary Fig. 7f, g). These results indicate that morphine inhibits DMV-projecting PVN$^{Glu}$ neuronal activity.

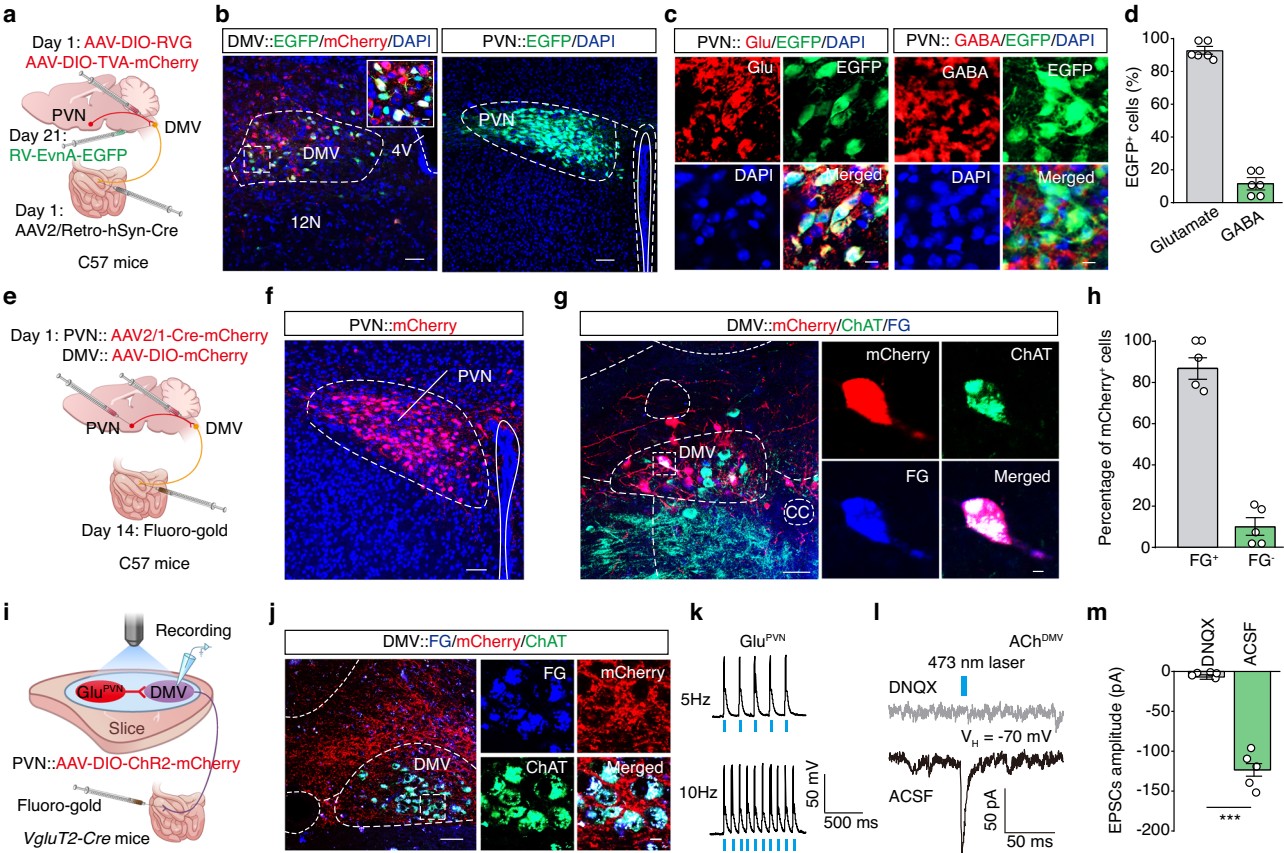

**Fig. 4 | Defining a PVN^Glu→DMV^Ach→small intestine circuit. a** Schematic diagram for the Cre-dependent retrograde trans-monosynaptic rabies virus (RV) tracing strategy. **b** Representative images of viral expression in the DMV (left) and EGFP-labeled neurons in the PVN (right). Scale bar, 50 μm. The inset depicts the area shown in the white box. Scale bar, 10 μm. **c,d** Representative images (**c**) and summary data (**d**) for the EGFP⁺ DMV-projecting PVN neurons that co-localized with Glutamate- and GABA-specific antibodies. Scale bar, 10 μm. **e,f** Schematic (**e**) and representative images (**f**) showing viral expression in the PVN. Scale bar, 50 μm. **g,h** Representative images (**g**) and summary data (**h**) for PVN-innervated mCherry⁺ DMV neurons that co-localized with FG signals and ChAT-specific antibody. Scale bar, 50 μm. The inset depicts the area shown in the white box. Scale bar, 5 μm.

**i** Schematic for FG infection and the whole-cell recording configuration in acute slices. **j** Representative images showing mCherry⁺ fibers surrounding the FG-labelled DMV^Ach neurons. Scale bars, 50 μm (overview) and 10 μm (zoom). **k** Representative traces of action potentials evoked by a 473 nm laser, recorded from mCherry⁺ PVN^Glu neurons. **l,m** Representative traces (**l**) and summary data (**m**) for light-evoked currents recorded from the FG⁺ DMV neurons before and after treatment with the DNQX agent ($n = 5$, $P < 0.0001$). Significance was assessed by a two-tailed unpaired Student's $t$ test (**m**). The data are presented as the mean ± SEMs. ***$P < 0.001$. See also Supplementary Table S1. Source data are provided as a Source data file.

To assess the potential role of PVN^Glu neurons in modulating morphine-induced constipation, we directly infused morphine into the bilateral PVN through implanted cannulas (Fig. 5g). Morphine injection significantly induced constipated symptoms in mice, including decreased the output and water content of feces, as well as a reduction in both the transit rate and motility of small intestine (Fig. 5h–j and Supplementary Fig. 7h, i). In contrast, bilateral injection of an opioid receptor antagonist, naloxone (i.e., pharmacological inhibition of MOR in the PVN), significantly reversed the morphine-induced constipation in naïve mice, compared to the vehicle-injected control group (Fig. 5k–n and Supplementary Fig. 7j, k).

To investigate the functional relevance of MOR expressed in other regions projecting to DMV^Ach neurons, we bilaterally implanted cannulas into the S1, IC, PAG, PBP, and LC, respectively. After one week of recovery and acclimation, we injected morphine, and two hours later, performed local infusions of naloxone or ACSF. We found that blocking MOR in these five brain regions did not significantly alter fecal output or small intestinal motility compared to that of ACSF-infused vehicle controls (Supplementary Fig. 8). These results indicated that, unlike the PVN, MOR expression in these brain regions does not play a major role, if any, in morphine-induced constipation.

## Conditional MOR knockdown in PVN^Glu neurons alleviates morphine-induced constipation

The *Oprm1* gene, encoding MOR, is known to regulate analgesic response to pain and also control intestinal motility[32,33]. To further verify that MOR-expressing neurons in PVN mediate morphine-induced constipation, we constructed a Cre-dependent AAV virus expressing *Oprm1* short-hairpin RNAs (shMOR) to induce MOR knockdown specifically in PVN^Glu neurons. We injected the rAAV-CMV-DIO-(EGFP-U6)-shRNA (*Oprm1*)-WPRE-hGH pA (AAV-DIO-shMOR-EGFP) virus or AAV-DIO-EGFP vector control into the bilateral PVN of *VgluT2-Cre* mice (Fig. 6a). After three weeks of virus expression, immunofluorescence staining confirmed that EGFP⁺ signal largely co-localized with signal from glutamate-specific antibody in PVN, indicating specific targeting of PVN^Glu neurons (Fig. 6b). Western blot analysis showed that MOR protein levels were significantly reduced in PVN^Glu neurons of mice injected with AAV-DIO-shMOR-EGFP compared to AAV-DIO-EGFP-infected controls, validating the efficacy of MOR knockdown (Fig. 6c).

Next, we examined neuronal activity in PVN^Glu neurons of morphine-treated mice with MOR knockdown. As expected, whole-cell patch-clamp electrophysiological recordings showed that PVN^Glu neuronal activity was significantly higher in the AAV-DIO-shMOR-EGFP

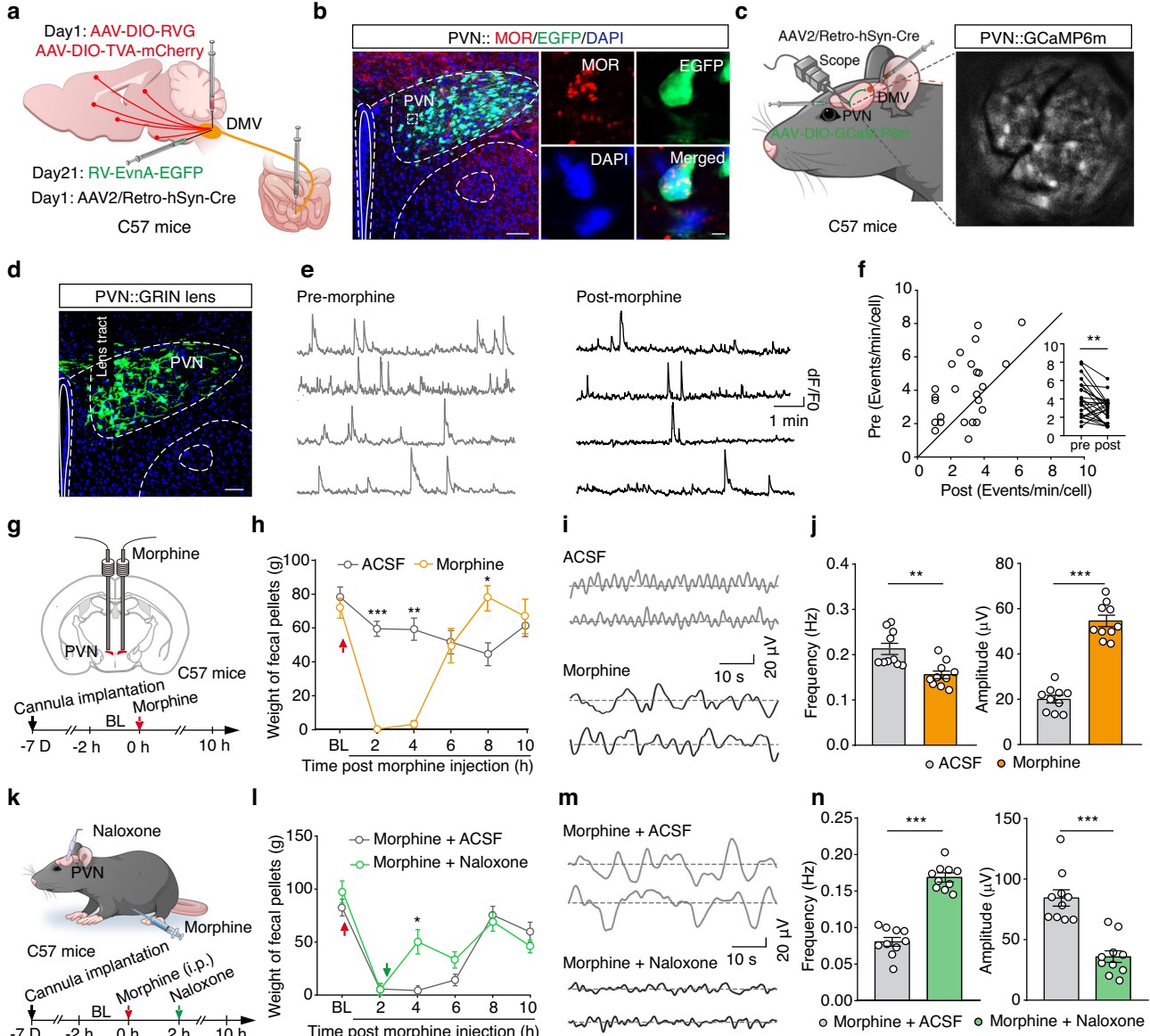

**Fig. 5 | Morphine reduces DMV-projecting PVN^Glu neuronal activity to induce constipation. a** Schematic diagram for the Cre-dependent retrograde trans-monosynaptic rabies virus (RV) tracing strategy. **b** Representative images for EGFP-labelled PVN neurons that co-localized with a MOR-specific antibody. Scale bars, 50 μm (overview) and 5 μm (zoom). **c** Schematic of microendoscopic calcium imaging in the PVN of freely moving mice. **d** Representative images showing a filed view from the GRIN lens (left) and the viral expression of GCaMP6m in the PVN (right). Scale bar, 50 μm. **e,f** Sample traces (**e**) and summary data (**f**) for calcium transients of GCaMP6m-expressing PVN neurons ($n = 25$ cells per group, $P = 0.0031$). **g** Schematic diagram for cannula implantation and morphine administration. **h** Summary data for the fecal weight in mice after administration of morphine into the PVN via cannulas ($n = 10$, $F_{1,18} = 16.89$, $P < 0.0001$ for 2 h, $P < 0.0001$ for 4 h). **i, j** Sample traces (**i**), summary data showing the frequency and

amplitude values for small intestinal motility (**j**) in mice after morphine administration into the PVN ($n = 10$, $P = 0.0021$ for frequency, $P < 0.0001$ for amplitude). **k** Schematic diagram for cannula implantation and naloxone administration. **l** Summary data for the fecal weight in mice after administration of naloxone into the PVN via cannulas ($n = 10$, $F_{1,18} = 5.612$, $P = 0.0166$ for 4 h). **m, n** Sample traces (**m**), summary data showing the frequency and amplitude values for small intestinal motility (**n**) in mice after naloxone administration into the PVN ($n = 10$, $P < 0.0001$ for frequency, $P < 0.0001$ for amplitude). Significance was assessed by two-tailed paired Student's $t$ test (**f**), two-way repeated-measures ANOVA with post hoc comparison between groups (**h**, **l**), Mann-Whitney U test and two-tailed unpaired Student's $t$ test (**j**, **n**). The data are presented as the mean ± SEMs. *$P < 0.05$, **$P < 0.01$, ***$P < 0.001$; n.s., not significant. See also Supplementary Table S1. Source data are provided as a Source Data file.

group than that in AAV-DIO-EGFP-expressing controls (Fig. 6d, e). Subsequent analysis of morphine-induced constipation in *VgluT2-Cre* mice injected with AAV-DIO-shMOR-EGFP revealed that PVN^Glu-specific MOR knockdown resulted in significantly higher fecal output and fecal water contents, as well as significantly higher transit rates and motility in small intestine of morphine-treated mice compared to that of corresponding AAV-DIO-EGFP control animals (Fig. 6f–h and Supplementary Fig. 9a–c). However, PVN^Glu-specific MOR knockdown did not significantly impact colonic dysfunction (Supplementary Fig. 9d–f),

suggesting that MOR-expressing PVN^Glu neurons may innervate specific DMV^Ach neuronal subpopulations to participate in morphine-induced dysfunction of the small intestine. These collective findings provide direct evidence that MOR on PVN^Glu neurons is involved in regulating morphine-induced constipation.

Previous studies have reported that MOR on intestinal neurons contributes to opioid-induced constipation[8–10]. The naloxone derivative, naloxegol, contains a polyethylene glycol chain that limits its permeability across the blood-brain barrier[22]. To assess whether

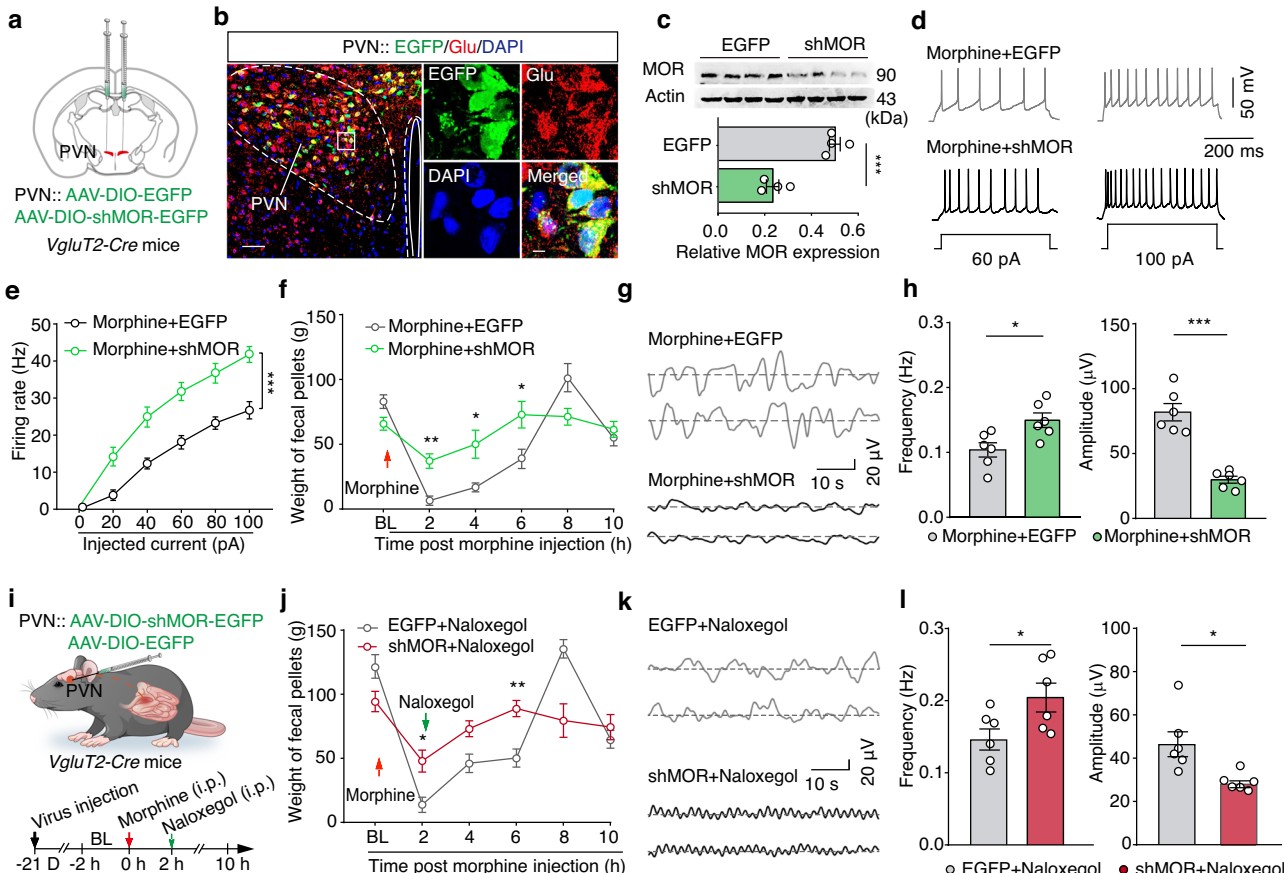

**Fig. 6 | Conditional MOR knockdown in PVN^Glu neurons alleviates morphine-induced constipation. a** Schematic illustrating the strategy for MOR knockdown in the PVN. **b,c** Representative images showing viral expression (**b**) and western blot analysis (**c**) of MOR protein levels in the PVN ($n = 4$ mice per group; $P = 0.0003$). The samples derive from the same experiment, and the gels were processed in parallel. Scale bars, 50 μm and 5 μm (zoom). **d,e** Representative traces (**d**) and summary data (**e**) for evoked action potentials recorded from PVN^Glu neurons with conditional MOR knockdown ($n = 11$ cells for Morphine + EGFP, $n = 10$ cells for Morphine + shMOR, $F_{1,19} = 23.10$, $P = 0.0001$). **f** Summary data for fecal weight in EGFP-expressing control mice and shMOR-expressing mice following morphine administration ($n = 10$, $F_{1,18} = 4.748$, $P = 0.0059$ for 2 h, $P = 0.0414$ for 4 h, $P = 0.0381$ for 6 h). **g,h** Representative traces (**g**) and summary data (**h**) of showing the frequency and amplitude values for small intestinal motility in the EGFP-expressing controls

and shMOR-expressing mice after morphine injection ($n = 6$, $P = 0.0189$ for Frequency; $P < 0.0001$ for Amplitude). **i** Schematic diagram for assessing the effect of PVN^Glu-specific MOR knockdown on constipation in naloxegol-treated mice. **j** Summary data for fecal weight in EGFP-expressing controls and shMOR-expressing mice after morphine and naloxegol administration ($n = 8$, $F_{1,14} = 3.035$, $P = 0.0390$ for 2 h, $P = 0.0071$ for 6 h). **k,l**, Sample traces (**k**) and summary data (**l**) showing the frequency and amplitude values for small intestinal motility in EGFP-expressing controls and shMOR-expressing mice following morphine and naloxegol injection ($n = 6$; Frequency: $P = 0.0421$; Amplitude: $P = 0.0115$). Significance was assessed by two-tailed unpaired Student's $t$ test (**c**, **h** and **l**), two-way repeated-measures ANOVA with post hoc comparison between groups (**e**, **f** and **j**). The data are presented as the mean ± SEMs. *$P < 0.05$, **$P < 0.01$, ***$P < 0.001$. See also Supplementary Table S1. Source data are provided as a Source Data file.

enteric MORs also contribute to morphine-induced constipation in mice, we administered different naloxegol doses (10 mg/kg, 30 mg/kg, and 100 mg/kg, i.p.) at 2 h post morphine treatment (Supplementary Fig. 10a). The results showed that naloxegol-treated mice had significantly increased fecal output and small intestinal transit rates compared to saline-treated vehicle controls (Supplementary Fig. 10b, c). As 30 mg/kg naloxegol produced the greatest relief of constipation (i.e., 58% higher fecal output; 70% faster transit rate), this concentration was applied in subsequent experiments (Supplementary Fig. 10d). In addition, fecal water content and small intestinal motility also significantly increased following naloxegol treatment compared to the vehicle control in constipated mice (Supplementary Fig. 10e–g). These results suggest that naloxegol could partially relieve morphine-induced constipation by rescuing intestinal transit and motility.

To further investigate the relative contributions of MOR-expressing enteric versus MOR-expressing PVN^Glu neurons in morphine-induced constipation, we infused AAV-DIO-shMOR-EGFP or AAV-DIO-EGFP control virus into the PVN of *VgluT2-Cre* mice, then induced constipation by i.p. morphine injection after three weeks of

virus expression (Fig. 6i). Naloxegol treatment at 2 h post morphine injection resulted in significantly increased fecal output and small intestinal motility in mice expressing AAV-DIO-shMOR-EGFP compared to that of AAV-DIO-EGFP-expressing control mice (Fig. 6j–l). Together, these results supported contributions of both enteric and PVN^Glu MOR-expressing neurons in morphine-induced constipation.

**Morphine suppresses tonic currents from the PVN to the DMV**

A previous study examining rats reported that opioid peptides inhibited excitatory but not inhibitory synaptic transmission in the DMV[34]. We explored whether morphine suppresses the synaptic transmission from PVN^Glu neurons to DMV^Ach neurons by infusing AAV-DIO-EGFP virus into the PVN and FG into the small intestine of *VgluT2-Cre* mice (Supplementary Fig. 11a, b). Three weeks later, we recorded tonic NMDA receptor-mediated currents with a selective NMDA receptor antagonist, D-AP5 (50 μM), to monitor any changes in the resting current of the PVN-innervated FG⁺ DMV neurons at +40 mV (Supplementary Fig. 11c). The tonic NMDA receptor-mediated currents were significantly decreased in morphine-treated mice compared to saline-

treated controls (Supplementary Fig. 11c, d), suggesting that morphine suppresses tonic NMDA receptor-mediated currents in small intestine-projecting DMV[Ach] neurons.

Next, we directly assessed whether the PVN[Glu] neurons contribute to these tonic currents in the DMV[Ach] neurons, specifically by ablating PVN[Glu] neurons based on infusion of a Cre-dependent genetically engineered caspase3 (AAV-DIO-taCaspase3) and AAV-DIO-EGFP viruses (or AAV-DIO-EGFP as a control) into the PVN, as well as FG injection into the small intestine of *VgluT2-Cre* mice (Supplementary Fig. 11e). Three weeks later, immunofluorescence staining showed that PVN[Glu] neurons were significantly ablated in taCaspase3-expressing mice compared to EGFP-expressing control mice (Supplementary Fig. 11f). Whole-cell voltage-clamp recordings revealed that the amplitude of NMDA receptor-mediated tonic currents in the FG⁺ DMV neurons were significantly decreased in the taCaspase3 group compared to control group (Supplementary Fig. 11g, h), confirming that the recorded tonic currents on DMV[Ach] neurons are derived from PVN[Glu] neurons. Together, these results reveal that morphine reduces tonic NMDA receptor-mediated currents of small intestine-projecting DMV[Ach] neurons from the PVN[Glu] neurons.

### Activation of the PVN[Glu]→DMV[Ach]→small intestine circuit reverses morphine-induced constipation

To assess whether artificial inhibition of the PVN[Glu] projections to the DMV mimics the effects of morphine-induced constipation, we employed again DREADD approach. To this end, we performed bilateral injection of an AAV-DIO-hM4Di-EGFP virus into the PVN and implanted cannulas to locally perfuse CNO into the DMV of *VgluT2-Cre* mice (Fig. 7a, b). Chemogenetic inhibition of DMV-projecting PVN[Glu] neurons resulted in significant reductions in both the output and water content of feces (Fig. 7c, d), accompanied by significantly decreased the transit rate and motility of the small intestine in hM4Di-expressing mice compared to EGFP-expressing control mice (Fig. 7e–g). Conversely, optogenetic activation of DMV-projecting PVN[Glu] neurons led to significantly increased fecal output and fecal water content, as well as faster motility and small intestinal transit rates in ChR2-expressing mice compared to mCherry-expressing controls (Fig. 7h–n). Similarly, chemogenetic activation of this circuit reversed morphine-induced constipation in mice, including significantly increased the output and water content of feces, as well as a significant higher transit rate and motility of the small intestine in hM3Dq-expressing mice than mCherry-expressing controls (Supplementary Fig. 12).

Together, these observations support a model wherein the PVN[Glu]→DMV[Ach]→small intestine circuit controls morphine-induced constipation by influencing the small intestinal motility (Fig. 8).

## Discussion

Extensive research on opioid-induced constipation has focused on the peripheral enteric nervous system[8–10], as its functional state has been shown to directly affect the severity of constipation by regulating gut motility, absorption, and digestion[35]. PAMORAs, such as methylnaltrexone, naldemedine, and naloxegol[36–38], can effectively alleviate opioid-induced constipation by selectively blocking MORs in the enteric nervous system without affecting central opioid analgesia[10,21]. However, only ~ 50% patients with opioid-induced constipation exhibit satisfactory response to PAMORA-based clinical management strategies targeting MORs in the enteric nervous system, and we observed that naloxegol could only partially alleviate morphine-induced constipation. This partial efficacy implies that non-peripheral mechanisms likely contribute to the pathophysiology of opioid-induced constipation.

In this study, we identify a brain-to-gut circuit—PVN[Glu]→DMV[Ach]→small intestine—as essential for opioid-induced constipation. Specifically, in the morphine-induced constipation model mice, we demonstrate that morphine inhibits PVN[Glu] neuronal activity through MOR, thereby reducing DMV[Ach] neuronal activity, which leads to decreased small intestinal motility and a reduced transit rate, ultimately manifesting in constipation. These findings support the notion that, similar to the development of peripheral opioid receptor antagonists, developing therapeutic targeting specific brain regions is one of several avenues that warrant exploration for patients unresponsive to current therapies.

Notably, MOR is widely distributed throughout the brain, and the DMV receives a broad array of projections from various nuclei[39,40]. Our findings add to the growing body of research on the role of the brainstem in opioid-induced constipation by revealing a PVN[Glu]→DMV[Ach] circuit that directly modulates small intestinal function in response to morphine. Although the large intestine has received a majority of attention in the opioid-induced constipation research field, with studies showing MOR activation in the large intestine during opioid-induced constipation[7], our study implicates the small intestine is also a critical site for morphine-induced constipation. These results suggest that targeting MOR in the brain could offer a more effective approach to manage opioid-induced constipation. Moreover, our study provides a mechanistic framework that could potentially explain other opioid-related side effects beyond constipation that may arise through modulation of central autonomic pathways (e.g., nausea, urinary retention, pruritus, and respiratory suppression)[41–43].

The DMV serves as a center for vagal regulation and brain-gut communication[44,45]. Most of the previous studies examining projections from DMV[ACh] to the gastrointestinal tract have focused on the stomach, whereas there is relatively limited direct evidence of these projections innervating the small intestine. For example, Tao et al. focused on *Cck*⁺ and *Pdyn*⁺ DMV neurons projecting specifically to the glandular stomach, where they mediate opposing functions via acetylcholine versus nitric oxide, respectively[46]; Wang et al. reported a PVN[CRH]→DMV[Ach] circuit that regulates gastric dilation via inhibitory postsynaptic currents[47], while Carson et al. showed that high-fat diet exposure induces tonic PVN→DMV activity and disrupts brain-stomach stress responses[48]. Similarly, Travagli's review emphasized the vago-vagal control of gastric motility and its modulation by descending central nervous system inputs, as well as hormonal and afferent feedback from the digestive process that can potently regulate sensitivity of the vago-vagal reflex[49]. Another recent study from our research team, also highlighting DMV involvement in stomach dysfunction under chronic stress exposure (*e.g.*, DRN[5-HT] → DMV[ACh] → stomach axis), did not address innervation of the small intestine[50]. However, none of these studies addressed DMV projections to the small intestine. In contrast, through advanced viral tracing and tissue clearing techniques, our study provides direct anatomical and functional evidence of DMV[Ach] projections to the small intestine. Moreover, we demonstrate that this circuit contributes to morphine-suppressed small intestinal motility—a functional link not previously demonstrated in this context.

We also found that morphine reduces PVN[Glu] neuronal activity and diminishes tonic currents in small intestine-projecting DMV[Ach] neurons, which support that the functional activity of the PVN is critical for the maintenance of normal gastrointestinal physiology[47,48]. Given that the PVN is a crucial brain region for sensing stress[51,52], these findings could also explain that acute stress and chronic negative mood disorders cause disturbances in intestinal functions[53], such as diarrhea and constipation[54,55].

Previous studies have shown that opioids inhibit colonic motility primarily via peripheral MOR activation[33,56] and that both the small intestine and the colon receive vagal innervation from the DMV[57,58]. Consistent with this, we observed morphine suppresses colonic motility. However, MOR knockdown in PVN[Glu] neurons does not affect morphine-induced colonic hypomotility, instead increased fecal water content and fecal size. This suggests that MORs on PVN[Glu] neurons may influence colonic water absorption, but not appear to directly regulate

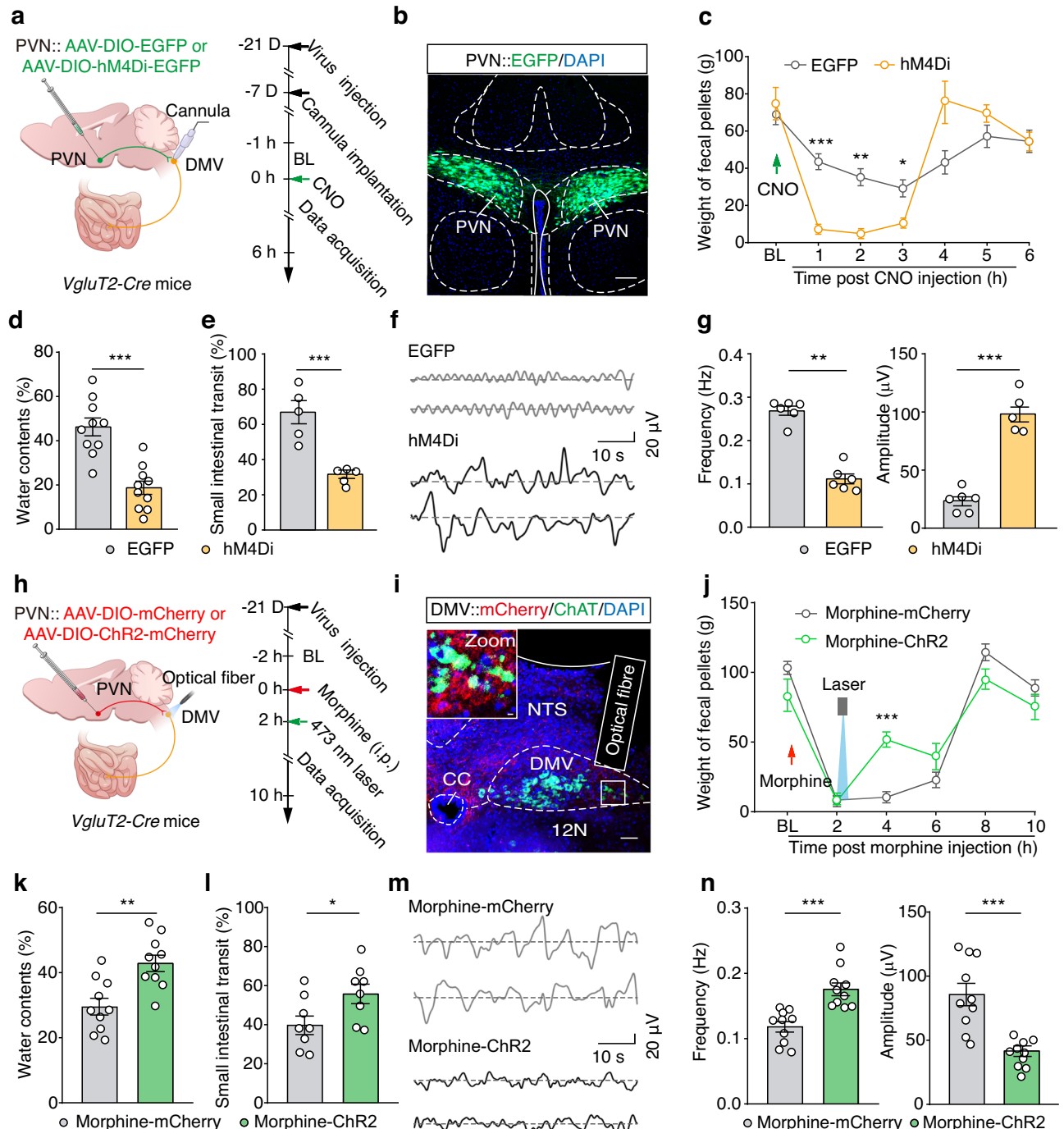

**Fig. 7 | Activation of DMV-projecting PVN$^{Glu}$ neurons reverses morphine-induced constipation. a** Schematic diagram for chemogenetic inhibition of DMV-projecting PVN$^{Glu}$ neurons and cannula implantation in the DMV of *VgluT2-Cre* mice. **b** Representative images for EGFP expression in the PVN. Scale bar, 100 μm. **c–e** Summary data for fecal weight (**c**), fecal water content (**d**), and small intestinal transit rate (**e**) in the EGFP-expressing controls and hM4Di-expressing mice ($n = 10$, $F_{1,18} = 1.999$, $P < 0.0001$ for 1 h, $P = 0.0004$ for 2 h, $P = 0.0291$ for 3 h; water contents: $n = 10$, $P < 0.0001$; small intestinal transit: $n = 5$, $P = 0.0009$). **f,g** Sample traces (**f**) and summary data (**g**) showing the frequency and amplitude values for small intestinal motility in the EGFP-expressing controls and hM4Di-expressing mice ($n = 6$, $P = 0.0022$ for frequency, $P < 0.0001$ for amplitude). **h,i** Schematic (**h**) for AAV-DIO-ChR2-mCherry virus injection in the PVN of *VgluT2-Cre* mice and representative images (**i**) showing the site of the optical fiber and mCherry$^+$ fibers in the DMV. Scale bars, 50 μm and 10 μm (zoom). **j–l** As indicated in panels (**c–e**) but for mCherry-expressing controls and ChR2-expressing mice after morphine injection and 473 nm blue light illumination ($n = 10$, $F_{1,18} = 0.1220$, $P < 0.0001$ for 4 h; water contents: $n = 10$, $P = 0.0017$; small intestinal transit: $n = 8$, $P = 0.0360$). **m,n** As indicated in panels (**f,g**), but for mCherry-expressing controls and ChR2-expressing mice after morphine injection and 473 nm blue light illumination ($n = 10$, $P = 0.0003$ for frequency, $P < 0.0001$ for amplitude). Significance was assessed by two-way repeated-measures ANOVA with post hoc comparison between groups (**c,j**), two-tailed unpaired Student's *t* test (**d, e, g** for Amplitude and **k–n**) and Mann-Whitney U test (**g** for Frequency). The data are presented as the mean ± SEMs. *$P < 0.05$, **$P < 0.01$, ***$P < 0.001$; n.s., not significant. See also Supplementary Table S1. Source data are provided as a Source Data file.

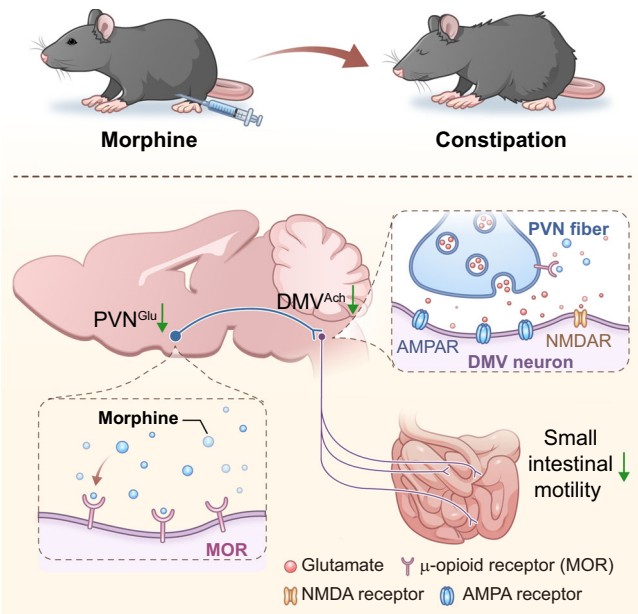

**Fig. 8 | A brain-to-small intestine circuit mediates morphine-induced constipation.** We found that a PVN^Glu→DMV^Ach→small intestine circuit mediates morphine-induced constipation in mice. Specifically, morphine inhibits PVN^Glu neuronal activity through MOR, which in turn leads to suppression of its tonic NMDA receptor-mediated currents to DMV^Ach neurons, and ultimately, the overall hypoactivity of this circuit mediates small intestinal dysfunction. Activation of this circuit (as well as pharmacologic inhibition or knockdown of MOR in PVN^Glu neurons) relieves symptoms of morphine-induced constipation.

colonic motility. This is plausible given that the vagus nerve innervates the small intestine with higher density fibers compared to the colon[33,57]. Moreover, other recent work indicates that the nigro-vagal pathway can modulate tone and motility of the proximal colon via D1-like receptors in the DMV, implying that the colon and small intestine, although both innervated by DMV, are regulated by distinct upstream inputs and receptor pathways[59]. However, whether peripheral mechanisms, together with or independent of other central pathways, also participate in morphine-induced colonic hypomotility requires further investigation in future studies.

Fecal output is influenced by multiple factors, including food intake, gut motility, microbiota composition, and mucus secretion[60,61]. This mucus is secreted by goblet cells and typically contains several major components. In the small intestine, the mucus layer limits the number of bacteria that can reach the epithelium and Peyer's patches. By contrast, in the large intestine, the inner mucus layer separates commensal bacteria from the host epithelium, and the outer colonic mucus layer provides a habitable niche for commensal bacteria[62,63]. Defects in the mucus layer increase susceptibility to pathogens and have been linked to the development of inflammatory bowel diseases, especially those related to the abovementioned dysfunction in limiting overall pathogenic and commensal microbe abundance on the gut mucosal surface[64,65]. Although modulating DMV^Ach neurons does not directly promote mucus secretion by small intestinal goblet cells, promoting secretion via Brunner's glands can lead to conditions conducive for the proliferation of lactobacilli[66]. As numerous studies have identified a close relationship between gut microbiota and functional constipation[60,67], promoting duodenal mucus secretion may therefore indirectly affect morphine-induced constipation.

In summary, our findings offer insights into the neural mechanisms of opioid-induced constipation and help clarify the limited efficacy peripheral opioid receptor antagonists in treating opioid-induced constipation patients. Exploring alternative neuromodulatory

interventions targeting this circuit could provide therapeutic options for managing opioid-induced constipation.

## Methods

### Animals

In this study, *C57BL/6 J, ChAT-Cre* and *VgluT2-Cre* mouse lines at the age of 8–10 weeks old were used for experiments. All animals were purchased from Charles River or Jackson Laboratories and housed in groups of five per cage, except when a GRIN lens was implanted or a strain gauge was attached. They were maintained under a 12 h light-dark cycle (lights on from 8:00 AM to 6:00 PM) at a temperature conducive to mouse survival (23–25 °C) and 50% humidity, with *ad libitum* access to water and food. To generate litters, two females were mated with one male in our facility. The male was removed 10 days later, and the females were kept separately 1–4 days before delivery. All protocols involving mice were approved by the Animal Care and Use Committee of the University of Science and Technology of China (USTCACUC26080123082). Unless otherwise specified, all experiments were conducted using male mice.

### Morphine-induced constipation mouse model

We established a constipated mouse model by intraperitoneal injection of morphine in C57 mice. The specific methodology is as follows: Newly acquired mice were acclimatized in the animal facility for at least one week, then placed on a metal grid for 12 h each day in the experimental room (8:00 AM-8:00 PM), and the fecal output was recorded. Once daily fecal output stabilized (approximately one week later), mice were injected intraperitoneally (i.p.) with different doses of morphine (3 mg/kg, 6 mg/kg, and 10 mg/kg). Water and food were available *ad libitum* during the investigation. Mice did not show a decrease in fecal output following morphine injection was excluded from the analysis due to individual differences in morphine-induced constipation.

### Weight recording of fecal pellets

Mice were placed on a metal grid, and fecal pellets excreted over a 12 h period were collected on a glass plate placed below the grid and subsequently weighed. Specifically, after a 2 h baseline measurement, mice were injected i.p. with morphine or saline. The weights of fecal pellets were recorded every 2 hours for the next 10 h. Water and food were available *ad libitum* during the investigation.

### Fluid content of fecal pellets

During the investigation period, each fecal pellet was collected and weighed first as wet weight and then again as dry weight after being placed in a 37 °C oven overnight. The percentage of fecal fluid content was calculated using the following formula:

$$\text{Fluid content (\%)} = [\text{wet weight} - \text{dry weight}]/\text{wet weight} \times 100 \quad (1)$$

### Small intestinal transit recording

The charcoal meal test is a commonly used protocol for measuring small intestinal transit[17]. In this study, mice were fasted for 6 hours before the experiments. While food was restricted, animals had free access to water. Thirty minutes after morphine administration (10 mg/kg, i.p.), mice were intragastrically administered a 0.25 mL suspension of 10% vegetable charcoal Norit A®(Sigma-Aldrich) in 5% gum acacia (Sigma-Aldrich). Twenty minutes later, mice were perfused, and the small intestine were dissected to collect the intestinal segment between the stomach and the ileocecal junction. The distance from the pyloric sphincter to the ileocecal junction was used to represent the total length of the small intestine, while the distance from the pylorus to the front of the activated charcoal was measured as the migration distance. The small intestinal transit rate was calculated using the

following formula:

$$\text{The small intestinal transit rate (\%)} = [\text{MAC}/\text{LSI}] \times 100 \qquad (2)$$

Where MAC and LSI represent the migration distance of activated charcoal and the total length of the small intestine, respectively.

## Intestinal motility recording

After fasting overnight, mice were deeply anaesthetized using isoflurane. The surgical areas were shaved and sterilized. A ventral midline laparotomy was performed, and a strain gauge (120 Ω) was attached to the surface of the small intestinal wall. Another skin incision was made along the dorsal midline, and the wires from the transducer were exteriorized through the abdominal wall, running under the skin toward the back of the neck. Following surgery, mice were housed individually with access to standard food and water. The strain gauge signal was amplified by using a bridge pod (ML-301, AD Instruments) and a digitizer (PowerLab 26/04, AD Instruments). Each mouse was recorded for at least 30 minutes, with the average amplitude and frequency calculated automatically using LabChart 8 software (AD Instruments). Real-time patch recordings of colonic peristalsis were similar approach employed in the small intestine.

## Food and water intake measurement

For acclimatization, mice were housed individually in a cage with *ad libitum* access to water and food for at least one week. Following this period, Mice were injected i.p. with morphine or saline, and food and water intakes were measured before and after the injection.

## Immunohistochemistry

Mice were deeply anesthetized with isoflurane and then perfused with saline for 3 min and 4% ice-cold paraformaldehyde solution for 4 min. After perfusion, the mouse brain was taken out carefully, and immersed in 4% paraformaldehyde solution overnight, then followed with 20% and 30% sucrose solution at 4 °C for dehydration until isotonic. Brains were cut into 40 μm coronal slices at − 20 °C using a cryostat microtome system (CM1860, Leica). For immunofluorescence, the slices were washed 3 times with PBS. After being blocked in the buffer (containing 0.5% Triton X-100, 5% donkey serum in PBS) for 1 hr at room temperature, they were incubated in primary antibodies, including anti-glutamate (1:500, rabbit, Sigma), anti-GABA (1:500, rabbit, Sigma), anti-ChAT (1:500, goat, Millipore),anti-mu opioid receptor (1:500, rabbit, Abcam), and anti-TUBB3 (1:500, rabbit, Abcam) at 4 °C for 24 hr. Then, slices were washed 3 times with PBS, and incubated with the secondary antibodies (Key Resources Table) at room temperature in a dark place for 1.5 hr. Cell nuclei were stained with DAPI (1:2000, Cat. No. D9542, Sigma-Aldrich) for 5 min. The slices were scanned and imaged with a Zeiss LSM 980 microscope.

## Tissue clarity and morphological reconstruction

The sections of the small intestine were cleared using three-dimensional imaging of solvent-cleared organs with superior fluorescence-preserving capability (FDISCO) protocol. For tissue clearing, the sectioned small intestine was incubated in 4% PFA at 4 °C with shaking overnight and washed twice with 1 × PBS for ~ 2 h at room temperature. After clearing, the fixed small intestine samples were dehydrated with THF solutions (mixed with dH$_2$O, pH adjusted to 0.9 with triethylamine) at a series of concentrations of 50, 70, 80, and 100 volumes % (twice or thrice). Pure DBE (108014, Sigma−Aldrich) was used as a refractive index matching solution to clear the tissue after dehydration. All steps were performed at 6−8 °C with slight shaking. During clearing, the tissues were placed in glass chambers covered with aluminum foil in the dark. Then, three-dimensional fluorescence images of the cleared samples were obtained using a light sheet microscope (LiToneXL, Light Innovation Technology, China) equipped

with a 4 × objective lens (NA = 0.28), and thin light sheets were used to illuminate the four sides of the samples during imaging. To acquire images, the cleared samples were manually attached to the sample holder adapter. Subsequently, the samples were immersed in imaging reagent within a 3D printing sample chamber and excited with light sheets with a wavelength of 594 nm. The raw data (tiff. images) of the small intestine were stitched and converted using LitScan software, and all the data were visualized by IMARIS software (Bitplane).

## Virus injection and cannula implantation

Mice were deeply anesthetized with pentobarbital (20 mg/kg, i.p.) and were immobilized on a stereotaxic frame (RWD Life Science Inc., China). Then, sterile ointment was applied to each eye, and a miniature heating pad was placed under the mouse's body to maintain the temperature at 37 °C. After simple disinfection and a midline scalp incision, the skull surface was exposed with a midline scalp incision and its position was leveled. A syringe tip and adjustable speed dental drill (B67275, Meisinger, Germany) were used to open a small (~ 0.5 mm) craniotomy. Then, the virus was injected into the brain region specifically at a speed of 30 nL/min by using a 10 μL microsyringe (Gaoge) assembled with a calibrated glass microelectrode (1B 100-3, WPI, USA). Following the injection, the microsyringe was left in the injection site for 5–10 min to minimize virus leakage in the track. Finally, the incision was sutured, and the surgical wound was sterilized with sterile ointment. The injections were performed using the following stereotaxic coordinates: the DMV coordinates: anterior/posterior (AP), − 7.50 mm; medial/lateral (ML), ± 0.25 mm; dorsoventral (DV), − 3.60 mm; the PVN coordinates: AP, − 0.70 mm; ML, ± 0.25 mm; DV, − 4.95 mm.

For anterograde tracing of the PVN → DMV circuit, rAAV-hSyn-Cre-mCherry-WPRE-hGH polyA (AAV2/1-hSyn-Cre-mCherry, AAV2/1, 1.03 × 10$^{13}$ vg/mL) virus was injected into the PVN and Cre-dependent rAAV-EF1α-DIO-mCherry-WPRE-hGH pA (AAV-DIO-mCherry, AAV2/9, 5.31 × 10$^{12}$vg/mL) virus was injected into the DMV of C57 mice. After three weeks, brain slices were prepared for examining the mCherry$^+$ neurons innervating by PVN in the DMV. For anterograde tracing of the DMV→small intestine circuit, rAAV-Ef1α-DIO-hChR2(H134R)-mCherry-WPRE-pA (AAV-DIO-ChR2-mCherry, AAV2/9, 1.63 × 10$^{13}$ vg/mL, 200 nL) was injected into the DMV of *ChAT-Cre* mice. Three weeks later, the mCherry signals and the co-staining with TUBB3-specific antibody were investigated in the small intestinal wall.

For retrograde tracing from the small intestine to the DMV, FG (fluorochrome 4% in PBS, 10 μL) was injected into the small intestine, or scAAV2/Retro-hSyn-Cre-WPRE (20 μL) was injected into the small intestinal wall at different sites (200 nL per site) and AAV-DIO-mCherry was injected into the DMV of C57 mice; the pipette was left in place for an additional 2 min after the injection and then slowly withdrawn. For retrograde monosynaptic tracing of the PVN → DMV circuit, the helper viruses containing rAAV-EF1α-DIO-RVG-WPRE-hGH pA (AAV-DIO-RVG, AAV2/9, 4.59 × 10$^{12}$vg/mL) and rAAV-EF1α-DIO-H2B-mCherry-T2A-TVA-WPRE-hGH pA (AAV-DIO-TVA-mCherry, AAV2/9, 5.56 × 10$^{12}$ vg/mL) (1:2, 150 nL) were injected into the DMV, and scAAV2/Retro-hSyn-Cre-WPRE (20 μL) was injected into the small intestinal wall of C57 mice. Three weeks later, RV-EnvA-ΔG-EGFP (3.10 × 10$^8$ IFU/mL, 100 nL) was injected into the same site of the DMV using identical conditions and coordinates. Seven days after the last injection, EGFP signals were stained with a glutamate-specific or GABA-specific antibody in the PVN of the brain slices.

For conditional MOR knockdown in PVN$^{Glu}$ neurons, we injected the rAAV-CMV-DIO-(EGFP-U6)-shRNA (*Oprm1*)-WPRE-hGH pA (AAV-DIO-shMOR-EGFP) virus or rAAV-CMV-DIO-(EGFP-U6)-shRNA(scramble)-WPRE-hGH polyA (AAV-DIO-EGFP) vector control into the bilateral PVN of *VgluT2-Cre* mice. After three weeks of virus expression, the expression and functional impact of the MOR knockdown in PVN$^{Glu}$ neurons were assessed using immunofluorescence staining, western blotting, and ex vivo brain slice electrophysiological recordings.

Stainless steel guide cannulas were bilaterally implanted into the DMV, the PVN or the other five brain regions following stereotaxic coordinates: the S1 coordinates: AP, − 0.47 mm; ML, ±0.20 mm; DV, − 0.15 mm; the IC coordinates: AP, −1.07 mm; ML, ± 3.5 mm; DV, − 3.5 mm; the PAG coordinates: AP, − 4.50 mm; ML, ± 0.45 mm; DV, − 2.65 mm; the PBP coordinates: AP, −3.63 mm; ML, ± 0.5 mm; DV, − 4.25 mm; the LC coordinates: AP, − 5.41 mm; ML, ± 0.8 mm; DV, − 3.75 mm. The cannulas were anchored to the skull by dental acrylic. The animals were allowed a one-week postoperative recovery period prior to initiation of experiments. All drugs were dissolved in the ACSF and administered intra-DMV, intra-PVN, intra-S1, intra-IC, intra-PAG, intra-PBP, or intra-LC.

## Recording of nerve activity innervating the small intestinal wall

The AAV-DIO-ChR2-mCherry virus was injected into the DMV of *ChAT-Cre* mice. Three weeks later, the mice were deeply anaesthetized with isoflurane, and the nerve innervating the small intestinal wall was exposed. A platinum-customized tetrode was hooked to the nerve fibers, and the electrode was connected to a multichannel recording and signal processing system (Powerlab 26/04, AD Instruments). A 473 nm laser light (5 mW, 50 Hz) was delivered to the ChR2-containing fibers. For data analysis, all signals were digitally bandpass-filtered between 10 Hz and 60 Hz to reduce noise. Firing spikes were automatically collected using the LabChart 8 software's peak analysis function.

## In vivo optogenetic manipulations

To implant the wireless optogenetic device, the AAV-DIO-ChR2-mCherry virus was injected into the DMV of *ChAT-Cre* mice. After two weeks, mice were deeply anaesthetized with isoflurane, and the surgical areas were shaved and sterilized. A midline incision was made in the abdomen to expose the small intestine. The 400-μm-long needle (80–130 mm-thick) with a blue LED of the wireless optogenetic device (ThinkerTech) was glued to the small intestine with tissue glue. Strain gauges were placed on the same intestinal segment to monitor motility, as detailed in the "Intestinal motility recording" section. The LED light source and the strain gauges were positioned within the same segment and placed on the opposite side of the stimulation site, ensuring that the area monitored by the strain gauges was fully within the irradiation range of the LED light. A copper wire connected the needle to a 1 cm diameter copper coil, which was then buried between the skin and the peritoneum. After one week, these devices were wirelessly controlled using a radio-frequency power source to activate cholinergic terminals in the small intestine with a 473 nm laser (2–5 mW, 50 Hz).

For the optogenetic experiments targeting the PVN$^{Glu}$ → DMV pathway, we injected AAV-DIO-ChR2-mCherry or AAV-DIO-mCherry control virus into the PVN of *VgluT2-Cre* mice. At the same time, optical fibers were implanted above the DMV. To selectively activate the PVN → DMV circuit, 473 nm laser (2–5 mW, 20 Hz, 30 min pulses) was delivered through the implanted fibers.

## In vivo chemogenetic manipulations

For chemogenetic experiments in the DMV, rAAV-Ef1α-DIO-hM3D(Gq)-mCherry-WPRE-pA (AAV-DIO-hM3Dq-mCherry, AAV2/9, $5.00 \times 10^{12}$vg/mL) or rAAV-Ef1α-DIO-hM4D(Gi)-mCherry-WPREs (AAV-DIO-hM4Di-mCherry, AAV2/9, $2.25 \times 10^{12}$ vg/mL) was injected into the DMV of C57 mice. Simultaneously, retro-AAV-hSyn-CRE-WPRE-hGH (AAV2/Retro-hSyn-Cre, $5.40 \times 10^{12}$vg/mL) virus was injected into the small intestine wall. The rAAV-Ef1α-DIO-mCherry-WPRE-pA (AAV-DIO-mCherry, AAV2/9, $2.12 \times 10^{12}$ vg/mL) virus was used as the control. For chemogenetic activation, CNO (5 mg kg$^{-1}$) was administrated intraperitoneally 1 h after morphine injection. For chemogenetic inhibition, CNO (5 mg kg$^{-1}$) was given 1 hour after baseline investigation.

For the chemogenetic experiments in the PVN, AAV-DIO-hM4Di-EGFP or AAV-DIO-hM3Dq-mCherry was delivered into the PVN of *VgluT2-Cre* mice. AAV-DIO-EGFP/mCherry virus was used as the control. Simultaneously, cannulas (0.25 mm inner diameter, RWD) were implanted into the DMV, and CNO (3 μM) was administrated via the cannulas.

**In vitro electrophysiological recordings**. Mice were deeply anesthetized using isoflurane and then perfused with ice-cold oxygenated N-methyl-d-glucamine artificial cerebrospinal fluid (NMDG ACSF). Coronal brain slices (300 μm) were prepared using a vibrating microtome (VT1200s, Leica) and incubated in oxygenated HEPES ACSF solution (28 °C) for at least 1 hour. For whole-cell recordings, slices were transferred to a slice chamber (Warner Instruments) and continuously perfused with oxygenated standard ACSF solution at 32 °C. Neurons in the DMV and PVN regions were visualized with a water immersion objective (×40) on an upright microscope (BX51WI, Olympus). Signals were low-pass filtered at 2.8 kHz and digitized at 10 kHz via a Multiclamp 700B amplifier. Data were collected from the neurons with an input resistance above 100 MΩ and series resistance below 30 MΩ, using Clampex 10 (Molecular Devices). Current-evoked firing was recorded in current-clamp mode (I = 0 pA) with pipettes (5–7 MΩ) containing potassium gluconate-based internal solution. Optical stimulation (473 nm, 10 V, 20 ms) was delivered through an optical fiber (200 μm diameter, Inper) positioned 0.2 mm above the target region.

For tonic NMDAR-mediated current recording, neurons were held at + 40 mV in normal ACSF with internal solution containing (in mM): 110 Cs methylsulfate, 20 TEA-Cl, 15 CsCl, 4 ATP-Mg, 0.3 Na3-GTP, 0.5 EGTA, 10 HEPES, 4.0 QX-314, and 1.0 spermine, pH adjusted to 7.2 with CsOH, and osmolality set to 290–300 mOsm/kg with sucrose. Recordings were performed with 1 μM TTX, 100 μM picrotoxin, and 20 μM DNQX. After a stable baseline, tonic NMDAR currents were observed by bath application of 100 μM D-AP5[68].

## Microendoscopic imaging and data processing

To ensure the validity and reliability of our recordings, we engineered a customized approach for lens stabilization based on previous studies[40,50]. Specifically, before implanting the GRIN lens into the DMV, we used medical-grade adhesive to pre-attach a supportive scaffold composed of flexible tubing and tungsten wires to the lens. This design effectively minimized motion artifacts and ensured stable recording. For microendoscopic imaging, three weeks post-virus injection and the gradient index (GRIN) lens (Diameter: 0.5 mm; Length: 6.1 mm; Inscopix, USA) implantation, a miniature fluorescence microscope (Inscopix) was mounted onto the GRIN lens base plate. Final data were recorded for 30 min before and after saline or morphine injection.

For image processing, raw data underwent spatial down sampling (2 × binning) and we corrected motion artifacts using Inscopix Data Processing Software (v1.3.1) through the following sequential steps: (i) defective pixel correction and file size reduction during preprocessing; (ii) spatial frequency filtering; and (iii) motion correction via frame registration based on normalized cross-correlation. After these steps, (iv) signal was normalized as Δ*F/F* through reference frame subtraction/division, and (v) potential cells were initially identified using a hybrid of principal component analysis/independent component analysis (PCA/ICA) algorithms with region-of-interest (ROI) spatial footprints and stringent exclusion criteria, including anatomical abnormalities (atypical size/shape or lens boundary proximity), low signal fidelity (SNR < 3), or signal contamination (spatial overlap > 60% combined with Pearson's *r* > 0.6 across entire recording sessions). Notably, this PCA/ICA approach is also used to identify signals from independent components in video imaging data. These signals correspond to spatially filtered images, which define target cell locations,

and temporal traces, which define calcium fluorescent signals produced by target cells.

Every extracted cell was manually checked for circular spatial footprints, and $Ca^{2+}$ transients were characterized by sharp rises and slow decays[69]. While each component ideally represents an in-focus neuronal soma, the identification process also captures confounding components which can arise from out-of-focus somata, non-somatic structures (e.g., dendrites or blood vessels), or non-biological sources (e.g., fluctuations in background fluorescence), all of which are considered abnormal signals or artifacts. In addition, we examined those calcium traces that precisely aligned with the timing of transient somatic fluorescent signals in video imaging data. Those calcium traces that showed atypical rise/decay dynamics were also excluded as artifacts, as they commonly originate from the hemodynamic function of the mouse vasculature. By manually sorting these putative cells based on $\Delta F/F$ values and cellular morphology determined by source pixel locations, we could identify high-quality target cells[70]. Furthermore, we also manually checked $F$-values in the final dataset, which further allowed us to exclude time windows associated with unstable imaging (such as pronounced motion) from subsequent analysis. Following this protocol, only accepted traces were included in subsequent quantitative analyses. Longitudinal registration was processed with Inscopix software and a custom MATLAB script to track the same neurons across sessions as described[71,72] (https://github.com/mukamel-lab/CellSort).

### Quantification and statistical analysis

OriginPro 2017 software (OriginLab Corporation, USA) and GraphPad Prism 8 (GraphPad Software, Inc., USA) were used for the statistical analyses and graphing. Offline analysis of the data obtained from electrophysiological recordings was conducted using Clampfit software version 10.7 (Axon Instruments, Inc., USA). The Shapiro-Wilk test was used to check the normality of the data. We conducted statistical comparisons between two groups using paired or unpaired Student's $t$ tests. One-way or Two-way analysis of variance (ANOVA) and Bonferroni post hoc analyses were used in analyses with multiple experimental groups. Nonparametric Mann-Whitney U test or Wilcoxon matched-paired signed rank test was performed if data were not normally distributed. Data are shown as individual values or expressed as the means ± SEMs, and significance levels are indicated as $*P < 0.05$, $**P < 0.01$, $***P < 0.001$ and not significant (n.s.). $P$-values are not provided as exact values when they are less than 0.0001. All illustrations in the manuscript were created using the Adobe Illustrator 2024 software (Adobe Systems Incorporated, USA).

### Reporting summary

Further information on research design is available in the Nature Portfolio Reporting Summary linked to this article.

## Data availability

All data necessary to understand and assess the conclusion of this study are available in the main text or the supplementary materials. There are no restrictions on data availability in the paper. Source data are provided in this paper.

## Code availability

To tracking the same neurons across different time points, longitudinal registration was conducted using Inscopix software as described in ref. 71 and readily available at https://github.com/zivlab/CellReg. Raw $Ca^{2+}$ imaging data were preprocessed using Inscopix software and custom-written scripts in MATLAB as described in ref. 72 and available at https://github.com/mukamel-lab/CellSort.

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

## Acknowledgements

This work was supported by the Plans for Major Provincial Science & Technology Projects (202303a07020002 to Z.Z.), the National Key Research and Development Program of China (STI2030-Major Projects 2021ZD0203100 to Z.Z.), the National Natural Science Foundation of China (32025017 to Z.Z., 32121002 to Z.Z., 32400824 to P.C., 32522044 to P.C., U24A20702 to P.C.), the Anhui Provincial Natural Science Foundation (2308085QH264 to P.C.), the CAS Project for Young Scientists in Basic Research (YSBR-013 to Z.Z.), the Fundamental Research Funds for the Central Universities (WK9100250108 and WK9100000068 to P.C.), and the USTC Research Funds of the Double First-Class Initiative (YD9100002501 to Z.Z.).

## Author contributions

Conceptualization: J.M., P.C., and Z.Z.; Methodology: J.M., X.P., M.Z., W.G., X.Y., and S.W.; Investigation: J.M., X.P., M.Z., and Z.W.; Visualization: J.M., X.P., and M.Z.; Funding acquisition: P.C. and Z. Z.; Project administration: P.C. and Z.Z.; Supervision: Z.Z. and P.C.; Writing – original draft: J.M., M.Z., and P.C.; Writing – review & editing: J.M., X.C., Z.Z., S.W., and P.C.

## Competing interests

The authors declare no competing interests.
