## [Transparent Peer Review file · Nature Communications]

A brain-to-small intestine circuit mediates morphine-induced constipation in male mice

Corresponding Author: Professor Zhi Zhang

Version 0:

Reviewer comments:

Reviewer #1

(Remarks to the Author)

Opioid-induced constipation is an important clinical issue for patients who rely on the analgesic effects of opioids. The prevailing view in the field is that opioids induce constipation via the enteric nervous system. Here, Ma and colleagues challenged this canonical understanding by arguing that morphine leads to intestinal motor stasis by silencing μ -opioid receptor (MOR)-expressing neurons in the paraventricular nucleus (PVN) of the hypothalamus, which decreases the activities of parasympathetic motor neurons in the dorsal motor nucleus of the vagus nerve (DMV). Although the manuscript aims to address a significant health issue, the findings presented here are expected from previous literature. Furthermore, the data provided in the paper do not support that the PVN-DMV-small intestine circuit mediates morphine-induced constipation, and the validity of several experiments is of concern.

Major issues:

The most crucial evidence needed to support the authors' conclusion is not provided, and the current data is logically unsound.

1. The key experiment to prove that the MOR-expressing neurons in the PVN mediate morphine-induced constipation is to conditionally knock out MOR in the PVN, and morphine would no longer induce intestinal stasis. Without this critical experiment, the authors would not be able to differentiate if morphine's effect is mediated by the PVN, or if PVN and other pathways control intestinal motility independently.
2. With the conditional knockout allele, the authors can also evaluate the relative contributions of MOR-expressing enteric neurons and DMV neurons in morphine-induced constipation. These experiments would provide a definitive conclusion on the mechanisms by which morphine suppresses intestinal motility.
3. Evidence provided in the manuscript—using naloxone to reverse the effect of morphine, only shows partial effects, aligning with the independent, additive effect discussed above. The current findings are largely derivable from existing literature and not novel.
4. The DMV is well-known to regulate intestine motility, and the PVN is well-known to be upstream of the DMV. Thus, optogenetic and pharmacogenetic neuronal manipulation experiments add little value to the field. "Fig. 1: Morphine induces constipation in mice." "Fig. 2: DMV neurons directly project to the small intestine."—these conclusions have been known for decades. Figures 1, 2, and 4 contain almost entirely information well-predicted from a large body of previous literature, and Fig. 6 is also well expected. The PVN-DMV-intestine circuit functional manipulation work was also done by others using similar strategies (Wang et al). This reviewer does not appreciate the authors' efforts to reinvent the wheel. In addition, several important concerns question the validity of key experiments.
5. The wireless optogenetics setup only illuminates a tiny proportion of the intestine. It is not conceivable how light stimulation of this small area would drastically impact (2x difference) intestinal motility. The authors should quantify the area covered by the light and address the possibility of non-specific stimulation of neurons in adjacent organs.
6. Proof-of-principle data should be provided to support the validity of the DMV calcium imaging experiment. This reviewer is not aware that imaging neurons in the DMV has been done before. No example video is available. The sample traces, with large "calcium transients" in Fig. 3 are at odds with the neuronal firing kinetics from previous electrophysiological recording data (from the Renehan, Browning, and Travaglini labs). The authors should demonstrate "that the "calcium transients" shown here are genuine neuronal activity and not caused by movement artefacts.
7. Pharmacogenetic manipulation of PVN neuronal axons in the DMV by applying CNO to the brainstem not only activated

these neurons. It is also widely recognised that the PVN projects to the solitary nucleus, which sits adjacent to the DMV and likewise influences intestinal motility (Holt et al). A strategy favouring more specific targeting should be used.

Reviewer #2

(Remarks to the Author)

The paper 'A brain-to-small intestine circuit mediates morphine-induced constipation' by Ma et al., reports a central pathway involved in morphine-induced constipation. The authors make use of a plethora of techniques to support their hypothesis that morphine disrupts the normal PVNGLU⁺ DMVACh⁺ intestines neural signals, such as behavioral monitoring, anatomical tracing, optogenetic / chemogenetic activation/inactivation, in vivo calcium imaging, and slice electrophysiology. My major concern is that the paper is framed in a way to overstate its novelty. The projection from PVN  DMV  intestines has been previously described in multiple groups to promote gut motility. What is novel here is that morphine can inhibit this circuit, which can lead to morphine-induced constipation, and that stimulating these projections can partially compensate for morphine-induced constipation. However, by framing this study as a 'morphine-induced constipation circuit', the authors are downplaying/ ignoring some of the existing literature in order to over highlight morphine's role.

See below for a list of specific comments.

- 1) Some of these results have been reported previously, but no citations are given to indicate what is or is not novel. For example, multiple previous studies have reported that Chat+ DMV neurons innervate the digestive tract including the small intestine (see Tao 2021, or review Travagli 2006). Additionally, multiple studies have identified that PVN DMV intestinal connection is functionally relevant for gastric motility (see Carson 2024, Wang 2023, etc). In general, more attention needs to be paid to the scholastic rigor to ensure prior works are adequately referenced.
- 2) The authors lay out one pathway involved in motility (PVNGLU⁺ DMVACh⁺ intestines). However, they do not explain why this pathway is specific to morphine-induced constipation, nor do they explore any alternatives in the results or discussion. For example, they stated that because the vagus nerve is well known for mediating brain-to-gut communication (In 98) they would look at the DMV. But there are additional sympathetic pathways that are also involved in brain-to-gut communication. Similarly, the authors searched for projections to the DVM and found multiple hits (Fig S6), They then state that since the PVN had the largest number of labelled cells and is involved in gastric function they would only consider it going forward. The authors should meaningfully engage with how specific they believe this pathway to be, and what alternative pathways might also be at play.
- 3) Imaging deep brain structures in freely moving animals is incredibly technically challenging, yet little data was provided to point to the feasibility of these experiments. For example, the only 'raw' imaging data shown is Figure 5d, where a single frame from the imaging window shows some active cell bodies and an over-exposed region in the lower right quadrant. No similar image is provided for DMV recordings. Ideally, the authors would include a supplemental video of the raw calcium imaging so that the stability of recordings can be assessed by readers.
- 4) The calcium imaging experiments are in general poorly described and more methodical transparency is needed to trust the results. For example, how many cells were detected in each animal using pca/ica, and how many of those were manually eliminated (In 557-558). How were calcium 'events' defined (Many of the raw traces shown in 3c, 5e have unstable baselines, meaning the details of the algorithm used can drastically change the results, but this information is not provided). Furthermore, I am confused by the analysis in figures 3d, 5f. The authors show a paired analysis for individual cells between morphine and saline conditions, indicating they are claiming these are the same cells (which is difficult to do with miniscope imaging). If the same cells are tracked across the same animals, it is crucial to know how many animals this data came from (not reported), and what the presentation order of the stimuli was. Given that there are only 28 (DMV) or 24 (PVN) cells, without a sufficient cohort it is hard to argue that firing rates are due to different injections as opposed to imaging quality across days.

Minor comments

- 1) The authors need to report the n of animals used in these different experiments, especially for the physiological studies. It is difficult to assess the robustness of these results without this information.
- 2) The authors repeatedly reuse white/black colored dots/bars to reference features of different groups. This sometimes becomes very confusing within the same figure; for example in figure 3 the 'control' of one experiment (f-h) involved morphine' and the 'control' of another experiment (j-l) did not, but both were in white. Using more colors would really help with clarity.
- 3) The calibration performed in figure 2h-k is only validating the viral optogenetic construct, and as such belongs in the supplement. Additionally, the entire optogenetic experiment is fairly redundant with the chemogenetic activation of DMV described in figure 3.
- 4) Two experiments were performed with DREADDs activation in either DMV (fig 3f) or PVN (fig 6j) in morphine injected mice. In both cases, this activation reduced the severity of the constipation phenotype, but did not recover it to baseline. It would be interesting in the discussion to touch on why this might be.
- 5) Why was the LED glued to the stomach for intestinal activation (In 502-503)

Reviewer #3

(Remarks to the Author)

In the present study, Ma et al. investigated the role of brain-to-gut circuitry in the regulation of opioid-induced constipation in mice. The authors report a neural circuit linking the brain to the intestine via the vagus nerve, which potentially mediates

constipation. Specifically, the authors claim that opioid-sensitive fibers arise from the paraventricular nucleus of the hypothalamus (PVH) to regulate the activity of intestine-innervating cholinergic neurons in the dorsal motor nucleus of the vagus (DMV). The excitation of either PVH or DMV neurons attenuates the morphine effects. The authors concluded that developing drugs that target brain circuits may be relevant for treating medication-resistant opioid-induced constipation.

The treatment of severe constipation caused by chronic opioid use is challenging. The present study attempts to identify new ways to address this issue. However, there are some important limitations.

First, the issue of central vs. peripheral mechanisms of opioid-induced constipation has not been fully addressed. The authors emphasized the roles of PVH and DMV in inducing constipation signs, but it is unclear whether peripheral receptors produce the symptoms independently of central receptors. A study using peripherally-restricted receptor antagonists would have provided a better picture of the physiology of this phenomenon.

The second issue is that although most motility measurements were performed on small intestinal tissue (see Figure 1F, depicting changes in the small intestinal transit rate at different morphine concentrations), the colon may be more relevant as one of its main functions is to absorb water and electrolytes and convert them into solid feces (PMID: 20011411). What are the effects of central manipulations on the colon? DMV cholinergic neurons innervate the stomach, small intestine, and proximal colon, promoting peristalsis (PMID: 25428846).

The same question applies to data shown in Figure 2 H-K, in which the authors only showed that small intestinal vagus nerve activities increased during optogenetic activation of DMV neurons.

The third concern is related to the specific role of MOR1 in mediating the observed effects. Note that DMV highly expresses Opioid Receptors *Oprm1* (MOR1) and *Oprl1* (NOP) (<https://mouse.brain-map.org>). The result of Figure 3 shows that morphine inhibits DMV neuronal activity and induces constipation, which could be a direct effect of morphine. Figure 5B shows MOR1 expression in PVH→DMV projections. The authors concluded that morphine-induced PVH neuronal activity decreases due to MOR1 expression synapses. However, PVH neurons also highly express *Oprl1* receptors (<https://mouse.brain-map.org>), which the authors did not consider.

Finally, DMV stimulation induces mucus secretion in the intestine (PMID: 39121857). The authors should consider and comment on the possibility that such effects also influence the migration of fecal boluses across the intestine, independent of changes in motility.

Version 1:

Reviewer comments:

Reviewer #1

(Remarks to the Author)

The authors have sufficiently revised the manuscript. I congratulate the authors for the hard work and support the publication of this manuscript.

Reviewer #2

(Remarks to the Author)

Thank you for largely addressing my points, and I welcome the addition of your new experiments which strengthen the paper. However, I still have some concerns about the calcium imaging experiments.

1) I am confused by some of the language you added to describe your cell inclusion criteria: "The manual verification protocol required precise temporal alignment between somatic fluorescence signal transients in imaging videos and the timing of onset of corresponding calcium events in trace plots (as illustrated below). Temporally mismatched cellular traces were classified as false-positive identifications and rejected, while those exhibiting complete co-occurrence were validated as accurate detections and accepted." (In 809-814 of the response). What do you mean by aligning 'somatic fluorescent signal transients' and 'onset of corresponding calcium events in trace plots'? Do you just mean that you manually marked calcium events (if so, there are established algorithms for this, which can identify transients across conditions in an unbiased way which should be applied). The referenced figure is just a screen shot of Inscopix software and a table with raw numbers highlighting the fact that a cell was manually rejected which does not clarify anything. Additionally, what is a temporally mismatched cellular trace? Are you referencing a calcium trace with atypical rise/decay dynamics? A cell footprint with low correlation among pixels? Please clarify this language.

2) Thank you for attaching the supplemental videos showing raw imaging. However, as suspected for freely moving imaging experiments in deep brain structures, there are some serious motion artifacts present (see for example ~23s into supplemental video 2. These sorts of large motion can be erroneously detected as calcium events, and since they will occur in all of the cells at a given time point can skew results (imaging that after receiving morphine the animal sat more still and so had fewer motion artifacts, and therefore fewer detected calcium events). Your methods do not describe any exclusion criteria for motion artifacts (such as eliminating time windows from analysis surrounding unstable imaging times), but this should be incorporated.

3) In your response you indicate that ~30 cells were detected per animal then eliminated manually to get to ~10. While some manual curation is typical in the field, eliminating 2/3 of the cells is higher than typical, and I worry that some of the manual selection could be skewing the results. Additionally, you only reported 25-28 cells for each of the groups but stated that 5 animals were in each group. This points to closer to an average of 5-6 cells per animal. Much of the criteria you mentioned using to manually reject cells could be automated (for example calculating the circularity or distance to the lens edge of a given footprint and setting reported cutoff values) for inclusion that can be applied evenly to all datasets. Again, while I understand that manual curation can be common in the field, when it is used to this extent the worry is that inclusion or exclusion of individual data points might be severely skewing your results.

**Response to Reviewers**

**Manuscript ID:** NCOMMS-25-01400-T

**Title:** A brain-to-small intestine circuit mediates morphine-induced constipation

We sincerely appreciate the positive and helpful feedback from the Editor and Reviewers. In
light of these thoughtful comments, we have performed additional experiments to address each
of their specific concerns. We have also substantially revised the manuscript and incorporated
the guidance from the comments into the revised manuscript. The revised version of our study,
with tracked changes (**highlighted in blue**), has been uploaded as a separate file. Our point-by-
point responses to the Reviewers' questions are presented below.

**Contents:**

Response to Reviewer 1: Page 2-18

Response to Reviewer 2: Page 19-35

Response to Reviewer 3: Page 36-43

**Reviewer's comments:**

**Reviewer #1:**

Opioid-induced constipation is an important clinical issue for patients who rely on the analgesic
effects of opioids. The prevailing view in the field is that opioids induce constipation via the
enteric nervous system. Here, Ma and colleagues challenged this canonical understanding by
arguing that morphine leads to intestinal motor stasis by silencing μ -opioid receptor (MOR)-
expressing neurons in the paraventricular nucleus (PVN) of the hypothalamus, which decreases
the activities of parasympathetic motor neurons in the dorsal motor nucleus of the vagus nerve
(DMV). Although the manuscript aims to address a significant health issue, the findings
presented here are expected from previous literature. Furthermore, the data provided in the
paper do not support that the PVN-DMV-small intestine circuit mediates morphine-induced
constipation, and the validity of several experiments is of concern.

**Major issues**

The most crucial evidence needed to support the authors' conclusion is not provided, and the
current data is logically unsound.

1. The key experiment to prove that the MOR-expressing neurons in the PVN mediate
morphine-induced constipation is to conditionally knock out MOR in the PVN, and morphine
would no longer induce intestinal stasis. Without this critical experiment, the authors would not
be able to differentiate if morphine's effect is mediated by the PVN, or if PVN and other
pathways control intestinal motility independently.

**Response:** We would first like to thank the Reviewer for their careful examination of our text
and constructive insights, which have guided several improvements to our study. We agree that
this would be a valuable experiment to validate our conclusions.

Although conditional KO of MOR in PVN presents several non-trivial technical obstacles,
following the Reviewer's advice, we constructed a Cre-dependent AAV virus expressing
*Oprm1* short-hairpin RNAs (shMOR) to induce MOR knockdown specifically in PVN^{Glu}
neurons. We injected the rAAV-CMV-DIO-(EGFP-U6)-shRNA (*Oprm1*)-WPRE-hGH pA
(AAV-DIO-shMOR-EGFP) virus or AAV-DIO-EGFP vector control into the bilateral PVN of
*Vglut2-Cre* mice (please see Response Document Fig. 1a, and also see new Fig. 6a). After
three weeks of virus expression, immunofluorescence staining confirmed that EGFP⁺ signal
largely co-localized with signal from glutamate-specific antibody in PVN, indicating specific
targeting of PVN^{Glu} neurons (please see Response Document Fig. 1b, and also see new Fig. 6b).
Western blot analysis showed that MOR protein levels were significantly reduced in PVN^{Glu}
neurons of mice injected with AAV-DIO-shMOR-EGFP compared to AAV-DIO-EGFP-
infected controls, validating the efficacy of MOR knockdown (please see Response Document
Fig. 1c, and also see new Fig. 6c).

Next, we examined neuronal activity in PVN^{Glu} neurons of morphine-treated mice with
MOR knockdown. As expected, whole-cell patch-clamp electrophysiological recordings
showed that PVN^{Glu} neuronal activity was significantly higher in the AAV-DIO-shMOR-EGFP
group than that in the AAV-DIO-EGFP-expressing controls (please see Response Document
Fig. 1d,e and also see new Fig. 6d,e). Subsequent analysis of morphine-induced constipation in
*Vglut2-Cre* mice injected with AAV-DIO-shMOR-EGFP revealed that PVN^{Glu}-specific MOR
knockdown resulted in significantly higher fecal output and fecal water contents, as well as
significantly higher transit rates and motility in small intestine of morphine-treated mice
compared to that of corresponding AAV-DIO-EGFP control animals (please see Response
Document Fig. 1f-j, and also see new Fig. 6f-h and supplementary Fig. 9a-c). These findings
provide direct evidence that MOR in PVN^{Glu} neurons contributes to morphine-induced
constipation.

The PVN is well-established as a key center for integrating autonomic regulation, stress
responses, and emotional processing^{1,2}. Previous studies have shown that perinatal high-fat diet
exposure downregulates oxytocin (OXT), but upregulates corticotropin releasing factor (CRF)
inputs from the PVN to DMV, leading to stress-associated gastric dysfunction³. Moreover, the
PVN^{CRF} → DMV^{ACh} pathway has been implicated in the mechanism for modulating gastric
motility⁴, supporting a broader role of the PVN in gastrointestinal regulation. Our current study
expands this understanding by identifying a PVN^{Glu}→DMV^{ACh}→small intestine circuit
essential for opioid-induced constipation. Importantly, our functional and molecular evidence
demonstrates that MOR, specifically expressed in PVN^{Glu} neurons, is required for morphine-
induced intestinal hypomotility. These findings not only establish the PVN→DMV→small
intestine neural pathway as a key component in mediating opioid-induced constipation, but also
highlight a more general role of the PVN in brain-gut interactions beyond its previously
described functions.

Regarding the Reviewer's concern about whether other pathways control intestinal
motility, we first used a rabies virus (RV)-based monosynaptic retrograde tracing strategy,
similar to our approach in original Figure S6, to assess MOR expression in multiple upstream
brain regions projecting to DMV. Immunofluorescent staining for MOR and quantitative
analysis of EGFP⁺/MOR⁺ co-labelled cells across several regions, including the parabrachial
pigmented nucleus (PBP), inferior colliculus (IC), primary somatosensory cortex (S1),
periaqueductal gray (PAG), locus coeruleus (LC), mesencephalic reticular formation (mRt),
raphe magnus nucleus (RMg), parasubthalamic nucleus, dorsal raphe nucleus (DRN), and
retrorubral field (RRF). Among all of these brain regions, the PVN had the highest number of
EGFP⁺/MOR⁺ dual-labelled cells (please see Response Document Fig. 2a-c, and also see new
supplementary Fig. 6d,e).

To investigate whether and how MOR expression in these regions may be functionally
 relevant to morphine-induced constipation, we bilaterally implanted cannulas into the S1, IC,
 PAG, PBP, and LC, respectively. After one week of recovery and acclimation, we injected
 morphine, and two hours later performed local infusions of naloxone or ACSF. We found that
 blocking MOR in these five brain regions did not significantly alter fecal output or small
 intestinal motility compared to that of ACSF-infused vehicle controls (please see Response
 Document Fig. 3, and also see new supplementary Fig. 8), suggesting that, unlike the PVN,
 MOR expression in these brain regions does not likely play a major role, if any, in morphine-
 induced constipation.

We also carefully considered potential alternative brain regions projecting to the DMV
 that might contribute to the regulation of peripheral organ function. For example, the
 S1HL→DMV circuit has been shown to regulate splenic immune responses under acute pain⁵;
 the IC could also potentially control the peripheral immune system via the sympathetic and
 parasympathetic systems⁶; a PAG→DMV circuit has been implicated in diarrhea, visceromotor
 response, and pain-related behaviors, although the precise mechanisms remain unclear⁷; DMV-
 projecting DRN neurons have been shown to regulate gastrointestinal function in the context
 of emotional stress⁸; and finally, MORs in the mRt, RMg, and RRF were found to be more
 closely associated with analgesia⁹⁻¹¹, addiction, and withdrawal reactions¹²⁻¹⁵. Despite the
 presence of these alternative circuits, our data suggest that the PVN→DMV pathway plays a
 prominent role in mediating morphine-induced constipation.

These new results and their corresponding descriptions are presented in the revised
 manuscript.

**Response Document Figure 1. Conditional MOR knockdown in PVN^{Glu} neurons alleviates**
 **morphine-induced constipation.**

**a**, Schematic illustrating the strategy for MOR knockdown in the PVN. **b,c**, Representative
 images showing viral expression (**b**) and western blot analysis (**c**) of MOR protein levels in the
 PVN (n = 4 mice per group; *P* = 0.0003). The samples were obtained from the same experiment,

and the gels were processed in parallel. Scale bars, 50 μm and 5 μm (zoom). **d,e**, Representative
 traces (**d**) and summary data (**e**) for evoked action potentials recorded from PVN^{Glu} neurons
 with conditional MOR knockdown (n = 10 cells for Morphine + EGFP, n = 11 cells for
 Morphine + shMOR, $F_{1,19} = 23.10$, $P = 0.0001$). **f-h**, Summary data for fecal weight (**f**), fecal
 water contents (**g**), and small intestinal transit rate (**h**) in EGFP-expressing control mice and
 shMOR-expressing mice following morphine administration (Fecal weight: n = 10, $F_{1,18} =$
 4.748, $P = 0.0059$ for 2 h, $P = 0.0414$ for 4 h, $P = 0.0381$ for 6 h; Water contents: n = 10, $P <$
 0.0001; Transit rate: n = 5, $P = 0.0080$). **i,j**, Representative traces (**i**) and summary data (**j**)
 showing the frequency and amplitude values for small intestinal motility in the EGFP-
 expressing controls and shMOR-expressing mice after morphine injection (n = 6, $P = 0.0189$
 for Frequency; $P < 0.0001$ for Amplitude).
 The data are presented as the mean \pm SEMs. * $P < 0.05$, ** $P < 0.01$, *** $P < 0.001$. Details of
 the statistical analyses are presented in Supplementary Table S1.

 **Response Document Figure 2. MOR expression in brain regions projecting to DMV^{ACh}**
 **neurons.**
 **a**, Schematic diagram for the Cre-dependent retrograde trans-monosynaptic rabies virus (RV)
 tracing strategy. **b**, Representative images showing MOR expression in EGFP-labeled neurons
 across multiple brain regions, identified using a Cre-dependent retrograde trans-monosynaptic
 rabies virus (RV) tracing strategy. Brain regions include: parabrachial pigmented nucleus (PBP),

inferior colliculus (IC), primary somatosensory cortex (S1), periaqueductal gray (PAG), locus
 coeruleus (LC), mesencephalic reticular formation (mRt), raphe magnus nucleus (RMg),
 retrorubral field (RRF), parasubthalamic nucleus (PSTN), and dorsal raphe nucleus (DRN).
 Scale bars, 10 μm . **c**, Summary data showing the distribution of MOR-positive EGFP-labeled
 neurons across these brain regions. The data are presented as the mean \pm SEMs.

 **Response Document Figure 3. Administering MOR antagonist in S1, IC, PAG, PBP or LC**
 **does not affect morphine-induced constipation.**

**a-d**, Representative images (**a**), summary data for fecal weight (**b**), sample traces from strain
 gauge (**c**) and summary data of frequency and amplitude values (**d**) from S1, Scale bar, 100 μm
 152 m ($n = 8$, $F_{1,14} = 0.05293$, $P = 0.8214$ for weight; $n = 8$, $P = 0.7449$ for frequency, $P = 0.2324$
 for amplitude). **e-h**, As indicated in panels **a-d**, but for the region of IC ($n = 8$, $F_{1,14} = 0.3791$,

$P = 0.5480$ for weight; $n = 8$, $P = 0.2318$ for frequency, $P = 0.6870$ for amplitude). **i-l**, As
indicated in panels **a** to **d**, but for the region of PAG ($n = 8$, $F_{1,14} = 0.05882$, $P = 0.8119$ for
weight; $n = 8$, $P = 0.9303$ for frequency, $P = 0.8820$ for amplitude). **m-p**, As indicated in panels
**a** to **d**, but for the region of PBP ($n = 8$, $F_{1,14} = 0.4154$, $P = 0.5297$ for weight; $n = 8$, $P = 0.3870$
for frequency, $P = 0.8465$ for amplitude). **q-t**, As indicated in panels **a** to **d**, but for the region
of LC ($n = 8$, $F_{1,14} = 0.1911$, $P = 0.6687$ for weight; $n = 8$, $P = 0.5058$ for frequency, $P = 0.7852$
for amplitude).

The data are presented as the mean \pm SEMs. Details of the statistical analyses are presented in
Supplementary Table S1.

2. With the conditional knockout allele, the authors can also evaluate the relative contributions
of MOR-expressing enteric neurons and DMV neurons in morphine-induced constipation.
These experiments would provide a definitive conclusion on the mechanisms by which
morphine suppresses intestinal motility.

**Response:** We greatly appreciate this valuable advice regarding possible contributions of
MOR-expressing enteric neurons that may also participate in morphine-induced constipation.
We agree that elucidating the respective roles of peripheral and central MOR populations is
critical for defining the mechanisms driving morphine-induced constipation.

Previous studies have reported that peripherally acting μ -opioid receptor antagonists
(PAMORAs), such as methylnaltrexone, naldemedine, and naloxegol¹⁶⁻¹⁸, effectively alleviate
opioid-induced constipation by selectively blocking MORs in the enteric nervous system
without affecting central opioid analgesia^{19,20}. In particular, the naloxone derivative, Naloxegol,
contains a polyethylene glycol chain that limits its permeability across the blood-brain barrier,
improving its oral bioavailability and reducing first-pass drug metabolism²¹.

We therefore employed naloxegol to assess the involvement of enteric MORs in opioid-
related constipation. For this experiment, mice with morphine-induced constipation were
administered different naloxegol doses (10 mg/kg, 30 mg/kg, and 100 mg/kg) by intraperitoneal
injection at 2 hours post morphine treatment (please see Response Document Fig. 4a, and also
see new supplementary Fig. 10a). The results showed that naloxegol-treated mice had
significantly increased fecal output and small intestinal transit rates compared to saline-treated
vehicle controls (please see Response Document Fig. 4b,c, and also see new supplementary Fig.
10b,c). As 30 mg/kg naloxegol produced the greatest relief of constipation (i.e., 58% higher
fecal output; 70% faster transit rate), this concentration was applied in subsequent experiments
(please see Response Document Fig. 4d, and also see new supplementary Fig. 10d).
Additionally, fecal water content and small intestinal motility also significantly increased
following naloxegol treatment compared to the vehicle control in constipated mice (please see
Response Document Fig. 4e-g, and also see new supplementary Fig. 10e-g). These results

demonstrated that naloxegol could partially relieve morphine-induced constipation by rescuing
 intestinal transit and motility, which was consistent with clinical reports²²⁻²⁴.

In this context, we then examined the relative contributions to morphine-induced
 constipation by MOR-expressing enteric versus MOR-expressing PVN^{Glu} neurons. To this end,
 we infused the PVN of *VgluT2-Cre* mice with AAV-DIO-shMOR-EGFP or the AAV-DIO-
 EGFP control virus, then induced constipation by i.p. morphine injection after three weeks of
 virus expression. Naloxegol treatment at 2 hours post morphine injection (please see Response
 Document Fig. 4h, and also see new Fig. 6i) resulted in significantly increased fecal output and
 small intestinal motility in mice expressing AAV-DIO-shMOR-EGFP (i.e., with conditional
 MOR knockdown in PVN) compared to that of AAV-DIO-EGFP-expressing control mice
 (please see Response Document Fig. 4i-k, and also see new Fig. 6j-l). These results supported
 contributions of both enteric and PVN^{Glu} MOR-expressing neurons in morphine-induced
 constipation.

These new data and related descriptions of the experiments are now included in the revised
 manuscript.

 **Response Document Figure 4. Conditional MOR knockdown in PVN^{Glu} neurons enhances**
 **the effects of naloxegol in relieving morphine-induced constipation.**

a, Schematic diagram for intraperitoneal injection of naloxegol at different concentrations. b,c,
 Summary data for fecal weight (b) and small intestinal transit rate (c) of morphine-treated mice
 at different naloxegol concentrations (Fecal weight: n = 10, $F_{3,36} = 3.899$, $P < 0.0001$ for 4 h; P
 < 0.0001 for 6 h; Transit rate: n = 8, $F_{3,28} = 8.797$, $P = 0.0003$). d, Dose-dependent reduction in

fecal weight at different naloxegol concentrations ($n = 10$, $F_{3,36} = 28.08$, $P < 0.0001$). **e**,
Summary data for fecal water contents following intraperitoneal injection of morphine and
naloxegol ($n = 10$, $P = 0.0025$). **f,g**, Sample traces (**f**) and summary data (**g**) showing the
frequency and amplitude values for small intestinal motility following intraperitoneal injection
of morphine and naloxegol ($n = 8$; Frequency: $P < 0.0001$; Amplitude: $P = 0.0041$). **h**,
Schematic diagram for assessing the effect of PVN^{Glu}-specific MOR knockdown on
constipation in naloxegol-treated mice. **i**, Summary data for fecal weight in EGFP-expressing
controls and shMOR-expressing mice after morphine and naloxegol administration ($n = 8$, $F_{1,14}$
$= 3.035$, $P = 0.0390$ for 2 h, $P = 0.0071$ for 6 h). **j,k**, Sample traces (**j**) and summary data (**k**)
showing the frequency and amplitude values for small intestinal motility in EGFP-expressing
controls and shMOR-expressing mice following morphine and naloxegol injection ($n = 6$;
Frequency: $P = 0.0421$; Amplitude: $P = 0.0115$).
The data are presented as the mean \pm SEMs. * $P < 0.05$, ** $P < 0.01$, *** $P < 0.001$. Details of
the statistical analyses are presented in Supplementary Table S1.

3. Evidence provided in the manuscript--using naloxone to reverse the effect of morphine, only
shows partial effects, aligning with the independent, additive effect discussed above. The
current findings are largely derivable from existing literature and not novel.

**Response:** Thanks for this important comment. We agree that such pharmacological reversal
by naloxone alone offers only partial insights into the mechanisms underlying opioid-induced
constipation. Indeed, as the Reviewer noted, another previous study also reported finding only
partial relief of opioid-associated constipation following naloxone administration²⁵.
Additionally, chemogenetic modulation of the PVN^{Glu}→DMV pathway in our originally
submitted manuscript also demonstrated partial effects of naloxone (Fig. 6a-g and
Supplementary Fig. 14), consistent with the Reviewer's view that peripheral and central
mechanisms contribute additive or independent effects.

Our new results, presented in response to your comment #2, above, thus provide direct
experimental evidence that both MOR-expressing enteric neurons and MOR-expressing
PVN^{Glu} neurons are jointly responsible for morphine-induced suppression of intestinal motility
(please see Response Document Fig. 4h-k). We have expanded the Discussion to clearly
distinguish this study's contributions from that of previous pharmacological studies, and better
emphasize the novelty of our discoveries.

Prior research has predominantly focused on the functional state of enteric nervous system,
which has been shown to regulate gut motility, absorption, and digestion, thereby directly
affecting the severity of constipation. However, targeting MOR in the enteric nervous system
with PAMORAs demonstrates only ~50% efficacy in alleviating opioid-induced constipation,
thus implicating non-peripheral mechanisms in opioid-induced constipation^{16,18,19}. Supporting

this likelihood, intracerebroventricular injection of the MOR antagonist, naloxone, could
alleviate opioid-induced constipation²⁶, suggesting that central MOR expression also
contributes to the gastrointestinal effects of opioids. However, current understanding of the
neural mechanisms driving these phenomena has remained incomplete.

To explore the neural circuits and molecular mechanisms driving morphine-induced
constipation, we employed a multimodal approach, encompassing viral tracing, *in vivo*
microendoscopic calcium imaging, real-time electrophysiological recordings, pharmacology,
and cell-type-specific chemogenetic or optogenetic manipulation, to uncover a previously
uncharacterized PVN^{Glu}→DMV^{Ach}→small intestine brain-to-gut circuit. This neural circuit is
essential for opioid-induced constipation. Specifically, our findings demonstrate that morphine
inhibits PVN^{Glu} neuronal activity through MOR, which in turn leads to suppression of its tonic
NMDA receptor-mediated currents to DMV^{Ach} neurons. This inhibition results in overall
hypoactivity of the circuit mediating small intestinal function, consequently decreasing small
intestinal motility and transit rates, and ultimately manifesting as constipation. This central
mechanism complements the effects of peripheral MOR activation and explains why
PAMORAs alone are only partially effective.

Our study presents several novel findings: first, our study reveals that the PVN mediates
the effects of opioids on the brain-gut axis, providing a new paradigm for integrative,
multimodal research. In contrast, previous studies have largely reported hyperactivation of the
PVN as an autonomic regulatory hub^{1,2} in the hypothalamic-pituitary-adrenal (HPA) axis under
various negative stimuli (*e.g.*, acute or chronic physical stress, high fat diet, etc.). This
hyperactivation can drive excessive release of CRH and CORT, which could disrupt intestinal
flora composition or promote inflammatory responses via enteric glial cell activation, triggering
gastrointestinal dysfunction and accelerating the progression of inflammatory bowel disease^{8,27-}
³⁰. Second, this study identifies MOR on PVN^{Glu} neurons as a key molecule mediating
morphine-induced constipation, resolving a long-standing gap in our understanding of opioid-
induced constipation and suggesting a potentially druggable target for interventions.
Additionally, our study provides a mechanistic framework that could potentially explain other
opioid-related side effects beyond constipation that may arise through modulation of central
autonomic pathways (*e.g.*, nausea, urinary retention, pruritus, and respiratory suppression).

We respectfully assert that these findings extend substantially beyond the existing
literature and enhance our mechanistic understanding of opioid-induced constipation and its
treatment. These new data and interpretations have been incorporated into the revised
manuscript.

4. The DMV is well-known to regulate intestine motility, and the PVN is well-known to be
upstream of the DMV. Thus, optogenetic and pharmacogenetic neuronal manipulation

experiments add little value to the field. “Fig. 1: Morphine induces constipation in mice.” “Fig.
2: DMV neurons directly project to the small intestine.”—these conclusions have been known
for decades. Figures 1, 2, and 4 contain almost entirely information well-predicted from a large
body of previous literature, and Fig. 6 is also well expected. The PVN-DMV-intestine circuit
functional manipulation work was also done by others using similar strategies (Wang et al).
This reviewer does not appreciate the authors’ efforts to reinvent the wheel. In addition, several
important concerns question the validity of key experiments.

**Response:** We thank the Reviewer for the critical feedback and for their opportunity to clarify
and refine our presentation. We acknowledge that some foundational elements of the
PVN→DMV→gastrointestinal tract have been previously reported, but respectfully emphasize
that our study provides new mechanistic insights, cell-type organization, and foundational
circuit-level validation that is absent in previous literature. We respond to each of these specific
concerns in detail, below.

We agree that opioid-induced constipation has been widely investigated, however, the
methods used to establish models vary considerably among studies³¹⁻³³. For instance, dosages
and opioid types for inducing constipation both differ widely, with some studies applying
tramadol (3-100 mg/kg, p.o.) to dose-dependently inhibit small intestinal transit in rats³⁴; while
other studies use oxycodone (10 mg/kg, s.c.) to inhibit small intestinal transit in mice³⁵; and
still other work employing morphine (0.3-3 mg/kg, i.p.) to demonstrate a dose-dependent
decrease in fecal pellet output of mice²⁵. In Fig. 1, to establish a stable mouse model, we
compared multiple morphine doses and quantitatively analyzed changes in fecal output, fecal
water content, motility, and small intestinal transit rates to systematically optimize and validate
morphine-induced constipation.

Given that the colon functions in water and electrolyte absorption to convert waste into
solid feces³⁶, it is likely that the colon also participates in morphine-induced constipation.
Indeed, our new findings indicate that morphine administration significantly reduces the
frequency but increases the amplitude of colonic motility compared to the saline control
treatment (please see Response Document Fig. 5a-c, and also see new Fig. 1f,i,j). However,
MOR knockdown in PVN^{Glu} neurons failed to alleviate colonic dysfunction in our study (please
see Response Document Fig. 5d-f, and also see new supplementary Fig. 9d-f), suggesting that
MOR-expressing PVN^{Glu} neurons may innervate specific subpopulations of DMV^{Ach} neurons
and participate in morphine-induced dysfunction of the small intestinal, but not the colon. This
discovery was facilitated by re-analysis of published data from an unbiased transcriptional
profiling of DMV cholinergic neurons that found distinct DMV neuronal subpopulations send
discrete projections to different organs, such as the stomach, pancreas, cecum, small intestine³⁷.
Overall, the results in Fig. 1 depict a sufficiently detailed and reproducible model framework

to support the reliability of our downstream neural circuit investigations and provide a reliable
reference for future studies.

For Fig. 2 and Fig. 4, we agree that the existence of DMV projections to the gut and PVN
inputs to DMV have long been recognized. However, our work goes beyond confirming these
connections with simple fluorescent staining assays or verifying the effects on gastric or
pancreatic function^{3,38-40}. Instead, we combine advanced 3D tissue clearing, Cre-dependent
three-level retrograde monosynaptic tracing, optogenetic modulation and *in vivo*
electrophysiological recordings to construct a complete and functionally validated
$PVN^{Glu} \rightarrow DMV^{ACh} \rightarrow$ small intestine circuit with high anatomical precision, which is rarely
achieved in a single study. Moreover, our results demonstrate that morphine suppresses tonic
NMDA receptor-mediated currents in DMV^{ACh} neurons, a mechanism that has not been
previously reported in the context of small intestinal motility regulation. Collectively, these
results minimize tracing artifacts, substantially refine the brain-gut circuit map, and extend
current understanding of excitatory control by the $PVN \rightarrow DMV \rightarrow$ small intestine axis.

The PVN is indeed well-understood to serve as an important autonomous control center
due to its secretion of multiple peptides that function in coordinating autonomic gastrointestinal
responses to stress^{3,41}. Wang et al. found that gastric dilation induces gastric motility disorders
through PVN projections secreting corticotropin-releasing hormone (CRH) to DMV^{ACh} . In
addition, their study demonstrated that this effect is derived by inhibitory postsynaptic currents
from PVN^{CRH} neurons to DMV^{ACh} neurons. However, we identified a $PVN^{Glu} \rightarrow DMV^{ACh} \rightarrow$ small
intestine circuit involved in morphine-induced constipation, wherein morphine reduces tonic
NMDA receptor-mediated currents from PVN^{Glu} neurons to small intestine-projecting DMV^{ACh}
neurons. While we appreciate that our studies share anatomical and functional similarities in
research topics, we maintain the important fundamental differences in both the scientific
questions and related neural mechanisms between our work and research conducted by Wang
and colleagues.

**Response Document Figure 5. MOR knockdown in PVN^{Glu} neurons does not affect**
 **morphine-induced inhibition of colonic motility.**

**a**, Schematic diagram for monitoring colonic motility using a strain gauge. **b,c**, Sample traces
 **(b)** and summary data **(c)** showing the frequency and amplitude of colonic motility in saline-
 treated controls and morphine-treated mice (n = 5 mice; Frequency: $P = 0.0144$; Amplitude: P
 = 0.0019). **d**, Schematic diagram for MOR knockdown in the PVN^{Glu} neurons of morphine-
 treated mice. **e,f**, Sample traces **(e)** and summary data **(f)** showing the frequency and amplitude
 values for colon motility in the PVN^{Glu} neurons-specific MOR knockdown mice after morphine
 injection (n = 5, $P = 0.2910$ for frequency, $P = 0.9904$ for amplitude).

The data are presented as the mean \pm SEMs. * $P < 0.05$, ** $P < 0.01$, n.s., not significant. Details
 of the statistical analyses are presented in Supplementary Table S1.

5. The wireless optogenetics setup only illuminates a tiny proportion of the intestine. It is not
 conceivable how light stimulation of this small area would drastically impact (2x difference)
 intestinal motility. The authors should quantify the area covered by the light and address the
 possibility of non-specific stimulation of neurons in adjacent organs.

**Response:** We appreciate the Reviewer's careful examination of our manuscript. We regret
 omitting sufficient detail about the wireless optogenetic stimulation parameters and setup,
 which may have led to this concern.

As the Reviewer correctly pointed out, the illuminated area covers a limited portion of the
 small intestine. However, the strain gauges used to detect intestinal wall activity were placed
 on the opposite side of the stimulation site. Although the light does not directly illuminate the
 entire intestine, local stimulation of ChR2-containing DMV^{Ach} fibers around the intestinal wall
 was sufficient to modulate motility in the region where measurements were acquired. To better
 illustrate this new experimental design, we now provide schematic illustrations accompanied

with photographs of the wireless optogenetic system, including the spatial relationship between
the light-emitting probe and the recording site, in the revised manuscript (please see Response
Document Fig. 6a-c, and also see new Fig. 2h,i).

Regarding the Reviewer's point about the observed two-fold change in motility, we would
like to emphasize that the recordings were conducted in morphine-treated mice, which exhibit
markedly suppressed baseline intestinal motility. In this low-activity context, even modest
activation can translate as a proportionally large increase, making the observed changes
physiologically plausible. Notably, similar approaches have been employed in previous studies
investigating DMV control of gastric motility, which also reported comparably robust effects
following focal optogenetic stimulation⁸.

These new images and related descriptions are now included in the revised manuscript.

**Response Document Figure 6. Experimental design and placement sites of strain gauge**
**and wireless optogenetic devices for assessing small intestine motility in mice.**

a, Schematic illustrating the placement sites for the strain gauge and wireless optogenetic
device on the small intestine. b,c, Representative photographs showing the strain gauge and
wireless optogenetic device under resting conditions (b) and during illumination (c).

6. Proof-of-principle data should be provided to support the validity of the DMV calcium
imaging experiment. This reviewer is not aware that imaging neurons in the DMV has been
done before. No example video is available. The sample traces, with large “calcium transients”
in Fig. 3 are at odds with the neuronal firing kinetics from previous electrophysiological
recording data (from the Renehan, Browning, and Travagli labs). The authors should
demonstrate “that the “calcium transients” shown here are genuine neuronal activity and not
caused by movement artefacts.

**Response:** We thank the Reviewer for raising this important point. As Reviewer mentioned,
calcium imaging in the DMV presents unique technical challenges due to its anatomical
location adjacent to the brainstem. To ensure the validity and reliability of our recordings, we
engineered a customized approach for lens stabilization based on previous studies^{5,8}.
Specifically, before implanting the GRIN lens into the DMV, we used medical-grade adhesive
to pre-attach a supportive scaffold composed of flexible tubing and tungsten wires to the lens.
This design effectively minimized motion artifacts and ensured stable recording. As requested,

we have provided detailed images of the assembly process and supplementary videos of the
DMV calcium imaging in the revised manuscript (please see Response Document Fig. 7, and
new supplementary videos S2 and S3).

For image processing, raw data underwent spatial down sampling ($2\times$ binning) followed
by motion artifact correction through Inscopix Data Processing Software (v1.3.1) using a
standardized workflow comprising: i) defective pixel correction and file size reduction during
preprocessing; ii) spatial frequency filtering; iii) motion compensation via frame registration
(normalized cross-correlation); and iv) $\Delta F/F$ normalization through reference frame
subtraction/division. Additionally, v) cell identification employed a hybrid of principal
component analysis/independent component analysis (PCA/ICA) algorithms with manual
verification of region-of-interest (ROI) spatial footprints and stringent exclusion criteria,
including anatomical abnormalities (atypical size/shape or lens boundary proximity), low signal
fidelity ($\text{SNR} < 3$), or signal contamination (spatial overlap $> 60\%$ combined with Pearson's $r >$
0.6 across entire recording sessions) (please see Response Document Fig. 8).

Regarding the Reviewer's concern about the apparent discrepancy between the observed
calcium transients and previously reported DMV electrophysiological data (e.g., Renehan,
Browning, and Travagli labs), we noted that these previous studies were conducted through
whole-cell patch-clamp electrophysiological recordings in brain slices *in vitro*⁴²⁻⁴⁵. By contrast,
we performed microendoscopic calcium imaging in freely moving mice *in vivo* under
physiologically intact conditions. Differences in neuronal firing kinetics between *in vitro* and
*in vivo* preparations—especially when comparing electrophysiology to genetically encoded
calcium indicators—are well recognized and reflect both methodological and contextual factors.

These new results, with a detailed description of the calcium imaging methodology, have
now been added to the revised manuscript.

**Response Document Figure 7. Lens stabilization for deep-brain calcium imaging.**

**a**, Schematic illustrating the reinforcement design. Prior to implantation into the mouse brain,
a supportive scaffold composed of flexible tubing and tungsten wires was pre-attached to the
imaging lens using medical-grade adhesive. **b**, Representative photographs showing the
reinforced lens assembly.

7. Pharmacogenetic manipulation of PVN neuronal axons in the DMV by applying CNO to the
 brainstem not only activated these neurons. It is also widely recognized that the PVN projects
 to the solitary nucleus, which sits adjacent to the DMV and likewise influences intestinal
 motility (Holt et al). A strategy favouring more specific targeting should be used.

**Response:** We thank the Reviewer for this important comment. We agree that the NTS, located
 adjacent to the DMV, plays a key role in regulating stress-induced hypophagia and
 cardiovascular responses, as well as gastrointestinal function, via projections from PVN^{3,46-48},
 as supported by the work of Holt and colleagues⁴⁹⁻⁵³.

To address the Reviewer's concerns regarding the specificity of our previous
 chemogenetic approach, we have conducted additional optogenetic experiments targeting the
 PVN^{Glu}→DMV circuit with greater precision. Specifically, we bilaterally infused AAV-DIO-
 ChR2-mCherry or AAV-DIO-mCherry viruses into the PVN of *VgluT2-Cre* mice, and
 implanted optical fibers above the DMV (please see Response Document Fig. 9a,b, and also
 see new Fig. 7h,i). After three weeks of virus expression, we established the morphine-induced
 constipation mouse model. Upon optical activation of DMV-projecting PVN^{Glu} neurons, these
 ChR2-expressing morphine-treated mice exhibited significantly increased fecal output and
 fecal water contents, with significantly higher transit and small intestinal motility rates
 compared to mCherry-expressing control mice (please see Response Document Fig. 9c-g, and
 also see new Fig. 7j-n). These findings were consistent with our previous data obtained by
 chemogenetic activation and further confirmed the functional relevance of the PVN^{Glu}→DMV
 circuit in regulating morphine-induced constipation.

These new data, with a corresponding description, have been incorporated into the revised
 manuscript.

**Response Document Figure 9. Optical activation of the PVN^{Glu}→DMV pathway alleviates**

**morphine-induced constipation.**

**a,b**, Schematic for the injection of AAV-DIO-ChR2-mCherry virus into the PVN of *Vglut2-*
*Cre* mice (**a**), and representative images showing optical fiber placement and mCherry-labelled
fibers in the DMV (**b**). Scale bars, 50 μm and 10 μm (zoom). **c-e**, Summary data for fecal weight
(**c**), fecal water contents (**d**), and small intestinal transit rate (**e**) in the mCherry-expressing
controls and ChR2-expressing mice following morphine injection and 473 nm laser stimulation
($n = 10$, $F_{1,18} = 0.1220$, $P < 0.0001$ for 4 h; water contents: $n = 10$, $P = 0.0017$; small intestinal
transit: $n = 8$, $P = 0.0360$). **f,g**, Sample traces (**f**) and summary data (**g**) showing the frequency
and amplitude values for small intestinal motility in the mCherry-expressing controls and
ChR2-expressing mice following morphine injection and 473 nm laser stimulation ($n = 10$, $P =$
0.0003 for frequency, $P < 0.0001$ for amplitude).

The data are presented as the mean \pm SEMs. * $P < 0.05$, ** $P < 0.01$, *** $P < 0.001$. Details of
the statistical analyses are presented in Supplementary Table S1.

We would like to take this opportunity to again thank the Reviewer for their professional
critique, insightful guidance, and highly constructive questions, which have helped us to bolster
both the rigor and impact of our conclusions.

**Reviewer #2:**

The paper ‘A brain-to-small intestine circuit mediates morphine-induced constipation’ by Ma
et al., reports a central pathway involved in morphine-induced constipation. The authors make
use of a plethora of techniques to support their hypothesis that morphine disrupts the normal
$PVN^{GLU} \rightarrow DMV^{ACh} \rightarrow$ intestines neural signals, such as behavioral monitoring, anatomical
tracing, optogenetic / chemogenetic activation/inactivation, in vivo calcium imaging, and slice
electrophysiology.

My major concern is that the paper is framed in a way to overstate its novelty. The
projection from PVN \rightarrow DMV \rightarrow intestines has been previously described in multiple groups
to promote gut motility. What is novel here is that morphine can inhibit this circuit, which can
lead to morphine-induced constipation, and that stimulating these projections can partially
compensate for morphine-induced constipation. However, by framing this study as a
‘morphine-induced constipation circuit’, the authors are downplaying/ ignoring some of the
existing literature in order to over highlight morphine’s role.

See below for a list of specific comments.

1. Some of these results have been reported previously, but no citations are given to indicate
what is or is not novel. For example, multiple previous studies have reported that Chat+ DMV
neurons innervate the digestive tract including the small intestine (see Tao 2021, or review
Travagli 2006). Additionally, multiple studies have identified that $PVN \rightarrow DMV \rightarrow$ intestinal
connection is functionally relevant for gastric motility (see Carson 2024, Wang 2023, etc). In
general, more attention needs to be paid to the scholastic rigor to ensure prior works are
adequately referenced.

**Response:** We would first like to thank the Reviewer for their careful examination of our text,
productive comments, and valuable insights, which have together helped guide important
improvements to our study. We acknowledge the Reviewer’s concern that some foundational
elements of the $PVN \rightarrow DMV \rightarrow$ gastrointestinal tract have been previously reported. We have
now cited the relevant literature and included a detailed discussion comparing our findings with
existing studies, highlighting both the novelty of our work and the distinctions from prior
research in the revised manuscript. (Page 15 Lines 423-442)

We respectfully maintain that our study provides new mechanistic insights, cell-type
organization, and circuit-level foundational validation, thus expanding our understanding of
this pathway through evidence that is not available in previous literature. Below, we respond in
detail to each of the Reviewer’s specific comments.

Kindly note that the large majority of previous studies examining projections from
DMV^{ACh} to the gastrointestinal tract have focused on the stomach, whereas there is relatively
limited direct evidence of these projections innervating the small intestine. For example, Tao et

al. focused on *Cck*⁺ and *Pdyn*⁺ DMV neurons projecting specifically to the glandular stomach,
where they mediate opposing functions via acetylcholine versus nitric oxide, respectively³⁷.
Similarly, Travagli's review emphasized the vago-vagal control of gastric motility and its
modulation by descending central nervous system inputs, as well as hormonal and afferent
feedback from the digestive process that can potentially regulate sensitivity of the vago-vagal
reflex⁵⁴. Another recent study from our research team, also highlighting DMV involvement in
stomach dysfunction under chronic stress exposure (e.g., DRN^{5-HT} → DMV^{ACh} → stomach axis),
did not address innervation of the small intestine⁸. In contrast, through advanced viral tracing
and tissue clearing techniques, our study provides direct anatomical and functional evidence of
DMV^{ACh} projections to small intestine. Moreover, we demonstrate that this circuit contributes
to morphine-suppressed small intestinal motility — a functional link not previously
demonstrated in this context.

Notably, Wang et al. reported a PVN^{CRH} → DMV^{ACh} circuit that regulates gastric dilation
via inhibitory postsynaptic currents⁴, while Carson et al. showed that high-fat diet exposure
induces tonic PVN → DMV activity and disrupts brain-stomach stress responses. Both of these
studies focus on stomach function³. Our work extends this concept to the small intestine by
identifying an excitatory PVN^{Glu} → DMV^{ACh} → small intestine pathway, in which the PVN
regulates DMV neuronal activity via both AMPAR-mediated EPSCs and NMDAR-mediated
tonic currents, consequently revealing a previously unreported mechanism of this circuit in
controlling opioid-induced constipation.

To further distinguish our study's contribution to current understanding of this vago-vagal
regulation, we have completed additional experiments that directly validate the role of PVN^{Glu}
neurons in mediating morphine-induced constipation via MOR signaling. Specifically, we
constructed a Cre-dependent AAV virus expressing *Oprm1* short-hairpin RNAs (shMOR) to
induce MOR knockdown specifically in PVN^{Glu} neurons. We injected the rAAV-CMV-DIO-
(EGFP-U6)-shRNA (*Oprm1*)-WPRE-hGH pA (AAV-DIO-shMOR-EGFP) virus or AAV-
DIO-EGFP vector control into the bilateral PVN of *Vglut2-Cre* mice (please see Response
Document Fig. 10a, and also see new Fig. 6a). After three weeks of virus expression,
immunofluorescence staining confirmed that EGFP⁺ signal largely co-localized with signal
from glutamate-specific antibody in PVN, indicating specific targeting of PVN^{Glu} neurons
(please see Response Document Fig. 10b, and also see new Fig. 6b). Western blot analysis
showed that MOR protein levels were significantly reduced in PVN^{Glu} neurons of mice injected
with AAV-DIO-shMOR-EGFP compared to AAV-DIO-EGFP-infected controls, validating the
efficacy of MOR knockdown (please see Response Document Fig. 10c, and also see new Fig.
6c).

Next, we examined neuronal activity in PVN^{Glu} neurons of morphine-treated mice with
MOR knockdown. As expected, whole-cell patch-clamp electrophysiological recordings

showed that PVN^{Glu} neuronal activity was significantly higher in the AAV-DIO-shMOR-EGFP
group than that in the AAV-DIO-EGFP-expressing controls (please see Response Document
Fig. 10d,e, and also see new Fig. 6d,e). Subsequent analysis of morphine-induced constipation
in *VgluT2-Cre* mice injected with AAV-DIO-shMOR-EGFP revealed that PVN^{Glu}-specific
MOR knockdown resulted in significantly higher fecal output and fecal water contents, as well
as significantly higher transit rates and motility in small intestine of morphine-treated mice
compared to that of corresponding AAV-DIO-EGFP control animals (please see Response
Document Fig. 10f-j, and also see new Fig. 6f-h and supplementary Fig. 9a-c). These findings
provide direct evidence that MOR in PVN^{Glu} neurons contribute to morphine-induced
constipation.

Regarding the specificity of DMV^{Ach} projections to peripheral organs, our findings support
that vagus nerve contains intermingled sensory neurons constituting genetically definable
labeled lines with different anatomical connections and physiological roles. Tao et al. first
identified functionally distinct vagal subpopulations targeting different stomach domains³⁷. For
example, the nigro-vagal pathway modulates proximal colon motility via D1-like receptors in
the DMV, reinforcing the concept that different DMV output neurons are tuned to regulate
specific segments of the gut via distinct upstream inputs⁴². Our study expands this
understanding by identifying a PVN^{Glu}→DMV^{Ach}→small intestine circuit essential for opioid-
induced constipation (OIC).

Regarding MOR expression in the DMV, previous studies^{55,56} and in situ hybridization
data from the Allen brain atlas (<https://mouse.brain-map.org>) have shown that *Oprm1*
transcripts are expressed in this region. However, the cellular source of this MOR expression—
whether intrinsic DMV neurons or afferent projections—remains unclear. Our immunostaining
experiments revealed MOR⁺ signals in cells with nerve fiber-like morphology closely
surrounding DMV^{Ach} neuronal somata, and these signals co-localized with *VgluT2* neuronal
terminals in DMV. A recent single-cell RNA-seq study reported that only a minority of DMV^{Ach}
neurons express *Oprm1*, further supporting that MOR in the DMV is predominantly localized
to projection terminals from upstream brain regions rather than intrinsic DMV neurons. This
observation is crucial for understanding the contributions of central- versus peripherally-
expressed MOR to OIC.

Finally, most previous studies have emphasized mechanisms relevant to peripheral opioid
receptors, leading to subsequent development of peripherally acting MOR antagonists
(PAMORAs) as potential interventions for OIC^{20,23,24}, such as the naloxone derivative,
Naloxegol, which contains a polyethylene glycol chain that limits its permeability across the
blood-brain barrier, resulting in improved oral bioavailability and reduced first-pass drug
metabolism²¹. New experiments in our study assessing the involvement of enteric MORs
showed that 30 mg/kg naloxegol produced the greatest relief of constipation (please see

Response Document Fig. 11a-g, and also see new supplementary Fig. 10a-g). Interestingly,
 conditional MOR knockdown by AAV-DIO-shMOR-EGFP in morphine-treated mice
 exhibited significantly greater fecal output and small intestinal motility following naloxegol
 treatment compared to mice injected with the AAV-DIO-EGFP vector control (please see
 Response Document Fig. 11h-k, and also see new Fig. 6i-l). These results demonstrated
 silencing MOR expression in PVN^{Glu} neurons could enhance the effects of naloxegol in
 relieving constipation, supporting a scenario in which both peripheral MOR-expressing neurons
 and central MOR-expressing PVN^{Glu} neurons contribute to morphine-induced constipation.
 These findings may guide development of adjunctive therapies for patients with inadequate
 response to PAMORAs.

These new results, references, and related content have been added to the revised
 manuscript.

 Response Document Figure 10. Conditional MOR knockdown in PVN^{Glu} neurons
 alleviates morphine-induced constipation.

a, Schematic illustrating the strategy for MOR knockdown in the PVN. b,c, Representative
 images showing viral expression (b) and western blot analysis (c) of MOR protein levels in the
 PVN (n = 4 mice per group; $P = 0.0003$). The samples were obtained from the same experiment,
 and the gels were processed in parallel. Scale bars, 50 μm and 5 μm (zoom). d,e, Representative
 traces (d) and summary data (e) for evoked action potentials recorded from PVN^{Glu} neurons
 with conditional MOR knockdown (n = 10 cells for Morphine + EGFP, n = 11 cells for
 Morphine + shMOR, $F_{1,19} = 23.10$, $P = 0.0001$). f-h, Summary data for fecal weight (f), fecal
 water contents (g), and small intestinal transit rate (h) in EGFP-expressing control mice and
 shMOR-expressing mice following morphine administration (Fecal weight: n = 10, $F_{1,18} =$
 4.748, $P = 0.0059$ for 2 h, $P = 0.0414$ for 4 h, $P = 0.0381$ for 6 h; Water contents: n = 10, $P <$
 0.0001; Transit rate: n = 5, $P = 0.0080$). i,j, Representative traces (i) and summary data (j)
 showing the frequency and amplitude values for small intestinal motility in the EGFP-

expressing controls and shMOR-expressing mice after morphine injection ($n = 6$, $P = 0.0189$
 for Frequency; $P < 0.0001$ for Amplitude).
 The data are presented as the mean \pm SEMs. * $P < 0.05$, ** $P < 0.01$, *** $P < 0.001$. Details of
 the statistical analyses are presented in Supplementary Table S1.

**Response Document Figure 11. Conditional MOR knockdown in PVN^{Glu} neurons**
 **enhances the effects of naloxegol in relieving morphine-induced constipation.**

**a**, Schematic diagram for intraperitoneal injection of naloxegol at different concentrations. **b,c**,
 Summary data for fecal weight (**b**) and small intestinal transit rate (**c**) of morphine-treated mice
 at different naloxegol concentrations (Fecal weight: $n = 10$, $F_{3,36} = 3.899$, $P < 0.0001$ for 4 h; P
 < 0.0001 for 6 h; Transit rate: $n = 8$, $F_{3,28} = 8.797$, $P = 0.0003$). **d**, Dose-dependent reduction in
 fecal weight at different naloxegol concentrations ($n = 10$, $F_{3,36} = 28.08$, $P < 0.0001$). **e**,
 Summary data for fecal water contents following intraperitoneal injection of morphine and
 naloxegol ($n = 10$, $P = 0.0025$). **f,g**, Sample traces (**f**) and summary data (**g**) showing the
 frequency and amplitude values for small intestinal motility following intraperitoneal injection
 of morphine and naloxegol ($n = 8$; Frequency: $P < 0.0001$; Amplitude: $P = 0.0041$). **h**,
 Schematic diagram for assessing the effect of PVN^{Glu}-specific MOR knockdown on
 constipation in naloxegol-treated mice. **i**, Summary data for fecal weight in EGFP-expressing
 controls and shMOR-expressing mice after morphine and naloxegol administration ($n = 8$, $F_{1,14}$
 $= 3.035$, $P = 0.0390$ for 2 h, $P = 0.0071$ for 6 h). **j,k**, Sample traces (**j**) and summary data (**k**)
 showing the frequency and amplitude values for small intestinal motility in EGFP-expressing

controls and shMOR-expressing mice following morphine and naloxegol injection (n = 6;
Frequency: $P = 0.0421$; Amplitude: $P = 0.0115$).

The data are presented as the mean \pm SEMs. $*P < 0.05$, $**P < 0.01$, $***P < 0.001$. Details of
the statistical analyses are presented in Supplementary Table S1.

2. The authors lay out one pathway involved in motility (PVN^{GLU}→DMV^{Ach}→intestines).
However, they do not explain why this pathway is specific to morphine-induced constipation,
nor do they explore any alternatives in the results or discussion. For example, they stated that
because the vagus nerve is well known for mediating brain-to-gut communication (In 98) they
would look at the DMV. But there are additional sympathetic pathways that are also involved
in brain-to-gut communication. Similarly, the authors searched for projections to the DVM and
found multiple hits (Fig S6), They then state that since the PVN had the largest number of
labelled cells and is involved in gastric function they would only consider it going forward. The
authors should meaningfully engage with how specific they believe this pathway to be, and
what alternative pathways might also be at play.

**Response:** We appreciate the Reviewer's careful examination of our results and productive line
of reasoning regarding alternative pathways. Following this guidance, we conducted additional
experiments examining the specificity of the PVN^{Glu}→DMV^{Ach}→small intestine pathway in
morphine-induced constipation and exploring other pathways that may also participate in this
process.

First, we used a rabies virus (RV)-based monosynaptic retrograde tracing strategy, similar
to our approach in original Figure S6, to assess MOR expression in multiple upstream brain
regions projecting to DMV. Immunofluorescent staining for MOR and quantitative analysis of
EGFP⁺/MOR⁺ co-labelled cells across several regions, including the parabrachial pigmented
nucleus (PBP), inferior colliculus (IC), primary somatosensory cortex (S1), periaqueductal gray
(PAG), locus coeruleus (LC), mesencephalic reticular formation (mRt), raphe magnus nucleus
(RMg), parasubthalamic nucleus, dorsal raphe nucleus (DRN), and retrorubral field (RRF).
Among all of these brain regions, the PVN had the highest number of EGFP⁺/MOR⁺ dual-
labelled cells (please see Response Document Fig. 12a-c, and also see new supplementary Fig.
6d,e).

To investigate whether and how MOR expression in these regions may be functionally
relevant to morphine-induced constipation, we bilaterally implanted cannulas into the S1, IC,
PAG, PBP, and LC, respectively. After one week of recovery and acclimation, we injected
morphine, and two hours later performed local infusions of naloxone or ACSF. We found that
blocking MOR in these five brain regions did not significantly alter fecal output or small
intestinal motility compared to that of ACSF-infused vehicle controls (please see Response

Document Fig. 13, and also see new supplementary Fig. 8), suggesting that, unlike the PVN,
MOR expression in these brain regions does not likely play a major role, if any, in morphine-
induced constipation.

As the Reviewer mentioned, the central nervous system regulates intestinal motility
through both sympathetic and parasympathetic nerves of the autonomic nervous system^{57,58}.
The sympathetic system typically inhibits motility by suppressing smooth muscle contraction
and glandular secretion, while the parasympathetic system exerts excitatory regulatory effects⁵⁹.
Opioids, via MOR, inhibit neuronal excitability by activating a G protein-coupled pathway that
selectively blocks calcium channels and opens inward-rectifying potassium channels⁶⁰⁻⁶². Based
on these neurophysiological and pharmacological principles, we hypothesized that morphine-
induced constipation is primarily mediated via the parasympathetic nervous system —
specifically, the vagus nerve—a hypothesis supported by other studies⁶³⁻⁶⁵.

We also carefully considered potential alternative brain regions projecting to the DMV
that might contribute to the regulation of peripheral organ function. For example, the
S1HL→DMV circuit has been shown to regulate splenic immune responses under acute pain⁵;
the IC could also potentially control the peripheral immune system via the sympathetic and
parasympathetic systems⁶; a PAG→DMV circuit has been implicated in diarrhea, visceromotor
response, and pain-related behaviors, although the precise mechanisms remain unclear⁷; DMV-
projecting DRN neurons have been shown to regulate gastrointestinal function in the context
of emotional stress⁸; and finally, MORs in the mRt, RMg, and RRF were found to be more
closely associated with analgesia⁹⁻¹¹, addiction, and withdrawal reactions¹²⁻¹⁵. Despite the
presence of these alternative circuits, our data suggest that the PVN→DMV pathway plays a
prominent role in mediating morphine-induced constipation.

These new results and related content are now included in the revised manuscript.

**Response Document Figure 12. MOR expression in brain regions projecting to DMV^{ACh}**
 **neurons.**

**a**, Schematic diagram for the Cre-dependent retrograde trans-monosynaptic rabies virus (RV)
 tracing strategy. **b**, Representative images showing MOR expression in EGFP-labeled neurons
 across multiple brain regions, identified using a Cre-dependent retrograde trans-monosynaptic
 rabies virus (RV) tracing strategy. Brain regions include: parabrachial pigmented nucleus (PBP),
 inferior colliculus (IC), primary somatosensory cortex (S1), periaqueductal gray (PAG), locus
 coeruleus (LC), mesencephalic reticular formation (mRt), raphe magnus nucleus (RMg),
 retrorubral field (RRF), parasubthalamic nucleus (PSTN), and dorsal raphe nucleus (DRN).
 Scale bars, 10 μ m. **c**, Summary data showing the distribution of MOR-positive EGFP-labeled
 neurons across these brain regions. The data are presented as the mean \pm SEMs.

Response Document Figure 13. Administering MOR antagonist in S1, IC, PAG, PBP or LC does not affect morphine-induced constipation.

a-d, Representative images (a), summary data for fecal weight (b), sample traces from strain gauge (c) and summary data of frequency and amplitude values (d) from S1, Scale bar, 100 μ

m ($n = 8$, $F_{1,14} = 0.05293$, $P = 0.8214$ for weight; $n = 8$, $P = 0.7449$ for frequency, $P = 0.2324$

for amplitude). **e-h**, As indicated in panels **a-d**, but for the region of IC ($n = 8$, $F_{1,14} = 0.3791$,

$P = 0.5480$ for weight; $n = 8$, $P = 0.2318$ for frequency, $P = 0.6870$ for amplitude). **i-l**, As

indicated in panels **a to d**, but for the region of PAG ($n = 8$, $F_{1,14} = 0.05882$, $P = 0.8119$ for

weight; $n = 8$, $P = 0.9303$ for frequency, $P = 0.8820$ for amplitude). **m-p**, As indicated in panels

a to d, but for the region of PBP ($n = 8$, $F_{1,14} = 0.4154$, $P = 0.5297$ for weight; $n = 8$, $P = 0.3870$

for frequency, $P = 0.8465$ for amplitude). **q-t**, As indicated in panels **a to d**, but for the region

of LC ($n = 8$, $F_{1,14} = 0.1911$, $P = 0.6687$ for weight; $n = 8$, $P = 0.5058$ for frequency, $P = 0.7852$
 for amplitude).

The data are presented as the mean \pm SEMs. Details of the statistical analyses are presented in
 Supplementary Table S1.

**3. Imaging deep brain structures in freely moving animals is incredibly technically challenging,**
 **yet little data was provided to point to the feasibility of these experiments.** For example, the
 only ‘raw’ imaging data shown is Figure 5d, where a single frame from the imaging window
 shows some active cell bodies and an over-exposed region in the lower right quadrant. No
 similar image is provided for DMV recordings. Ideally, the authors would include a
 supplemental video of the raw calcium imaging so that the stability of recordings can be
 assessed by readers.

**Response:** We thank the Reviewer for raising this important point. Indeed, calcium imaging in
 the DMV presents unique technical challenges due to its anatomical location adjacent to the
 brainstem. To ensure the validity and reliability of our recordings, we engineered a customized
 approach for lens stabilization based on previous studies^{5,8}. Specifically, before implanting the
 GRIN lens into the DMV, we used medical-grade adhesive to pre-attach a supportive scaffold
 composed of flexible tubing and tungsten wires to the lens. This design effectively minimized
 motion artifacts and ensured stable recording. As requested, we have provided detailed images
 of the assembly process and supplementary videos of the DMV calcium imaging in the revised
 manuscript (please see Response Document Fig. 14a-h, and new supplementary videos S2 and
 S3).

**Response Document Figure 14. GRIN lens stabilization for deep-brain calcium imaging**
**and experimental results in the DMV and PVN.**

**a**, Schematic illustrating the reinforcement design. Prior to implantation into the mouse brain,
a supportive scaffold composed of flexible tubing and tungsten wires was pre-attached to the
GRIN lens using medical-grade adhesive. **b**, Representative photographs showing the
reinforced lens assembly. **c**, Representative image showing the field of view obtained through
the GRIN lens in the DMV. **d,e**, Sample traces (**d**) and summary data (**e**) for calcium transients
of GCaMP6m-expressing DMV^{Ach} neurons (n = 28 cells per group, $P < 0.0001$). **f-h**, As
indicated in panels c-e, but for the DMV-projecting PVN neurons (n = 25 cells per group, $P =$
0.0031).

$**P < 0.01$, $***P < 0.001$. Details of the statistical analyses are presented in Supplementary
Table S1.

4. The calcium imaging experiments are in general poorly described and more methodical
transparency is needed to trust the results. For example, how many cells were detected in each
animal using *pca/ica*, and how many of those were manually eliminated (ln 557-558). How
were calcium ‘events’ defined (Many of the raw traces shown in 3c, 5e have unstable baselines,
meaning the details of the algorithm used can drastically change the results, but this information
is not provided). Furthermore, I am confused by the analysis in figures 3d, 5f. The authors show
a paired analysis for individual cells between morphine and saline conditions, indicating they
are claiming these are the same cells (which is difficult to do with miniscope imaging). If the
same cells are tracked across the same animals, it is crucial to know how many animals this
data came from (not reported), and what the presentation order of the stimuli was. Given that
there are only 28 (DMV) or 24 (PVN) cells, without a sufficient cohort it is hard to argue that
firing rates are due to different injections as opposed to imaging quality across days.

**Response:** We sincerely thank the Reviewer for their insightful comments regarding our
calcium imaging experiments. To clarify our methodology, approximately 30 cells per animal
were automatically detected by our system, with ~20 cells subsequently excluded through
rigorous manual curation.

For image processing, raw data underwent spatial down sampling ($2\times$ binning) followed
by motion artifact correction through Inscopix Data Processing Software (v1.3.1) using a
standardized workflow comprising: i) defective pixel correction and file size reduction during
preprocessing; ii) spatial frequency filtering; iii) motion compensation via frame registration
(normalized cross-correlation); and iv) $\Delta F/F$ normalization through reference frame
subtraction/division. Additionally, v) cell identification employed a hybrid of principal
component analysis/independent component analysis (PCA/ICA) algorithms with manual
verification of region-of-interest (ROI) spatial footprints and stringent exclusion criteria,

including anatomical abnormalities (atypical size/shape or lens boundary proximity), low signal
fidelity (SNR < 3), or signal contamination (spatial overlap > 60% combined with Pearson's $r >$
0.6 across entire recording sessions).

Every extracted cell was manually checked for circular spatial footprints, and Ca^{2+}
transients were characterized by sharp rises and slow decays⁶⁶. The manual verification protocol
required precise temporal alignment between somatic fluorescence signal transients in imaging
videos and the timing of onset of corresponding calcium events in trace plots (as illustrated
below). Temporally mismatched cellular traces were classified as false-positive identifications
and rejected, while those exhibiting complete co-occurrence were validated as accurate
detections and accepted (please see Response Document Fig. 15). Following this protocol, only
accepted traces were included in subsequent quantitative analyses. We have now formally
incorporated this calcium event validation pipeline into the revised Methods section.

We understand the Reviewer's concerns regarding the data presentation in Figures 3d and
5f. In our original submission, these figures were intended to display paired comparisons
between the saline and morphine conditions. We now realize that this data presentation was
inappropriate. In fact, prior to our initial submission, we systematically collected pre-saline,
post-saline, pre-morphine, and post-morphine datasets, then selectively presented only post-
saline vs post-morphine comparisons in the main figures to preserve space and provide a
logically clear comparison between the treatment and control. However, we now recognize that
this approach was methodologically unsound and have accordingly revised Figures 3d and 5f
to now present paired comparisons between pre-morphine and post-morphine calcium activity
within the same tracked cells, while corresponding comparisons between pre- and post-saline
treatment activity are shown in new Supplementary Figures 3e,f and 7f,g, along with full
statistical analyses (please see Response Document Fig. 16). Additionally, the figure legends
now clearly state sample sizes ($n = 28$ cells per group for DMV, $n = 25$ cells per group for
PVN), and all relevant statistical data have been provided in Supplementary Table S1.

Regarding the Reviewer's concern about sample size, we would like to clarify that all
calcium imaging data were acquired from 5 mice per group. The relatively small number of
detected neurons per animal reflects the anatomical constraints of the DMV and PVN, as well
as the need to use minimal viral volumes in order to restrict expression within these small target
regions. Furthermore, our viral strategies specifically labelled DMV^{ACh} neurons and DMV-
projecting PVN neurons, which also limited the number of cells that could be imaged in each
mouse. It warrants emphasis that our goal was not to conduct large-scale neuronal population
analyses, but rather to examine changes in morphine-responsive activity within anatomically
and functionally defined circuits, an approach that is consistent with previous studies^{5,8}.

whereas traces exhibiting complete coincidence were validated as accurate detections and
 accepted (included in the final analysis).

 **Response Document Figure 16. Deep-brain calcium imaging results in the DMV and PVN.**
 **a**, Representative images showing a filed view from the GRIN lens in the DMV. **b,c**, Sample
 traces (**b**) and summary data (**c**) for calcium transients of GCaMP6m-expressing DMV^{ACh}
 neurons before and after saline ($n = 28$ cells per group, $P = 0.2304$). **d-f**, As indicated in **a-**
 **c**, but for morphine injection ($n = 28$ cells per group, $P < 0.0001$). **g**, Representative images
 showing a filed view from the GRIN lens in the PVN. **h,i**, Sample traces (**h**) and summary data
 (**i**) for calcium transients of GCaMP6m-expressing PVN neurons before and after saline
 injection ($n = 25$ cells per group, $P = 0.3886$). **j-l**, As indicated in **g-i**, but for morphine
 injection ($n = 25$ cells per group, $P = 0.0031$).

** $P < 0.01$, *** $P < 0.001$, n.s., not significant. Details of the statistical analyses are presented
 in Supplementary Table S1.

**Minor comments**

1. The authors need to report the n of animals used in these different experiments, especially
for the physiological studies. It is difficult to assess the robustness of these results without this
information.

**Response:** We have incorporated sample sizes and detailed statistical information into Figure
legends throughout the revised manuscript.

2. The authors repeatedly reuse white/black colored dots/bars to reference features of different
groups. This sometimes becomes very confusing within the same figure; for example in figure
3 the ‘control’ of one experiment (f-h) involved morphine’ and the ‘control’ of another
experiment (j-l) did not, but both were in white. Using more colors would really help with
clarity.

**Response:** We appreciate this advice. The figures have been modified so that color codes are
now applied consistently to improve the ease of data interpretation in our revised manuscript.

3. The calibration performed in figure 2h-k is only validating the viral optogenetic construct,
and as such belongs in the supplement. Additionally, the entire optogenetic experiment is fairly
redundant with the chemogenetic activation of DMV described in figure 3.

**Response:** As suggested, we have moved the results of optogenetic experiments in Figures 2h-
k to the supplementary materials in the revised manuscript.

Regarding Figures 2i-o, we respectfully maintain that this optogenetic experiment provides
information complementary to rather than redundant with the chemogenetic approach in Figure
3. Specifically, the wireless optogenetic experiments involved activating ChR2-expressing
DMV^{Ach} neuronal terminals directly in the small intestinal wall via an LED probe implanted in
the abdominal cavity. This manipulation enabled recording of changes in intestinal wall activity
through a contralateral strain gauge, which subsequently provided direct functional evidence of
DMV→small intestine pathway *in vivo*. In contrast, the chemogenetic strategy in Figure 3
selectively modulates small intestine-projecting DMV neurons at the soma level and does not
directly assess terminal activity in the gut. Together, these approaches converge to demonstrate
the critical role of the DMV^{Ach}→small intestine pathway in morphine-induced constipation. To
further clarify this distinction, we have added schematics of the respective experimental designs
and photographs of the wireless optogenetic setup in the revised manuscript, including the
spatial relationship between the LED probe and the recording site (please see Response
Document Fig. 17, and also see new Fig. 2h,i).

Response Document Figure 17. Placement sites for the strain gauge and wireless optogenetic device in the small intestine of mice.

a, Schematic illustrating the placement sites for the strain gauge and wireless optogenetic device on the small intestine. **b,c**, Representative photographs showing the strain gauge and wireless optogenetic device under resting conditions (**b**) and during illumination (**c**).

4. Two experiments were performed with DREADDs activation in either DMV (fig 3f) or PVN (fig 6j) in morphine injected mice. In both cases, this activation reduced the severity of the constipation phenotype, but did not recover it to baseline. It would be interesting in the discussion to touch on why this might be.

Response: As we noted in our response to your first comment, we conducted additional experiments assessing the role of intestinal MOR in morphine-induced constipation. We found that naloxegol, a peripherally acting μ -opioid receptor antagonist, significantly alleviated constipation symptoms in morphine-treated mice (please see Response Document Fig. 11a-g, and also see new supplementary Fig. 10a-g). Notably, selective knockdown of MOR expression in PVN^{Glu} neurons further enhanced the constipation-relieving effects of naloxegol compared to that in vector control animals (please see Response Document Fig. 11h-k, and also see new Fig. 6i-l), supporting contributions of both peripheral MOR-expressing neurons and central MOR-expressing PVN^{Glu} neurons in morphine-induced constipation. Therefore, the partial efficacy observed upon chemogenetic activation of either the PVN or DMV is likely due to this approach modulating only the central component, without addressing ongoing inhibition mediated by peripheral MOR.

5. Why was the LED glued to the stomach for intestinal activation (In 502-503).

Response: We thank the Reviewer for pointing out this writing error. We have corrected the statement to now read: “A 400 μ m-long needle (80-130 μ m in diameter) delivering blue light from the wireless optogenetic device (ThinkerTech) was glued to the small intestinal wall using tissue adhesive” (Line 666-667). We have also carefully checked the whole manuscript for other typos.

We would like to take this opportunity to again thank the Reviewer for their very helpful
guidance and highly constructive questions about our study.

**Reviewer #3:**

In the present study, Ma et al. investigated the role of brain-to-gut circuitry in the regulation of
opioid-induced constipation in mice. The authors report a neural circuit linking the brain to the
intestine via the vagus nerve, which potentially mediates constipation. Specifically, the authors
claim that opioid-sensitive fibers arise from the paraventricular nucleus of the hypothalamus
(PVH) to regulate the activity of intestine-innervating cholinergic neurons in the dorsal motor
nucleus of the vagus (DMV). The excitation of either PVH or DMV neurons attenuates the
morphine effects. The authors concluded that developing drugs that target brain circuits may
be relevant for treating medication-resistant opioid-induced constipation.

The treatment of severe constipation caused by chronic opioid use is challenging. The
present study attempts to identify new ways to address this issue. However, there are some
important limitations.

1. First, the issue of central vs. peripheral mechanisms of opioid-induced constipation has not
been fully addressed. The authors emphasized the roles of PVH and DMV in inducing
constipation signs, but it is unclear whether peripheral receptors produce the symptoms
independently of central receptors. A study using peripherally-restricted receptor antagonists
would have provided a better picture of the physiology of this phenomenon.

**Response:** We would first like to thank the Reviewer for their supportive comments and expert
guidance in improving our study. We agree that elucidating the respective roles of peripheral
and central MOR populations is critical for defining the mechanisms driving morphine-induced
constipation.

Previous studies have reported that peripherally acting μ -opioid receptor antagonists
(PAMORAs), such as methylnaltrexone, naldemedine, and naloxegol⁵⁻⁷, effectively alleviate
opioid-induced constipation by selectively blocking MORs in the enteric nervous system
without affecting central opioid analgesia^{7,12}. In particular, the naloxone derivative, Naloxegol,
contains a polyethylene glycol chain that limits its permeability across the blood-brain barrier,
improving its oral bioavailability and reducing first-pass drug metabolism²¹.

We therefore employed naloxegol to assess the involvement of enteric MORs in opioid-
related constipation. For this experiment, mice with morphine-induced constipation were
administered different naloxegol doses (10 mg/kg, 30 mg/kg, and 100 mg/kg) by intraperitoneal
injection at 2 hours post morphine treatment (please see Response Document Fig. 18a, and also
see new supplementary Fig. 10a). The results showed that naloxegol-treated mice had
significantly increased fecal output and small intestinal transit rates compared to saline-treated
vehicle controls (please see Response Document Fig. 18b,c, and also see new supplementary
Fig. 10b,c). As 30 mg/kg naloxegol produced the greatest relief of constipation (i.e., 58% higher
fecal output; 70% faster transit rate), this concentration was applied in subsequent experiments

(please see Response Document Fig. 18d, and also see new supplementary Fig. 10d).
 Additionally, fecal water content and small intestinal motility also significantly increased
 following naloxegol treatment compared to the vehicle control in constipated mice (please see
 Response Document Fig. 18e-g, and also see new supplementary Fig. 10e-g). These results
 demonstrated that naloxegol could partially relieve morphine-induced constipation by rescuing
 intestinal transit and motility, which was consistent with clinical reports²²⁻²⁴.

In this context, we then examined the relative contributions to morphine-induced
 constipation by MOR-expressing enteric versus MOR-expressing PVN^{Glu} neurons. To this end,
 we infused the PVN of *Vglut2-Cre* mice with AAV-DIO-shMOR-EGFP or the AAV-DIO-EGFP
 EGFP control virus, then induced constipation by i.p. morphine injection after three weeks of
 virus expression. Naloxegol treatment at 2 hours post morphine injection (please see Response
 Document Fig. 18h, and also see new Fig. 6i) resulted in significantly increased fecal output
 and small intestinal motility in mice expressing AAV-DIO-shMOR-EGFP (i.e., with
 conditional MOR knockdown in PVN) compared to that of AAV-DIO-EGFP-expressing
 control mice (please see Response Document Fig. 18i-k, and also see new Fig. 6j-l). These
 results supported contributions of both enteric and PVN^{Glu} MOR-expressing neurons in
 morphine-induced constipation.

These new data and related descriptions of the experiments are now included in the revised
 manuscript.

**Response Document Figure 18. Conditional MOR knockdown in PVN^{Glu} neurons**

enhances the effects of naloxegol in relieving morphine-induced constipation.

**a**, Schematic diagram for intraperitoneal injection of naloxegol at different concentrations. **b,c**,
Summary data for fecal weight (**b**) and small intestinal transit rate (**c**) of morphine-treated mice
at different naloxegol concentrations (Fecal weight: $n = 10$, $F_{3,36} = 3.899$, $P < 0.0001$ for 4 h; P
< 0.0001 for 6 h; Transit rate: $n = 8$, $F_{3,28} = 8.797$, $P = 0.0003$). **d**, Dose-dependent reduction in
fecal weight at different naloxegol concentrations ($n = 10$, $F_{3,36} = 28.08$, $P < 0.0001$). **e**,
Summary data for fecal water contents following intraperitoneal injection of morphine and
naloxegol ($n = 10$, $P = 0.0025$). **f,g**, Sample traces (**f**) and summary data (**g**) showing the
frequency and amplitude values for small intestinal motility following intraperitoneal injection
of morphine and naloxegol ($n = 8$; Frequency: $P < 0.0001$; Amplitude: $P = 0.0041$). **h**,
Schematic diagram for assessing the effect of PVN^{Glu}-specific MOR knockdown on
constipation in naloxegol-treated mice. **i**, Summary data for fecal weight in EGFP-expressing
controls and shMOR-expressing mice after morphine and naloxegol administration ($n = 8$, $F_{1,14}$
$= 3.035$, $P = 0.0390$ for 2 h, $P = 0.0071$ for 6 h). **j,k**, Sample traces (**j**) and summary data (**k**)
showing the frequency and amplitude values for small intestinal motility in EGFP-expressing
controls and shMOR-expressing mice following morphine and naloxegol injection ($n = 6$;
Frequency: $P = 0.0421$; Amplitude: $P = 0.0115$).

The data are presented as the mean \pm SEMs. * $P < 0.05$, ** $P < 0.01$, *** $P < 0.001$. Details of
the statistical analyses are presented in Supplementary Table S1.

2. The second issue is that although most motility measurements were performed on small
intestinal tissue (see Figure 1F, depicting changes in the small intestinal transit rate at different
morphine concentrations), the colon may be more relevant as one of its main functions is to
absorb water and electrolytes and convert them into solid feces (PMID: 20011411). What are
the effects of central manipulations on the colon? DMV cholinergic neurons innervate the
stomach, small intestine, and proximal colon, promoting peristalsis (PMID: 25428846). The
same question applies to data shown in Figure 2 H-K, in which the authors only showed that
small intestinal vagus nerve activities increased during optogenetic activation of DMV neurons.

**Response:** We appreciate the Reviewer's perspective. This feedback productive in helping us
to bolster our study. As a key part of the large intestine, we agree that the colon plays a critical
role in water and electrolyte absorption, and ultimately the formation of solid feces^{36,67,68}.
Although most of our motility measurements focused on the small intestine, we also appreciate
the importance of assessing colonic function in the context of morphine-induced constipation,
particularly in relation to central manipulations.

Previous studies have shown that opioids inhibit colonic motility primarily via peripheral
MOR^{25,69}, yet the effects of modulating central MOR activity on colonic function remain
unknown. To address the Reviewer's concern, we conducted additional experiments using real-
time patch recordings of colonic peristalsis, similar to the approach employed in original Figure

1g. In morphine-treated mice with strain gauges fixed to the proximal colon wall, we observed
that frequency of colonic motility was significantly reduced, whereas amplitude increased
compared to that in saline-treated control animals (please see Response Document Fig. 19a-c
and also see new Fig. 1f,i,j). These results indicated that morphine suppresses colon motility,
consistent with previous research²⁵.

To evaluate the effects of conditional MOR knockdown in PVN^{Glu} neurons on colonic
function, we bilaterally injected AAV-DIO-shMOR-EGFP or AAV-DIO-EGFP control virus
into the PVN of *Vglut2-Cre* mice (please see Response Document Fig. 19d and also see new
supplementary Fig. 9d). In morphine-treated mice, real-time patch recordings detected no
significant difference in colonic motility parameters between the AAV-DIO-shMOR-EGFP-
infected group and AAV-DIO-EGFP-infected controls (please see Response Document Fig.
19e,f, and also see new supplementary Fig. 9e,f), suggesting that MOR expression in PVN^{Glu}
neurons does not affect morphine-induced colonic hypomotility. However, we found that mice
with PVN^{Glu}-specific MOR knockdown did exhibit significantly higher fecal water content
(66.69% vs. 31.59% in controls) and increased fecal size (please see Response Document Fig.
19g-i, and also see new Fig. 6f and supplementary Fig. 9b,c), suggesting that MORs on PVN^{Glu}
neurons may influence colonic water absorption, but not appear to directly regulate colonic
motility.

Previous studies have shown that the vagus nerve innervates the small intestine with
higher density fibers compared to the colon^{25,70}. Moreover, other recent work indicates that the
nigro-vagal pathway can modulate tone and motility of the proximal colon via D1-like receptors
in the DMV, implying that the colon and small intestine, although both innervated by DMV, are
regulated by distinct upstream inputs and receptor pathways⁴². This evidence collectively
suggests that morphine-induced colonic hypomotility may be primarily mediated via peripheral
mechanisms or alternative central pathways.

These new results and related content are now included in the revised manuscript.

**Response Document Figure 19. MOR knockdown in PVN^{Glu} neurons does not affect**
 **morphine-induced inhibition of colonic motility.**

**a**, Schematic diagram for monitoring colonic motility using a strain gauge. **b,c**, Sample traces
 **(b)** and summary data **(c)** showing the frequency and amplitude of colonic motility in saline-
 treated controls and morphine-treated mice (n = 5 mice; Frequency: $P = 0.0144$; Amplitude: $P =$
 0.0019). **d**, Schematic diagram for MOR knockdown in PVN^{Glu} neurons of morphine-treated
 mice. **e,f**, Sample traces **(e)** and summary data **(f)** showing the frequency and amplitude values
 for colon motility in the PVN^{Glu} neurons-specific MOR knockdown mice after morphine
 injection (n = 5, $P = 0.2910$ for frequency, $P = 0.9904$ for amplitude). **g**, PVN^{MOR} knockdown
 increased the fecal weight in morphine-induced constipation mice (n = 10, $F_{1,18} = 3.603$, $P =$
 0.0413 for 2 h; $P = 0.0131$ for 4 h; $P = 0.0049$ for 6 h;). **h**, Water contents in PVN^{MOR}
 knockdown was much higher than that in controls (31.59% vs. 66.69%, n = 10, $P < 0.0001$). **i**,
 The representative picture of fecal pellets in PVN^{MOR} knockdown and the controls.
 The data are presented as the mean \pm SEMs. * $P < 0.05$, ** $P < 0.01$, *** $P < 0.001$, n.s., not
 significant. Details of the statistical analyses are presented in Supplementary Table S1.

3. The third concern is related to the specific role of MOR in mediating the observed effects.
 Note that DMV highly expresses Opioid Receptors Oprm1 (MOR) and Oprl1 (NOP)
 (<https://mouse.brain-map.org>). The result of Figure 3 shows that morphine inhibits DMV
 neuronal activity and induces constipation, which could be a direct effect of morphine. Figure

5B shows MOR expression in PVH→DMV projections. The authors concluded that morphine-
induced PVH neuronal activity decreases due to MOR1 expression synapses. However, PVH
neurons also highly express Oprl1 receptors (<https://mouse.brain-map.org>), which the authors
did not consider.

**Response:** We understand the Reviewer’s concerns and greatly appreciate their insight. It
should be mentioned that the endogenous ligand of NOP is nociceptin/orphanin FQ (N/OFQ),
not morphine⁷¹⁻⁷³. Identified as the “fourth opioid receptor” in 1994⁷³, NOP reportedly shares
~60% sequence homology with classical opioid receptors (μ , δ , κ), but possesses distinct
pharmacological profiles and ligand selectivity⁷³. Relevant to this unique functionality, N/OFQ
exhibits no binding affinity for classical opioid receptors (including MOR), and similarly,
morphine shows negligible affinity for NOP⁷⁴. The three-dimensional structure of NOP,
resolved by X-ray crystallography in 2021, further highlights its unique binding specificity for
N/OFQ⁷⁵. Although the synthetic combined MOR/NOP agonist, cebranopadol, can reportedly
alleviate nociceptive and neuropathic pain with greater efficacy⁷⁶, morphine itself acts primarily
via classical opioid receptors (MOR, DOR, and KOR), not NOP. Therefore, morphine-induced
constipation in our current study is most likely mediated by MOR and not NOP.

Regarding the Reviewer’s concern about the MOR expression within the DMV, we
acknowledge that previous studies^{55,56} and in situ hybridization data from the Allen brain atlas
(<https://mouse.brain-map.org>) show that *Oprm1* transcripts are indeed expressed in this region.
However, the cellular source of this MOR expression—whether intrinsic DMV neurons or
afferent projections—remains unclear. In our original manuscript, immunostaining experiments
revealed that MOR⁺ signals in the DMV exhibited a nerve fiber-like morphology, closely
surrounding the DMV^{ACh} neuronal somata, and were co-labelled with VgluT2⁺ axon terminals
(Supplementary Fig. 5d-f). These findings suggest that MOR in the DMV is predominantly
localized on afferent terminals rather than within DMV^{ACh} neuronal somata.

This interpretation is further supported by recent single-cell RNA sequencing data
showing that only a minority of DMV^{ACh} neurons express *Oprm1*³⁷. Additionally, in response
to your first comment, we conducted additional experiments showing that selective knockdown
of MOR in PVN^{Glu} neurons significantly attenuates morphine-induced constipation (**please see**
**Response Document Fig. 18h-k, and also see new Fig. 6i-l**). This finding provides direct
functional evidence that MOR-mediated modulation of DMV^{ACh} neuronal activity arises
predominantly from upstream inputs, especially from the PVN.

4. Finally, DMV stimulation induces mucus secretion in the intestine (PMID: 39121857). The
authors should consider and comment on the possibility that such effects also influence the
migration of fecal boluses across the intestine, independent of changes in motility.

**Response:** We appreciate the Reviewer’s valuable comments on our study. We have carefully

examined the study referenced by the Reviewer, which reports that direct stimulation of the
vagal trunk via implanted cuff electrode was sufficient to induce robust calcium transits across
the Brunner's glands. However, closer scrutiny of their DMV^{Ach} chemogenetic activation
experiments shows that, in fact, they only found enhanced mucus thickness in duodenum, and
not in the ileum or colon, while selectively ablating DMV^{Ach} neurons markedly decreased it.
Most importantly, after injecting a Cre-dependent AAV-hSyn-DIO-GFP virus into the DMV of
*Chat-Cre* mice to label vagal efferent parasympathetic fibers, they observed dense efferent
innervation in Brunner's glands, but not in epithelial goblet mucous cells across the intestine.

As extensive surface area of the mucosal epithelium is potentially exposed to enteric
microorganisms in the gastrointestinal tract, the host produces a complex layer of mucus
covering the entire intestinal tract to protect the mucosa from invasion by pathogenic or even
commensal microbes⁷⁷. This mucus is secreted by goblet cells and typically contains several
major components. In the small intestine, the mucus layer limits the number of bacteria that can
reach the epithelium and Peyer's patches. By contrast, in the large intestine, the inner mucus
layer separates commensal bacteria from the host epithelium and the outer colonic mucus layer
provides a habitable niche for commensal bacteria^{78,79}. Defects in the mucus layer increase
susceptibility to pathogens and have been linked to development of inflammatory bowel
diseases, especially those related to the abovementioned dysfunction in limiting overall
pathogenic and commensal microbe abundance on the gut mucosal surface^{80,81}. Although
modulating DMV^{Ach} neurons does not directly promote mucus secretion by small intestinal
goblet cells, promoting secretion via Brunner's glands can lead to conditions conducive for
proliferation of lactobacilli⁸². As numerous studies have identified a close relationship between
gut microbiota and functional constipation^{83,84}, promoting duodenal mucus secretion may
therefore indirectly affect morphine-induced constipation.

In our present study, we measured fecal water content as an indirect indicator of changes
in intestinal secretory function, given that both mucus and digestive juices substantially
contribute to fecal water content^{85,86}. Notably, we observed that the reduced fecal output
following morphine administration was accompanied by decreased fecal water content,
suggesting a relationship between secretion and output (Figure 1d). Furthermore, as we noted
in response to your second comment, above, PVN^{Glu}-specific MOR knockdown in morphine-
treated mice leads to markedly higher fecal water content compared to that in the corresponding
control group (66.69% vs. 31.59%), suggesting that enhanced intestinal secretion, which likely
includes increased mucus production, may contribute to the observed increase in fecal output
(please see Response Document Fig. 19g-i, and also see new Fig. 6f and supplementary Fig.
9b,c).

We have expanded our discussion of this topic in the revised manuscript.

We again wish to express our sincere gratitude for the supportive comments and valuable
feedback which — we trust the Reviewer will agree — has helped us to substantially improve
our study.

**References:**

- Effinger, D. P. *et al.* Increased reactivity of the paraventricular nucleus of the
hypothalamus and decreased threat responding in male rats following psilocin
administration. *Nat Commun* **15**, 5321, doi:10.1038/s41467-024-49741-9
(2024).
- Rasiah, N. P., Loewen, S. P. & Bains, J. S. Windows into stress: a glimpse at
emerging roles for CRH(PVN) neurons. *Physiol Rev* **103**, 1667-1691,
doi:10.1152/physrev.00056.2021 (2023).
- Carson, K. E., Alvarez, J., Mackley, J. Q., Travagli, R. A. & Browning, K. N.
Perinatal high-fat diet exposure alters oxytocin and corticotropin releasing
factor inputs onto vagal neurocircuits controlling gastric motility. *J Physiol* **601**,
2853-2875, doi:10.1113/jp284726 (2023).
- Wang, X.-y. *et al.* A neural circuit for gastric motility disorders driven by gastric
dilation in mice. *Frontiers in Neuroscience* **17**,
doi:10.3389/fnins.2023.1069198 (2023).
- Zhu, X. *et al.* Somatosensory cortex and central amygdala regulate neuropathic
pain-mediated peripheral immune response via vagal projections to the spleen.
*Nat Neurosci* **27**, 471-483, doi:10.1038/s41593-023-01561-8 (2024).
- Rolls, A. Immunoception: the insular cortex perspective. *Cellular & Molecular*
*Immunology* **20**, 1270-1276, doi:10.1038/s41423-023-01051-8 (2023).
- Wan, J. *et al.* Electroacupuncture Attenuates Visceral Hypersensitivity by
Inhibiting JAK2/STAT3 Signaling Pathway in the Descending Pain Modulation
System. *Front Neurosci* **11**, 644, doi:10.3389/fnins.2017.00644 (2017).
- Dong, W. Y. *et al.* Brain regulation of gastric dysfunction induced by stress.
*Nat Metab* **5**, 1494-1505, doi:10.1038/s42255-023-00866-z (2023).
- Haghparast, A., Gheitasi, I. P. & Lashgari, R. Involvement of glutamatergic
receptors in the nucleus cuneiformis in modulating morphine-induced
antinociception in rats. *Eur J Pain* **11**, 855-862,
doi:10.1016/j.ejpain.2006.12.010 (2007).
- Bie, B. & Pan, Z. Z. Increased glutamate synaptic transmission in the nucleus
raphe magnus neurons from morphine-tolerant rats. *Mol Pain* **1**, 7,
doi:10.1186/1744-8069-1-7 (2005).
- Bie, B., Fields, H. L., Williams, J. T. & Pan, Z. Z. Roles of alpha1- and alpha2-
adrenoceptors in the nucleus raphe magnus in opioid analgesia and opioid
abstinence-induced hyperalgesia. *J Neurosci* **23**, 7950-7957,
doi:10.1523/jneurosci.23-21-07950.2003 (2003).
- Zhang, Z., Wang, X., Wang, W., Lu, Y. G. & Pan, Z. Z. Brain-derived
neurotrophic factor-mediated downregulation of brainstem K⁺-Cl⁻
cotransporter and cell-type-specific GABA impairment for activation of
descending pain facilitation. *Mol Pharmacol* **84**, 511-520,
doi:10.1124/mol.113.086496 (2013).

- Jacquet, Y. F., Klee, W. A., Rice, K. C., Iijima, I. & Minamikawa, J.
Stereospecific and nonstereospecific effects of (+)- and (-)-morphine: evidence
for a new class of receptors? *Science* **198**, 842-845, doi:10.1126/science.199942
(1977).
- Lammel, S., Lim, B. K. & Malenka, R. C. Reward and aversion in a
heterogeneous midbrain dopamine system. *Neuropharmacology* **76 Pt B**, 351-
359, doi:10.1016/j.neuropharm.2013.03.019 (2014).
- German, D. C., Speciale, S. G., Manaye, K. F. & Sadeq, M. Opioid receptors in
midbrain dopaminergic regions of the rat. I. Mu receptor autoradiography. *J*
*Neural Transm Gen Sect* **91**, 39-52, doi:10.1007/bf01244917 (1993).
- Pergolizzi, J. V., Jr., Christo, P. J., LeQuang, J. A. & Magnusson, P. The Use
of Peripheral μ -Opioid Receptor Antagonists (PAMORA) in the Management
of Opioid-Induced Constipation: An Update on Their Efficacy and Safety. *Drug*
*Des Devel Ther* **14**, 1009-1025, doi:10.2147/dddt.S221278 (2020).
- Vijayvargiya, P., Camilleri, M., Vijayvargiya, P., Erwin, P. & Murad, M. H.
Systematic review with meta-analysis: efficacy and safety of treatments for
opioid-induced constipation. *Aliment Pharmacol Ther* **52**, 37-53,
doi:10.1111/apt.15791 (2020).
- Blair, H. A. Naldemedine: A Review in Opioid-Induced Constipation. *Drugs*
**79**, 1241-1247, doi:10.1007/s40265-019-01160-7 (2019).
- Fernández-Montes, A. *et al.* Insights into the Use of Peripherally Acting μ -
Opioid Receptor Antagonists (PAMORAs) in Oncologic Patients: from
Scientific Evidence to Real Clinical Practice. *Curr Treat Options Oncol* **22**, 26,
doi:10.1007/s11864-021-00816-5 (2021).
- Hale, M., Wild, J., Reddy, J., Yamada, T. & Arjona Ferreira, J. C. Naldemedine
versus placebo for opioid-induced constipation (COMPOSE-1 and COMPOSE-
2): two multicentre, phase 3, double-blind, randomised, parallel-group trials.
*Lancet Gastroenterol Hepatol* **2**, 555-564, doi:10.1016/s2468-1253(17)30105-
x (2017).
- Costanzini, A., Ruzza, C. & Neto, J. A. Pharmacological characterization of
naloxegol: In vitro and in vivo studies. *Eur J Pharmacol* **903**, 174132,
doi:10.1016/j.ejphar.2021.174132 (2021).
- Kistemaker, K. R. J. *et al.* Pharmacological prevention and treatment of opioid-
induced constipation in cancer patients: A systematic review and meta-analysis.
*Cancer Treat Rev* **125**, 102704, doi:10.1016/j.ctrv.2024.102704 (2024).
- Slatkin, N. *et al.* Methylnaltrexone for treatment of opioid-induced constipation
in advanced illness patients. *J Support Oncol* **7**, 39-46 (2009).
- Thomas, J. *et al.* Methylnaltrexone for opioid-induced constipation in advanced
illness. *N Engl J Med* **358**, 2332-2343, doi:10.1056/NEJMoa0707377 (2008).
- Matsumoto, K. *et al.* Differences in the morphine-induced inhibition of small
and large intestinal transit: Involvement of central and peripheral μ -opioid

- receptors in mice. *Eur J Pharmacol* **771**, 220-228,
doi:10.1016/j.ejphar.2015.12.033 (2016).
- Tuhin, M. T. H. *et al.* Peripherally restricted transthyretin-based delivery system
for probes and therapeutics avoiding opioid-related side effects. *Nat Commun*
**13**, 3590, doi:10.1038/s41467-022-31342-z (2022).
- Chan, K. L., Poller, W. C., Swirski, F. K. & Russo, S. J. Central regulation of
stress-evoked peripheral immune responses. *Nat Rev Neurosci* **24**, 591-604,
doi:10.1038/s41583-023-00729-2 (2023).
- Schneider, K. M. *et al.* The enteric nervous system relays psychological stress
to intestinal inflammation. *Cell* **186**, 2823-2838.e2820,
doi:10.1016/j.cell.2023.05.001 (2023).
- Wei, W. *et al.* Psychological stress-induced microbial metabolite indole-3-
acetate disrupts intestinal cell lineage commitment. *Cell Metab* **36**, 466-
483.e467, doi:10.1016/j.cmet.2023.12.026 (2024).
- Gao, H. *et al.* μ -Opioid Receptor-Mediated Enteric Glial Activation Is Involved
in Morphine-Induced Constipation. *Mol Neurobiol* **58**, 3061-3070,
doi:10.1007/s12035-021-02286-0 (2021).
- Young, A. *et al.* Mouse model demonstrates strain differences in susceptibility
to opioid side effects. *Neurosci Lett* **675**, 110-115,
doi:10.1016/j.neulet.2018.03.022 (2018).
- Gao, H. *et al.* μ -Opioid Receptor-Mediated Enteric Glial Activation Is Involved
in Morphine-Induced Constipation. *Molecular Neurobiology* **58**, 3061-3070,
doi:10.1007/s12035-021-02286-0 (2021).
- Essmat, N. *et al.* Insights into the Current and Possible Future Use of Opioid
Antagonists in Relation to Opioid-Induced Constipation and Dysbiosis.
*Molecules* **28**, doi:10.3390/molecules28237766 (2023).
- Yasufuku, K. *et al.* Involvement of the Peripheral μ -Opioid Receptor in
Tramadol-Induced Constipation in Rodents. *Biol Pharm Bull* **44**, 1746-1751,
doi:10.1248/bpb.b21-00474 (2021).
- Yang, P. P. *et al.* Delta-opioid receptor antagonist naltrindole reduces
oxycodone addiction and constipation in mice. *Eur J Pharmacol* **852**, 265-273,
doi:10.1016/j.ejphar.2019.04.009 (2019).
- Frattini, J. C. & Nogueras, J. J. Slow transit constipation: a review of a colonic
functional disorder. *Clin Colon Rectal Surg* **21**, 146-152, doi:10.1055/s-2008-
1075864 (2008).
- Tao, J. *et al.* Highly Selective Brain-to-Gut Communication via
Genetically Defined Vagus Neurons. *Neuron* **109**, 2106-2115,
doi:10.1016/j.neuron.2021.05.004 (2021).
- Browning, K. N. & Carson, K. E. Central Neurocircuits Regulating Food Intake
in Response to Gut Inputs-Preclinical Evidence. *Nutrients* **13**,
doi:10.3390/nu13030908 (2021).

- Liang, S. L., Tong, Y. S., Hwang, L. L., Huang, Y. Z. & Chen, C. Y. CART
Peptides Differently Regulate Firing Rates and GABAergic Synaptic Inputs of
DMV Neurons Innervating the Stomach Antrum and Cecum of Adult Male Rats.
*Neuroendocrinology* **112**, 555-570, doi:10.1159/000518690 (2022).
- Love, J. A., Yi, E. & Smith, T. G. Autonomic pathways regulating pancreatic
exocrine secretion. *Auton Neurosci* **133**, 19-34,
doi:10.1016/j.autneu.2006.10.001 (2007).
- Bülbül, M., İzgüt-Uysal, V. N., Sinen, O., Birsen, İ. & Tanrıöver, G. Central
apelin mediates stress-induced gastrointestinal motor dysfunction in rats. *Am J*
*Physiol Gastrointest Liver Physiol* **310**, G249-261,
doi:10.1152/ajpgi.00145.2015 (2016).
- Xing, T., Nanni, G., Burkholder, C. R., Browning, K. N. & Travagli, R. A. The
substantia nigra modulates proximal colon tone and motility in a vagally-
dependent manner in the rat. *The Journal of Physiology* **601**, 4751-4766,
doi:10.1113/jp284238 (2023).
- Browning, K. N., Kalyuzhny, A. E. & Travagli, R. A. μ -Opioid Receptor
Trafficking on Inhibitory Synapses in the Rat Brainstem. *The Journal of*
*Neuroscience* **24**, 7344-7352, doi:10.1523/jneurosci.1676-04.2004 (2004).
- Browning, K. N., Zheng, Z., Gettys, T. W. & Travagli, R. A. Vagal afferent
control of opioidergic effects in rat brainstem circuits. *The Journal of*
*Physiology* **575**, 761-776, doi:10.1113/jphysiol.2006.111104 (2006).
- Browning, K. N., Renehan, W. E. & Travagli, R. A. Electrophysiological and
morphological heterogeneity of rat dorsal vagal neurones which project to
specific areas of the gastrointestinal tract. *J Physiol* **517 (Pt 2)**, 521-532,
doi:10.1111/j.1469-7793.1999.0521t.x (1999).
- Gama de Barcellos Filho, P., Dantzer, H. A., Hasser, E. M. & Kline, D. D.
Oxytocin and corticotropin-releasing hormone exaggerate nucleus tractus
solitarius neuronal and synaptic activity following chronic intermittent hypoxia.
*J Physiol* **602**, 3375-3400, doi:10.1113/jp286069 (2024).
- Savić, B., Murphy, D. & Japundžić-Žigon, N. The Paraventricular Nucleus of
the Hypothalamus in Control of Blood Pressure and Blood Pressure Variability.
*Front Physiol* **13**, 858941, doi:10.3389/fphys.2022.858941 (2022).
- Ruyle, B. C. *et al.* Paraventricular nucleus projections to the nucleus tractus
solitarius are essential for full expression of hypoxia-induced peripheral
chemoreflex responses. *J Physiol* **601**, 4309-4336, doi:10.1113/jp284907
(2023).
- Holt, M. K. *et al.* Modulation of stress-related behaviour by preproglucagon
neurons and hypothalamic projections to the nucleus of the solitary tract.
*Molecular Metabolism* **91**, doi:10.1016/j.molmet.2024.102076 (2025).
- Holt, M. K. *et al.* Synaptic Inputs to the Mouse Dorsal Vagal Complex and Its
Resident Preproglucagon Neurons. *J Neurosci* **39**, 9767-9781,
doi:10.1523/jneurosci.2145-19.2019 (2019).

- Holt, M. K. *et al.* PPG neurons in the nucleus of the solitary tract modulate heart
rate but do not mediate GLP-1 receptor agonist-induced tachycardia in mice.
*Mol Metab* **39**, 101024, doi:10.1016/j.molmet.2020.101024 (2020).
- Holt, M. K. & Rinaman, L. The role of nucleus of the solitary tract glucagon-
like peptide-1 and prolactin-releasing peptide neurons in stress: anatomy,
physiology and cellular interactions. *British Journal of Pharmacology* **179**, 642-
658, doi:10.1111/bph.15576 (2021).
- Holt, M. K. The ins and outs of the caudal nucleus of the solitary tract: An
overview of cellular populations and anatomical connections. *J*
*Neuroendocrinol* **34**, e13132, doi:10.1111/jne.13132 (2022).
- Travagli, R. A., Hermann, G. E., Browning, K. N. & Rogers, R. C. Brainstem
Circuits Regulating Gastric Function. *Annual Review of Physiology* **68**, 279-305,
doi:10.1146/annurev.physiol.68.040504.094635 (2006).
- Zhang, X.-Y. *et al.* Different neuronal populations mediate inflammatory pain
analgesia by exogenous and endogenous opioids. *eLife* **9**,
doi:10.7554/eLife.55289 (2020).
- Browning, K. N., Kalyuzhny, A. E. & Travagli, R. A. Opioid peptides inhibit
excitatory but not inhibitory synaptic transmission in the rat dorsal motor
nucleus of the vagus. *J Neurosci* **22**, 2998-3004, doi:10.1523/jneurosci.22-08-
02998.2002 (2002).
- Fülling, C., Dinan, T. G. & Cryan, J. F. Gut Microbe to Brain Signaling: What
Happens in Vagus.... *Neuron* **101**, 998-1002,
doi:10.1016/j.neuron.2019.02.008 (2019).
- Avetisyan, M., Schill, E. M. & Heuckeroth, R. O. Building a second brain in
the bowel. *J Clin Invest* **125**, 899-907, doi:10.1172/jci76307 (2015).
- Camilleri, M. Gastrointestinal motility disorders in neurologic disease. *J Clin*
*Invest* **131**, doi:10.1172/jci143771 (2021).
- Finnegan, T. F., Chen, S. R. & Pan, H. L. Mu opioid receptor activation inhibits
GABAergic inputs to basolateral amygdala neurons through Kv1.1/1.2 channels.
*J Neurophysiol* **95**, 2032-2041, doi:10.1152/jn.01004.2005 (2006).
- Gillis, A. *et al.* Critical Assessment of G Protein-Biased Agonism at the μ -
Opioid Receptor. *Trends Pharmacol Sci* **41**, 947-959,
doi:10.1016/j.tips.2020.09.009 (2020).
- Cuitavi, J., Hipólito, L. & Canals, M. The Life Cycle of the Mu-Opioid Receptor.
*Trends Biochem Sci* **46**, 315-328, doi:10.1016/j.tibs.2020.10.002 (2021).
- Prescott, S. L. & Liberles, S. D. Internal senses of the vagus nerve. *Neuron* **110**,
579-599, doi:10.1016/j.neuron.2021.12.020 (2022).
- Stewart, J. J., Weisbrodt, N. W. & Burks, T. F. Central and peripheral actions
of morphine on intestinal transit. *J Pharmacol Exp Ther* **205**, 547-555 (1978).
- Travagli, R. A. & Anselmi, L. Vagal neurocircuitry and its influence on gastric
motility. *Nat Rev Gastroenterol Hepatol* **13**, 389-401,
doi:10.1038/nrgastro.2016.76 (2016).

- Pan, Q. *et al.* Representation and control of pain and itch by distinct prefrontal
neural ensembles. *Neuron* **111**, 2414-2431.e2417,
doi:10.1016/j.neuron.2023.04.032 (2023).
- Costa, M. *et al.* Motor patterns in the proximal and distal mouse colon which
underlie formation and propulsion of feces. *Neurogastroenterol Motil* **33**,
e14098, doi:10.1111/nmo.14098 (2021).
- Parthasarathy, G. *et al.* Relationship Between Microbiota of the Colonic
Mucosa vs Feces and Symptoms, Colonic Transit, and Methane Production in
Female Patients With Chronic Constipation. *Gastroenterology* **150**, 367-
379.e361, doi:10.1053/j.gastro.2015.10.005 (2016).
- Lin, Y. M. *et al.* An opioid receptor-independent mechanism underlies motility
dysfunction and visceral hyperalgesia in opioid-induced bowel dysfunction. *Am*
*J Physiol Gastrointest Liver Physiol* **320**, G1093-g1104,
doi:10.1152/ajpgi.00400.2020 (2021).
- Browning, K. N. & Travagli, R. A. Central nervous system control of
gastrointestinal motility and secretion and modulation of gastrointestinal
functions. *Compr Physiol* **4**, 1339-1368, doi:10.1002/cphy.c130055 (2014).
- Dreborg, S., Sundström, G., Larsson, T. A. & Larhammar, D. Evolution of
vertebrate opioid receptors. *Proc Natl Acad Sci U S A* **105**, 15487-15492,
doi:10.1073/pnas.0805590105 (2008).
- Stevens, C. W. The evolution of vertebrate opioid receptors. *Front Biosci*
*(Landmark Ed)* **14**, 1247-1269, doi:10.2741/3306 (2009).
- Bunzow, J. R. *et al.* Molecular cloning and tissue distribution of a putative
member of the rat opioid receptor gene family that is not a mu, delta or kappa
opioid receptor type. *FEBS Lett* **347**, 284-288, doi:10.1016/0014-
5793(94)00561-3 (1994).
- Reinscheid, R. K. *et al.* Orphanin FQ: a neuropeptide that activates an opioidlike
G protein-coupled receptor. *Science* **270**, 792-794,
doi:10.1126/science.270.5237.792 (1995).
- Thompson, A. A. *et al.* Structure of the nociceptin/orphanin FQ receptor in
complex with a peptide mimetic. *Nature* **485**, 395-399,
doi:10.1038/nature11085 (2012).
- Calo, G. & Lambert, D. G. Nociceptin/orphanin FQ receptor ligands and
translational challenges: focus on cebranopadol as an innovative analgesic. *Br*
*J Anaesth* **121**, 1105-1114, doi:10.1016/j.bja.2018.06.024 (2018).
- Zhang, J. *et al.* Epithelial Gasdermin D shapes the host-microbial interface by
driving mucus layer formation. *Sci Immunol* **7**, eabk2092,
doi:10.1126/sciimmunol.abk2092 (2022).
- Pelaseyed, T. *et al.* The mucus and mucins of the goblet cells and enterocytes
provide the first defense line of the gastrointestinal tract and interact with the
immune system. *Immunol Rev* **260**, 8-20, doi:10.1111/imr.12182 (2014).

- Yang, D. *et al.* Nociceptor neurons direct goblet cells via a CGRP-RAMP1 axis
to drive mucus production and gut barrier protection. *Cell* **185**, 4190-
4205.e4125, doi:10.1016/j.cell.2022.09.024 (2022).
- Bergstrom, K. S. *et al.* Muc2 protects against lethal infectious colitis by
disassociating pathogenic and commensal bacteria from the colonic mucosa.
*PLoS Pathog* **6**, e1000902, doi:10.1371/journal.ppat.1000902 (2010).
- van der Post, S. *et al.* Structural weakening of the colonic mucus barrier is an
early event in ulcerative colitis pathogenesis. *Gut* **68**, 2142-2151,
doi:10.1136/gutjnl-2018-317571 (2019).
- Chang, H. *et al.* Stress-sensitive neural circuits change the gut microbiome via
duodenal glands. *Cell* **187**, 5393-5412.e5330, doi:10.1016/j.cell.2024.07.019
(2024).
- Lai, H. *et al.* Effects of dietary fibers or probiotics on functional constipation
symptoms and roles of gut microbiota: a double-blinded randomized placebo
trial. *Gut Microbes* **15**, 2197837, doi:10.1080/19490976.2023.2197837 (2023).
- Wang, L. *et al.* A randomised, double-blind, placebo-controlled trial of
*Bifidobacterium bifidum* CCFM16 for manipulation of the gut microbiota and
relief from chronic constipation. *Food Funct* **13**, 1628-1640,
doi:10.1039/d1fo03896f (2022).
- Sarosiek, I. *et al.* Lubiprostone Accelerates Intestinal Transit and Alleviates
Small Intestinal Bacterial Overgrowth in Patients With Chronic Constipation.
*Am J Med Sci* **352**, 231-238, doi:10.1016/j.amjms.2016.05.012 (2016).
- Jalanka, J. *et al.* The Effect of Psyllium Husk on Intestinal Microbiota in
Constipated Patients and Healthy Controls. *Int J Mol Sci* **20**,
doi:10.3390/ijms20020433 (2019).

**Response to Reviewers**

**Manuscript ID:** NCOMMS-25-01400A

**Title:** A brain-to-small intestine circuit mediates morphine-induced constipation in male mice

We sincerely appreciate the time and efforts of the Editor and Reviewers in evaluating our study.

As suggested, we have thoroughly revised the manuscript and incorporated these suggestions

where appropriate. The revised manuscript with tracked changes (**highlighted in blue**) has been

uploaded as a separate file. The detailed changes and our point-by-point responses to the

Reviewers' comments are presented below.

**REVIEWER COMMENTS**

**Reviewer #1 (Remarks to the Author):**

The authors have sufficiently revised the manuscript. I congratulate the authors for the hard
work and support the publication of this manuscript.

**Response:** We appreciate the Reviewer's thoughtful comments throughout the review process.

**Reviewer #2 (Remarks to the Author):**

Thank you for largely addressing my points, and I welcome the addition of your new
experiments which strengthen the paper. However, I still have some concerns about the calcium
imaging experiments.

1) I am confused by some of the language you added to describe your cell inclusion criteria:
"The manual verification protocol required precise temporal alignment between somatic
fluorescence signal transients in imaging videos and the timing of onset of corresponding
calcium events in trace plots (as illustrated below). Temporally mismatched cellular traces were
classified as false-positive identifications and rejected, while those exhibiting complete co-
occurrence were validated as accurate detections and accepted." (In 809-814 of the response).
What do you mean by aligning 'somatic fluorescent signal transients' and 'onset of
corresponding calcium events in trace plots'? Do you just mean that you manually marked
calcium events (if so, there are established algorithms for this, which can identify transients
across conditions in an unbiased way which should be applied). The referenced figure is just a
screen shot of Inscopix software and a table with raw numbers highlighting the fact that a cell
was manually rejected which does not clarify anything. Additionally, what is a temporally
mismatched cellular trace? Are you referencing a calcium trace with atypical rise/decay
dynamics? A cell footprint with low correlation among pixels? Please clarify this language.

**Response:** We regret any confusion caused by the description in our previous response. The
statement was intended to mean we performed a multi-step verification process following
automated processing, not manual marking of calcium events. We have attached the User Guide
for processing calcium imaging data with Inscopix Data Processing Software as a supplemental
file. Specifically, we first corrected motion artifacts using Inscopix through the following
sequential steps: i) defective pixel correction and file size reduction during preprocessing; ii)
spatial frequency filtering; and iii) motion correction via frame registration based on normalized
cross-correlation. After these steps, iv) signal was normalized as $\Delta F/F$, and v) potential cells
were initially identified using a hybrid of principal component analysis/independent component
analysis (PCA/ICA) algorithms^[1, 2]. Notably, this PCA/ICA approach is also used to identify
signals from independent components in video imaging data. These signals correspond to
spatially filtered images, which define target cell locations, and temporal traces, which define

calcium fluorescent signals produced by target cells (User Guide, p. 56). The final step was
conducted to reject components with abnormal signals and artifacts, ensuring the authenticity
and reliability of the data^[3] (User Guide, p. 16).

While each component ideally represents an in-focus neuronal soma, the identification
process also captures confounding components which can arise from out-of-focus somata, non-
somatic structures (e.g., dendrites or blood vessels), or non-biological sources (e.g., fluctuations
in background fluorescence), all of which are considered abnormal signals or artifacts (User
Guide, p. 56). As the Reviewer noted, we followed the above processing workflow with a
manual verification protocol wherein we examined those calcium traces that precisely aligned
with the timing of transient somatic fluorescent signals in video imaging data. Those calcium
traces that showed atypical rise/decay dynamics were also excluded as artifacts, as they
commonly originate from hemodynamic function of mouse vasculature (See Videos S2 and S3).
By manually sorting these putative cells based on $\Delta F/F$ values and cellular morphology
determined by source pixel locations, we could identify high-quality target cells^[4]. This signal
processing approach combines the objectivity of systematic algorithms with rational manual
inspection, which is widely used (and necessary) for accurately processing calcium imaging
data^[5-7].

2) Thank you for attaching the supplemental videos showing raw imaging. However, as
suspected for freely moving imaging experiments in deep brain structures, there are some
serious motion artifacts present (see for example ~23s into supplemental video 2. These sorts
of large motion can be erroneously detected as calcium events, and since they will occur in all
of the cells at a given time point can skew results (imaging that after receiving morphine the
animal sat more still and so had fewer motion artifacts, and therefore fewer detected calcium
events). Your methods do not describe any exclusion criteria for motion artifacts (such as
eliminating time windows from analysis surrounding unstable imaging times), but this should
be incorporated.

**Response:** We thank the Reviewer for raising this important point regarding motion artifacts.
We would like to clarify that our preprocessing pipeline does, in fact, include a motion
correction step, as described in the software documentation (User Guide, p. 51). Specifically,
for each frame of video imaging data, the motion correction algorithm estimates a translational
shift that minimizes differences from a reference frame through a previously described image
registration method^[7]. This process automatically identifies a valid field of view that is stable
across all displaced frames following alignment, ensuring that a given pixel consistently
corresponds to the same physical location throughout the recording. These procedures are
designed to correct for motion artifacts (User Guide, p. 52). Furthermore, as the Reviewer

rightly mentioned, in addition to this motion correction and PCA/ICA processing, we also
manually checked F -values in the final dataset, which further allowed us to exclude time
windows associated with unstable imaging (such as pronounced motion) from subsequent
analysis.

We have included these extended details in the revised Methods section of the manuscript.

3) In your response you indicate that ~30 cells were detected per animal then eliminated
manually to get to ~10. While some manual curation is typical in the field, eliminating 2/3 of
the cells is higher than typical, and I worry that some of the manual selection could be skewing
the results. Additionally, you only reported 25-28 cells for each of the groups but stated that 5
animals were in each group. This points to closer to an average of 5-6 cells per animal. Much
of the criteria you mentioned using to manually reject cells could be automated (for example
calculating the circularity or distance to the lens edge of a given footprint and setting reported
cutoff values) for inclusion that can be applied evenly to all datasets. Again, while I understand
that manual curation can be common in the field, when it is used to this extent the worry is that
inclusion or exclusion of individual data points might be severely skewing your results.

**Response:** We agree with the Reviewer's concerns that excluding a large proportion of initially
detected components could introduce bias. We would like to clarify the context and rationale
behind our procedure. Indeed, our manual exclusion rate was higher than that typically reported
in studies of cortical regions. We attribute this difference primarily to two factors inherent to
imaging deep brainstem nuclei, such as the DMV, in contrast to larger, more superficial cortical
areas reported in other studies (e.g., the prefrontal cortex^[7] or somatosensory cortex^[8]). The
DMV is a compact, slender nucleus located in a region susceptible to significant physiological
noise, most notably from cardiorespiratory pulsations^[9, 10], which generate pervasive, structured
background artifacts that are more severe and frequent than in stable cortical imaging.
Consequently, our initial PCA/ICA decomposition in the commercial software (designed for a
broad user base) yields a substantial number of components reflecting these motion-derived
artifacts and out-of-focus signals from adjacent tissue, which must be rigorously filtered out.
This technical issue thus presents a strong demand for new, tailored algorithms that are
specifically optimized for challenging, deep-brain imaging contexts.

We agree with the Reviewer that establishing more objective, automated criteria is a
crucial goal for the field. To enhance transparency and reproducibility, we have expanded our
description in the revised Methods section to explicitly detail the specific spatial (e.g., non-
somatic shape, low pixel correlation) and temporal (e.g., atypical calcium transient kinetics,
high correlation with background fluctuations) criteria used for data exclusion.

In addition, as noted in our response to your comment #4 during the last round of review:
“Regarding the Reviewer’s concern about sample size, we would like to clarify that all calcium
imaging data were acquired from 5 mice per group. The relatively small number of detected
neurons per animal reflects the anatomical constraints of the DMV and PVN, as well as the
need to use minimal viral volumes in order to restrict expression within these small target
regions. Furthermore, our viral strategies specifically labelled DMV^{Ach} neurons and DMV-
projecting PVN neurons, which also limited the number of cells that could be imaged in each
mouse. It warrants emphasis that our goal was not to conduct large-scale neuronal population
analyses, but rather to examine changes in morphine-responsive activity within anatomically
and functionally defined circuits, an approach that is consistent with previous studies^{[11, 12]”}.

We again deeply thank the Reviewer for the careful review and extremely helpful guidance
towards improving our study.

**References**

- 1. Mukamel, E.A., A. Nimmerjahn, and M.J. Schnitzer, *Automated analysis of cellular signals*
*from large-scale calcium imaging data*. Neuron, 2009. **63**(6): p. 747-60.
- 2. Kwan, A.C., *Toward reconstructing spike trains from large-scale calcium imaging data*. Hfsp
j, 2010. **4**(1): p. 1-5.
- 3. Tran, L.M., et al., *Automated Curation of CNMF-E-Extracted ROI Spatial Footprints and*
*Calcium Traces Using Open-Source AutoML Tools*. Front Neural Circuits, 2020. **14**: p. 42.
- 4. Huang, J.Y., et al., *Intra-somatosensory cortical circuits mediating pain-induced analgesia*. Nat
Commun, 2025. **16**(1): p. 1859.
- 5. Pan, Q., et al., *Representation and control of pain and itch by distinct prefrontal neural*
*ensembles*. Neuron, 2023. **111**(15): p. 2414-2431.e7.
- 6. Deng, H., et al., *A genetically defined insula-brainstem circuit selectively controls motivational*
*vigor*. Cell, 2021. **184**(26): p. 6344-6360.e18.
- 7. Wu, K., et al., *Distinct circuits in anterior cingulate cortex encode safety assessment and*
*mediate flexibility of fear reactions*. Neuron, 2023. **111**(22): p. 3650-3667.e6.
- 8. Liu, Y., et al., *Touch and tactile neuropathic pain sensitivity are set by corticospinal projections*.
Nature, 2018. **561**(7724): p. 547-550.
- 9. Sclocco, R., et al., *Challenges and opportunities for brainstem neuroimaging with ultrahigh*
*field MRI*. Neuroimage, 2018. **168**: p. 412-426.
- 10. Persson, K. and J.C. Rekling, *Population calcium imaging of spontaneous respiratory and novel*
*motor activity in the facial nucleus and ventral brainstem in newborn mice*. J Physiol, 2011.
**589**(Pt 10): p. 2543-58.

- 11. Zhu, X., et al., *Somatosensory cortex and central amygdala regulate neuropathic pain-mediated*
*peripheral immune response via vagal projections to the spleen*. Nat Neurosci, 2024. **27**(3): p.
471-483.
- 12. Dong, W.Y., et al., *Brain regulation of gastric dysfunction induced by stress*. Nat Metab, 2023.
**5**(9): p. 1494-1505.